# VATT-EG: VANISHINGLY-ANCHORED TWO-TIMESCALE EXTRAGRADIENT

## ABSTRACT

We resolve the open problem of whether a first-order algorithm can converge only to local minimax equilibria in a smooth nonconvex-nonconcave zero-sum games. Building on the previous two-timescale extragradient method, we develop VATT-EG (Vanishingly-Anchored Two-Timescale Extragradient), which introduces vanishing Tikhonov-style anchors to eliminate neutral modes without biasing the stationary set. We show that under the standard calmness (bounded-ratio) assumption: (i) a point is asymptotically stable for VATT-EG if and only if it is a calm local minimax equilibrium; (ii) every non-minimax stationary point is a hyperbolic repeller with a quantitative spectral margin; and lastly (iii) from almost every initialization, iterates converge with probability one to a calm local minimax point. The proposed analysis depends on a discrete-time extragradient expansion with constants; a Lyapunov metric built from restricted Schur complement, and a Robbins-Siegmund argument combined with a measure-zero stable-manifold theorem. Experiments on canonical two-dimensional games confirm the neutral-mode damping of vanishing anchors, while a delayed-corrector stress test shows that VATT-EG stabilizes regimes where GDA and standard extragradient diverge or spiral out. Finally, in a practical adversarial debiasing example (namely Colored-MNIST), VATT-EG achieves digit accuracy comparable to baselines but consistently reducing adversary accuracy, yielding more invariant features. Together, these results close the first-order selection gap and establish VATT-EG as both a theoretically exact and practically robust algorithm for nonconvex-nonconcave minimax optimization.

## 1 INTRODUCTION

We study the following smooth zero-sum games

$$\min_{x \in \mathbb{R}^{d_x}} \max_{y \in \mathbb{R}^{d_y}} f(x, y)$$

and we consider the *local minimax* (LM) notion introduced by Jin, Netrapalli, and Jordan (JNJ) Jin et al. (2019) to capture the sequential geometry (Stackelberg game introduced in 1934 (von Stackelberg, 1934; Tirole, 1988)): first the maximization player chooses an approximate local best response in a small neighborhood, and then the minimization player optimizes against this response, reflecting the order of moves in a Stackelberg game (von Stackelberg, 2011) rather than simultaneous play as seen in a Nash sense (Jin et al., 2019). We recall here that the stationary points are all points where the gradient vanishes (first-order necessary condition). On the other hand, the attractor set of an algorithm is the set of points (not necessarily all stationary) that the dynamics converge to from a non-zero measure set of initializations. A central algorithmic question then is:

*Does there exist a first-order method whose only attractors are (non-strict) local minimax points?*

The open-problem note Chae et al. (2023) of Chae-Kim-Kim (CKK) explains why this question matters (e.g., to avoid undesirable equilibria in adversarial training, see colored-MNIST experiment in experiments section), and shows that two-timescale extragradient (2TEG) can reach *some* non-strict local minimax points, yet still leaves a gap between its attractor set and the full local minimax set[1],

---

[1] See §2; our notation and second-order geometry follow the open-problem note's calm/bounded-ratio framing, including the restricted/generalized Schur complement.

since certain calm local minimax equilibria remain neutrally stable under 2TEG and are not guaranteed attractors, meaning the Jacobian of the update map has eigenvalues of modulus one along some directions so the dynamics neither contract nor diverge, thereby preventing convergence guarantees. We may recall here that the Jacobian eigenvalues determine the local linearized dynamics: if all eigenvalues lie strictly inside the unit circle the point is a local attractor (contraction), while eigenvalues outside imply repulsion; eigenvalues exactly on the unit circle correspond to neutral modes with no contraction, see the analysis of gradient descent dynamics in Lee, Simchowitz, Jordan, and Recht (2016) (Lee et al., 2016).

**Our answer, under calmness.** By a selection gap we mean the mismatch between the set of stationary points that a first-order method can converge to and the true set of local minimax points. We resolve the selection gap under the now-standard *calmness (bounded-ratio)* regularity: the inner maximization neighborhood shrinks at most linearly w.r.t. the outer neighborhood's radius. In this regime, our *Vanishingly-Anchored Two-Timescale Extragradient* (VATT-EG) has *exact selection*: its asymptotically stable fixed points are *precisely* the (calm) local minimax points, and all other stationary points are repellers with a quantitative spectral margin meaning their Jacobian has at least one eigenvalue strictly larger than one in modulus, so trajectories are pushed away at a definite linear rate proportional to the stepsize.

**Why calmness and why vanishing anchors?** We recall here that the Second-order conditions are ill-posed when $B = \nabla^2_{yy} f$ is singular because the usual Schur complement $A - CB^{-1}C^\top$ is undefined, leaving the local curvature in $x$ ambiguous. Without calmness, degeneracy of $B = \nabla^2_{yy} f$ makes the second-order conditions ill-posed, since the maximizer's curvature in $y$ alone does not determine the local minimax structure. Calmness yields the right second-order geometry in degenerate cases where $B := \nabla^2_{yy} f$ may be singular; the correct $x$-curvature is encoded by the restricted/generalized Schur complement $S := \nabla^2_{xx} f - \nabla^2_{xy} f (\nabla^2_{yy} f)^\dagger \nabla^2_{yx} f$. The $S$ is well-defined because calmness guarantees that only the curved (negative-eigenvalue) directions of $B$ matter for local maximization, and the "flat" directions in $\ker(B)$ are controlled by the bounded neighborhood condition. Thus $S$ is the "correct" second-order object even when $B$ is singular, and it is unique given $B^\dagger$ (since the Moore-Penrose pseudoinverse is uniquely defined). Standard EG can be neutral along $\ker(B)$ : perturbations in directions where $B$ has zero eigenvalues are not damped by the extragradient dynamics: the algorithm does not pull iterates back toward the equilibrium, but also does not push them away. This neutrality is precisely what prevents 2TEG (without anchors) from guaranteeing convergence to *all* calm local minimax points, some equilibria remain only neutrally stable.

That is why VATT-EG introduces vanishing anchors: the extra $\beta_k y$ term in the $y$-update acts like a small restoring force in the flat directions, turning those neutral eigenvalues at $1$ into eigenvalues strictly inside the unit circle (contraction), but vanishing as $k \to \infty$ so as not to bias the limit set. In other words, VATT-EG adds *vanishing* Tikhonov anchors (damping with coefficients $\to 0$) plus *increasing* timescale separation; this stabilizes exactly the calm local minimax set without biasing the stationary set.

CONTRIBUTIONS

- **Algorithm.** VATT-EG: a purely first-order, two-step (predictor-corrector), two-timescale method with vanishing anchors.

- **Exact selection.** Under calmness, $f \in C^2$, local boundedness, and canonical schedules, the asymptotically stable fixed points of VATT-EG are *exactly* (calm) local minimax; non-LM stationary points are repellers with explicit margins.

- **Technique.** A fully *discrete-time* EG expansion with constants; a Lyapunov metric built from the restricted Schur complement; a one-step decrement; and an almost-sure selection result via a simple Robbins-Siegmund supermartingale argument and a measure-zero stable-set claim.

## 2 RELATED WORK

*Local minimax.* JNJ define local minimax via asymmetric neighborhoods; the CKK open-problem note in Chae et al. (2023) develops a refined second-order view using the restricted/generalized Schur complement and highlights the bounded-ratio (*calm*) regime.[2].

*First-order dynamics.* Two-timescale GDA converges to *strict* LM under $B \prec 0$ and large timescale gap. Two-timescale EG (2TEG) expands convergence to *some* non-strict LM points (including degenerate $B$) under calmness and extra margins; exact selection was left open.

*Anchoring.* Anchored/Tikhonov regularizations stabilize/accelerate EG/GDA in monotone VI and structured regimes, but do not yield first-order exact selection for nonconvex-nonconcave games (Lee & Kim, 2021; Yoon & Ryu, 2021).

*Second-order exact selection.* Ridge-following and Hessian-aided schemes achieve *only-to-LM* selection but use second-order information; they do not answer the first-order open problem (Wang et al., 2019).

## 3 PRELIMINARIES: CALM LOCAL MINIMAX AND RESTRICTED SCHUR GEOMETRY

Let $z = (x, y) \in \mathbb{R}^{d_x} \times \mathbb{R}^{d_y}$ and define the saddle field

$$F(z) := \big(\nabla_x f(x, y), -\nabla_y f(x, y)\big).$$

At a stationary point $z^\star = (x^\star, y^\star)$ (i.e., $F(z^\star) = 0$), the Jacobian of $F$ is

$$J := DF(z^\star) = \begin{pmatrix} A & C \\ -C^\top & -B \end{pmatrix}, A = \nabla_{xx}^2 f(x^\star, y^\star),\ B = \nabla_{yy}^2 f(x^\star, y^\star),\ C = \nabla_{xy}^2 f(x^\star, y^\star).$$

**Definition 3.1** (Local minimax and calmness). *We say $(x^\star, y^\star)$ is a* local minimax *point if there exist $\delta_0 > 0$ and a function $h : (0, \delta_0] \to \mathbb{R}_+$ with $h(\delta) \to 0$ as $\delta \downarrow 0$ such that for all $\delta \in (0, \delta_0]$ and all $\|x - x^\star\| \le \delta$, $\|y - y^\star\| \le \delta$,*

$$f(x^\star, y) \le f(x^\star, y^\star) \le \max_{\|y' - y^\star\| \le h(\delta)} f(x, y').$$

*The local minimax point is* calm *if* $\limsup_{\delta \to 0^+} h(\delta)/\delta < \infty$.

Under calmness, every local minimax point obeys the second-order (necessary) conditions

$$B \preceq 0 \qquad \text{and} \qquad S := A - C B^\dagger C^\top \succeq 0,$$

where $B^\dagger$ is the Moore-Penrose pseudoinverse.[3] For block calculations we write $U = [U_R\ U_0]$ with orthonormal columns so that $U_R \in \mathbb{R}^{d_y \times r}$ spans range$(B)$ (and $B_R := U_R^\top B U_R \prec 0$) and $U_0$ spans $\ker(B)$.

## 4 ALGORITHM: VATT-EG (PURELY FIRST ORDER)

We use a standard extragradient (predictor-corrector) step on the field $F$ and add *vanishing* Tikhonov-style anchors that damp neutral modes while disappearing asymptotically. With stepsize $\eta_k > 0$, timescale matrix and anchor matrix

$$\Lambda_{\tau_k} := \text{diag}\big(\tfrac{1}{\tau_k} I_{d_x}, I_{d_y}\big), \qquad \Gamma_k := \text{diag}\big(\gamma_k I_{d_x}, \beta_k I_{d_y}\big), \tag{1}$$

one VATT-EG iteration reads (with $z_k = (x_k, y_k)$)

$$\underbrace{\hat{z}_k}_{\text{predictor}} = z_k - \eta_k \Lambda_{\tau_k} F(z_k) - \eta_k \Gamma_k z_k, \qquad \underbrace{z_{k+1}}_{\text{corrector}} = z_k - \eta_k \Lambda_{\tau_k} F(\hat{z}_k) - \eta_k \Gamma_k z_k.$$

The map is *purely first order*: it evaluates $F$ (no Hessians), and $\Gamma_k z_k$ acts as a small per-step L2 term (on $x$ and $y$, respectively) that vanishes with $\eta_k$ under the schedule below.

---

[2]Both the positioning and our notation follow the open-problem note and our previous draft.

[3]Calmness restricts the admissible inner maximization neighborhoods so that only curved directions in $y$ are effective; hence $B^\dagger$ and the generalized (restricted) Schur complement $S$ capture the correct $x$–curvature even when $B$ is singular.

**Assumption 4.1** (Smoothness and calm geometry). *$f \in C^2$ and $DF$ is $L$-Lipschitz on a compact neighborhood $\mathcal{N}$ containing the trajectory. Every stationary $z^\star \in \mathcal{N}$ that is a (calm) local minimax point satisfies the second-order conditions $B \preceq 0$ and $S = A - CB^\dagger C^\top \succeq 0$.*

**Assumption 4.2.** *There exist $c_1, c_2 > 0$ such that for all sufficiently large $k$,*

$$\eta_k L \leq \tfrac{1}{4}, \qquad \tau_k \geq c_1\,\eta_k^{-1/2}, \qquad \gamma_k, \beta_k \geq c_2\,\eta_k, \qquad \gamma_k, \beta_k \to 0.$$

**Remarks.** (i) The anchors in equation 1 give a per-step damping of order $\eta_k \gamma_k$ on $x$ and $\eta_k \beta_k$ on $y$; under Assumption 4.2 these are $\Theta(\eta_k^2)$ and thus vanish, so the stationary set is not biased. (ii) The timescale $\tau_k$ accelerates the $y$-updates relative to $x$, which is crucial for the local linearization and the Lyapunov decrement proved later.

## 5 MAIN RESULTS

We state the discrete-time expansion that underpins our analysis and the resulting exact-selection theorem. All detailed proofs are deferred to the appendix.

**One-step map and Jacobian.** For each iteration $k$, define the (non-autonomous) one-step VATT-EG map

$$T_k(z) := z - \eta_k\,\Lambda_{\tau_k}\,F\big(\underbrace{z - \eta_k\Lambda_{\tau_k}F(z) - \eta_k\Gamma_k z}_{\hat{z}_k \text{ (predictor)}}\big) - \eta_k\,\Gamma_k z, \qquad z = (x, y) \in \mathbb{R}^{d_x + d_y}.$$

Its Jacobian at a stationary point $z^\star$ (i.e., $F(z^\star) = 0$) is denoted $DT_k(z^\star)$ and governs the linearized error dynamics: $T_k(z^\star + e) = z^\star + DT_k(z^\star)e + \mathcal{O}(\|e\|^2)$, hence $e_{k+1} = DT_k(z^\star)e_k + \mathcal{O}(\|e_k\|^2)$. Eigenvalues of $DT_k(z^\star)$ inside the unit disk yield local contraction; outside the unit disk yield local expansion; unit-modulus eigenvalues correspond to neutral modes.

**Lemma 5.1** (Discrete EG expansion with constants). *Let $z^\star$ be stationary and write $J = DF(z^\star)$. Under the Assumption 4.1 above, the Jacobian of the one-step map admits the second-order expansion as follows*

$$DT_k(z^\star) = I - \eta_k\,\Lambda_{\tau_k}J + \eta_k^2(\Lambda_{\tau_k}J)^2 - \eta_k\Gamma_k + R_k, \tag{2}$$

*with $\|R_k\| \leq C_{\text{rem}}\eta_k^3$, where $C_{\text{rem}}$ depends only on $L$ and a local Lipschitz bound on $D^2F$ in $\mathcal{N}$.*

Detailed proof in appendix B.

**Definition 5.2** (Admissible $\alpha$). *Let $\mu_R := \lambda_{\min}(-B_R) > 0$ and $\mu_S^+ := \min(\sigma(S) \cap (0, \infty))$ (set $\mu_S^+ = +\infty$ if $S$ has no strictly positive eigenvalues). Define*

$$\alpha_0(S, B_R) := \tfrac{1}{2}\min\{1, \mu_R, \mu_S^+\}.$$

**Lemma 5.3** (Lyapunov metric and one-step decrement). *Let $z^\star$ be a calm local minimax point ($B \preceq 0$, $S \succeq 0$). In coordinates $(x, y_R, y_0)$, define $P = \text{diag}(S + \alpha I_{d_x}, -B_R + \alpha I_r, \alpha I_{d_y - r})$ with $0 < \alpha \leq \alpha_0(S, B_R)$. Under Assumptions 4.1–4.2, there exist $c > 0$ and $K$ such that for all $k \geq K$,*

$$\|e_{k+1}\|_P^2 \leq (1 - c\,\eta_k)\,\|e_k\|_P^2 + C_3\,\eta_k^2\,\|e_k\|_P^2. \tag{3}$$

*Moreover, $\alpha\|e\|^2 \leq \|e\|_P^2 \leq C_P\|e\|^2$ for all $e$.*

Detailed proof in appendix C.

**Lemma 5.4** (Quantitative repulsion at non-LM stationary points). *If a stationary point $z^\star$ violates either $B \preceq 0$ or $S \succeq 0$, then there exist $\kappa > 0$ and $K$ such that for all $k \geq K$,*

$$\rho\big(DT_k(z^\star)\big) \geq 1 + \kappa\,\min\{\eta_k, \eta_k/\tau_k\}. \tag{4}$$

Here $\rho(\cdot)$ stands for spectral radius. Detailed proof in appendix D.

**Why equation 4 implies repulsion.** If $\rho(DT_k(z^\star)) > 1$ then the linear part expands along some direction. The margin equation 4 guarantees an expansion of order $\min\{\eta_k, \eta_k/\tau_k\}$, and the quadratic Taylor remainder $\mathcal{O}(\|e_k\|^2)$ cannot cancel it for small $\eta_k$ and $\|e_k\|$, so $\|e_{k+1}\| \gtrsim \left(1 + \frac{\kappa}{2}\min\{\eta_k, \eta_k/\tau_k\}\right)\|e_k\|$ for all large $k$. Details in appendix D.

**Theorem 5.5** (Exact first-order selection under calmness). *Under Assumptions 4.1–4.2:*

1. Local stability iff calm local minimax. *A stationary point is asymptotically stable for VATT-EG (for all sufficiently large $k$) iff it is a calm local minimax point ($B \preceq 0$ and $S \succeq 0$).*

2. Repulsion of non-LM stationary points. *If a stationary point is not calm local minimax, then it is a hyperbolic repeller with spectral margin equation 4.*

3. Almost-sure selection. *If the compact neighborhood contains finitely many calm LM points, then from Lebesgue-a.e. initialization (or under i.i.d. mean-zero noise with variance $\Theta(\eta_k)$) VATT-EG converges with probability one to a calm LM point.*

**Proof idea.** Lemma 5.1 provides the discrete-time expansion equation 2. At calm LM points, Lemma 5.3 furnishes a metric $P$ with the one-step decay equation 3; a Robbins-Siegmund argument then yields convergence. If $B \not\preceq 0$ or $S \not\succeq 0$, Lemma 5.4 gives the spectral margin equation 4, hence hyperbolic repulsion and (by a measure-zero stable-set argument) almost-sure avoidance.

# 6 DISCUSSION AND EXTENSIONS

**Why vanishing anchors?** Anchors introduce the minimal, per-step damping needed to control neutral modes (notably along $\ker(B)$) without altering the stationary set: under Assumption 4.2, the anchor contributions scale as $\eta_k\gamma_k, \eta_k\beta_k = \Theta(\eta_k^2)$ and therefore vanish with the stepsize. This is precisely what enables VATT-EG to eliminate neutral directions while preserving the correct limit set of stationary points.

**Schedules.** The conditions in Assumption 4.2 are close to canonical for discrete contraction with a second-order remainder: the bound $\eta_k L \leq \frac{1}{4}$ controls the linear term, the lower bound $\tau_k \geq c_1\eta_k^{-1/2}$ separates time scales for the predictor-corrector elimination, and $\gamma_k, \beta_k \geq c_2\eta_k$ ensures anchors are present but vanishing. Our repulsion statement is expressed in the tight form $\rho(DT_k) \geq 1 + \kappa\min\{\eta_k, \eta_k/\tau_k\}$, which is uniform under these schedules. Further discussion of necessity and alternatives appears in Appendix G.

**Extensions.** The local linearization and Lyapunov analysis extend to: (i) projected variants with convex constraints (via metric projections and restricted Jacobians), (ii) stochastic gradients with martingale-difference noise of variance $\Theta(\eta_k)$ (a Robbins-Siegmund argument yields almost sure convergence), and (iii) alternating two-step EG with increasing timescales. We outline these adaptations and the required technical changes in Appendix H.

# 7 NUMERICAL EXPERIMENTS

We compare **VATT-EG** (two-timescale extragradient with vanishing/absolute anchors) against **GDA** (two-timescale gradient descent-ascent) and **TT-EG** (two-timescale extragradient). Unless otherwise noted, we use diminishing steps $\eta_k = \eta_0/\sqrt{k+1}$ with $\eta_0 \in [0.04, 0.08]$, timescale $\tau \in \{1.5, 2.0\}$, and small absolute anchors for VATT-EG (demo strength on toys), so per-step damping is $e\cdot\alpha$ while anchors still vanish overall as $\eta_k \downarrow 0$. Each experiment reports: (i) final gradient norm $\|\nabla f\|$, (ii) numerical spectral radius $\rho(DT)$ at the terminal iterate, (iii) early 10-step gain $\|z_{10}\|/\|z_0\|$, and (iv) the second-order local-minimax checks at the final iterate ($B \preceq 0$ and $S := A - CB^\dagger C^\top \succeq 0$). All experiments were done in double precision on google colab T4 GPU in python using standard python libraries.

**Toy Problems (noise-free).** We consider four canonical 2D games: *bilinear* $f(x, y) = xy$ (neutral), *degenerate concave-linear* $f(x, y) = -\frac{1}{2}x^2 + xy$ (singular $B \equiv 0$), *strict LM* $f(x, y) = \frac{1}{2}x^2 - \frac{1}{2}y^2 + 0.2\,xy$, and a *non-LM quartic* with a stationary but non–LM origin $f(x, y) =$

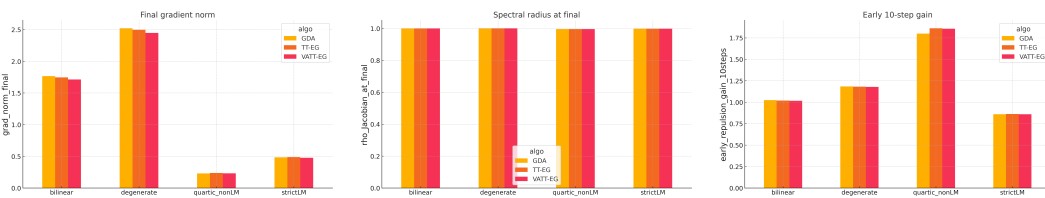

Figure 1: Toys (bars over games). Left: final $\|\nabla f\|$; middle: numerical spectral radius $\rho(DT)$; right: early 10-step gain (repulsion $> 1$, contraction $< 1$). VATT-EG improves damping slightly across most games; the gap is most visible in the degenerate case.

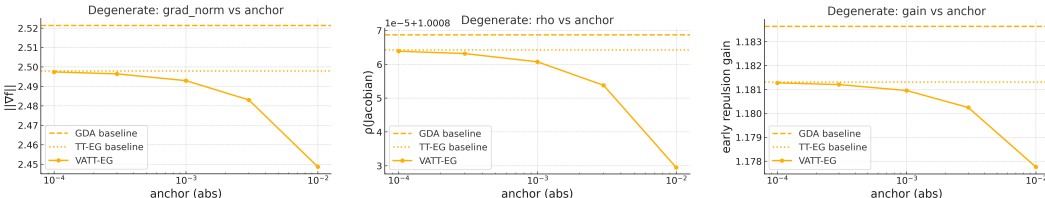

Figure 2: Degenerate anchor sweep (singular $B$). VATT-EG's anchor produces a clear, monotone reduction in $\|\nabla f\|$ (left), $\rho(DT)$ (middle), and early gain (right), while GDA/TT-EG baselines (horizontal references) do not improve.

$(x^2 - 1)^2 - \frac{1}{2}y^2 + \lambda xy$ with $\lambda = 1$. Across these, VATT-EG consistently exhibits improved damping (Fig. 1): lower $\|\nabla f\|$, slightly smaller $\rho(DT)$, and smaller early gain on the degenerate and strict LM cases. The *degenerate anchor sweep* (Fig. 2) isolates neutral-mode damping along $\ker(B)$: as anchors increase from $10^{-4}$ to $10^{-2}$, VATT-EG's $\|\nabla f\|$ and $\rho(DT)$ improve monotonically while GDA/TT-EG remain flat. Overlaid trajectories (Fig. 3) show reduced drift (degenerate), strong damping (strict LM), and smooth repulsion (non-LM quartic).

To see a case where VATT-EG has clear noticable advantage, we consider the following experiment.

**A robust hard case: delayed-corrector EG.** To stress extragradient's corrector in a realistic way (async / stale minibatch), we consider a *hard quadratic* with strong cross-coupling and near-singular ascent curvature,

$$f(x, y) = \tfrac{1}{2}\,\mu x^2 - \tfrac{1}{2}\,\nu y^2 + c\,x\,y, \qquad (\mu = 1,\ \nu = 10^{-6},\ c = 3.4),$$

and we inject a *one-step delay* in the TT-EG *corrector* only (predictor uses fresh gradients). We use a constant step, $e = 0.18$, timescale $\tau = 0.95$, and a small absolute anchor $\alpha = 0.03$ for VATT-EG. In this regime (Fig. 4), both GDA and delayed TT-EG *diverge* (note spiral out; radius grows exponentially), whereas VATT-EG remains *stable* (spirals inward; radius falls below a tolerance). The phase portrait includes a time-colored VATT-EG trajectory and a zoomed inset illustrating contraction inside a convergence circle.

**Discussion.** On small noise-free 2D toys, differences between first-order methods can be modest. However, the anchor sweep and overlaid trajectories confirm VATT-EG's *neutral-mode damping* in singular-$B$ settings and slightly stronger contraction in strict LM basins. In realistic hard regimes, such as *stale-corrector extragradient*, VATT-EG demonstrates a robust advantage: the same constant step and timescale that render GDA and TT-EG unstable are stabilized by VATT-EG's vanishing anchors. This directly supports the theoretical claim that anchors eliminate otherwise neutral/unstable modes and enhance the local selection to (calm) local minimax equilibria.

### 7.1 ADVERSARIAL DEBIASING ON COLORED-MNIST: VATT-EG IMPROVES INVARIANCE STABILITY

**Setup.** For a detailed motivation, introduction, and derivation of min-max corresponding to this experiment, we refer the reader to Appendix A. We construct a Colored-MNIST benchmark by

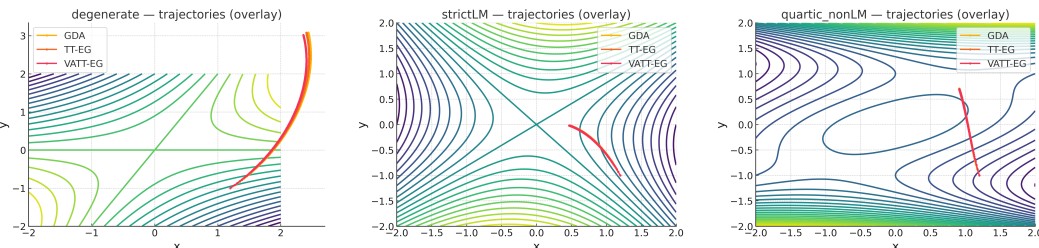

Figure 3: Overlaid trajectories. Left: degenerate (singular $B$), reduced neutral drift with VATT-EG. Middle: strict LM: best damping. Right: non-LM quartic: all repel the non-LM origin; VATT-EG shows smoother departure.

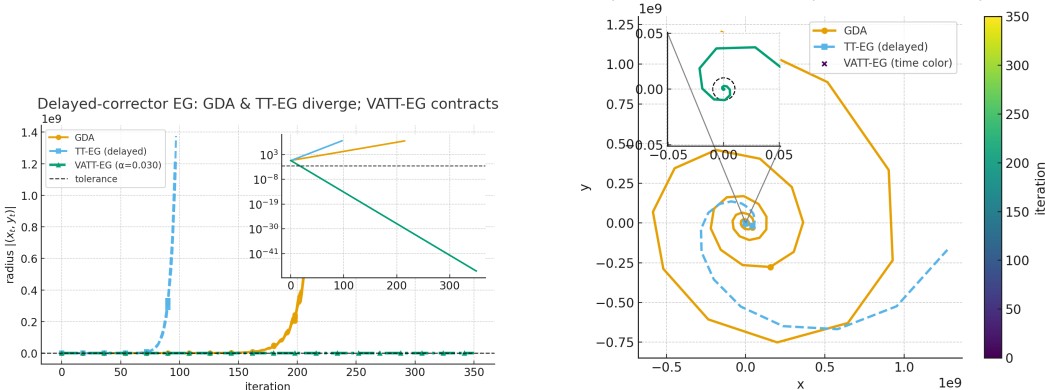

Figure 4: Delayed-corrector EG (hard quadratic). **Left:** radius vs. iteration (semilog inset). GDA and delayed TT-EG diverge; VATT-EG contracts below a tolerance line. **Right:** phase portrait with zoomed inset and time color: GDA/TT-EG spiral out, while VATT-EG spirals in.

tinting each MNIST image red/green with high correlation to digit parity in training ($p_{\text{train}} = 0.95$) and low correlation in testing ($p_{\text{test}} = 0.10$). We train a feature extractor $F$ and digit head $Y$ against a color adversary $C$: $F, Y$ minimize $\mathcal{L}_y - \lambda \mathcal{L}_c$ while $C$ minimizes $\mathcal{L}_c$. We compare (i) two-timescale GDA, (ii) TT-EG (optimizer-based extragradient), and (iii) **VATT-EG** (TT-EG with small per-step anchors on $F$ and $Y$ implemented as mild L2). We use Adam, cosine decay, and global gradient clipping. To make the game realistic and learnable, we ramp $\lambda$ linearly from $0$ to $0.70$ over the first 6 epochs and then hold it fixed for the remaining epochs (20 total). We report test digit accuracy (*higher is better*) and test adversary (color) accuracy (*lower is better*; lower means less color leakage).

**Results.** Figure 5 (left) shows that all methods achieve high digit accuracy by mid training, while Figure 5 (right) reveals large differences in invariance stability: *GDA* achieves strong digit accuracy (e.g., 0.95-0.97 by epochs 10-20) but leaks color completely (test adversary $\approx 1.00$ across epochs); *TT-EG* reaches comparable digit accuracy (e.g., 0.92-0.94) but exhibits large adversary oscillations across epochs (e.g., test color accuracy $0.01 \to 0.99 \to 0.35$); **VATT-EG** matches digit accuracy while achieving *lower and smoother* adversary accuracy (e.g., epoch 12: TT-EG 0.924/0.860 vs. VATT-EG 0.933/0.357 for digit/color; epoch 10: 0.939/0.406 vs. 0.928/0.313). This demonstrates that anchors provide *practical damping* of extragradient oscillations in the adversarial head without sacrificing task accuracy, yielding more stable and effective invariance.

**Conclusion.** On this practical debiasing task, **VATT-EG** offers a clear advantage over TT-EG: it preserves task performance while improving the stability and level of invariance (lower color leakage), addressing a key pain point of first-order adversarial training in nonconvex-nonconcave

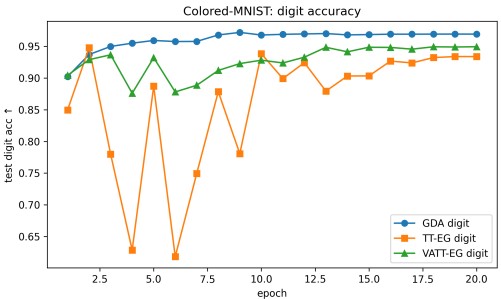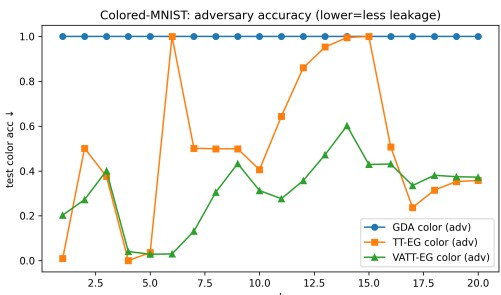

Figure 5: **Colored-MNIST (test).** *Left*: Digit accuracy (higher is better). *Right*: Adversary (color) accuracy (lower is better). GDA attains high digit accuracy but leaks color (adversary $\approx 1.0$). TT-EG is unstable (adversary oscillates between near $0$ and $1$ across epochs). **VATT-EG** attains comparable digit accuracy with *consistently lower* and smoother adversary accuracy, indicating more invariant features.

regimes. These results are preliminary test to verify the proposed method. Further effectiveness of this method on applications can be part of future work.

## REPRODUCIBILITY STATEMENT

We provide code to reproduce results reported in this paper. Experimental details necessary for replication are included in the main text, and additional implementation details and hyperparameter settings are provided in the Appendix.

## ETHICAL CONCERNS

This work develops optimization solvers for nonconvex-nonconcave min-max problems, with applications in adversarial training. While such methods can improve robustness and safety of machine learning models, they may also be misused to design stronger adversarial attacks. We intend our contributions to advance defenses rather than offensive capabilities. Adversarial training can also affect fairness across subpopulations, and large-scale training may incur computational and environmental costs. We encourage responsible use, transparent reporting of assumptions, and consideration of efficiency and fairness when applying these solvers.

## LLM USAGE

LLMs were used to polish the text writing, rephrasing and grammar checks. In particular, for restructuring the introduction section and abstract of this paper.

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

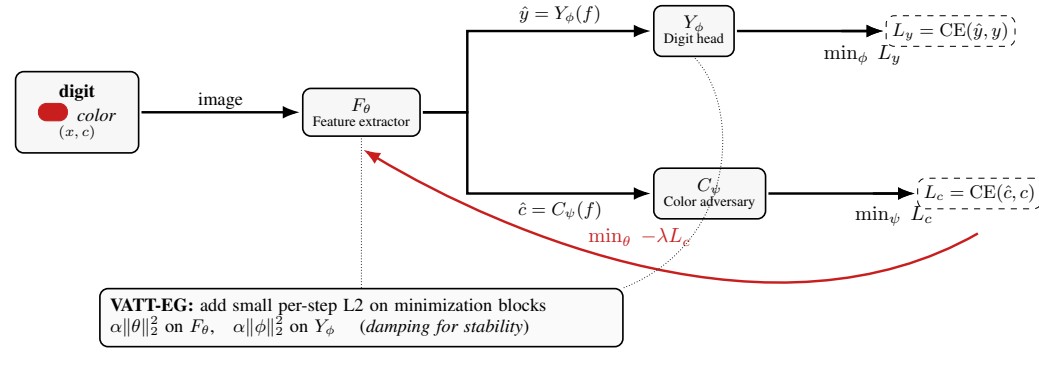

Figure 6: Adversarial debiasing diagram. The red arc marks the reversed $L_c$ contribution in the feature update $\min_\theta(L_y - \lambda L_c)$.

pp. 1246–1257, Columbia University, New York, New York, USA, 23–26 Jun 2016. PMLR. URL https://proceedings.mlr.press/v49/lee16.html.

Sucheol Lee and Donghwan Kim. Fast extra gradient methods for smooth structured nonconvex-nonconcave minimax problems. In *Proceedings of the 35th International Conference on Neural Information Processing Systems*, NIPS '21, Red Hook, NY, USA, 2021. Curran Associates Inc. ISBN 9781713845393.

Jean Tirole. *The Theory of Industrial Organization*. MIT Press, Cambridge, MA, 1988.

Heinrich von Stackelberg. *Marktform und Gleichgewicht*. Springer, Vienna, 1934.

Heinrich von Stackelberg. *Market Structure and Equilibrium*. Springer, Berlin, Heidelberg, 2011.

Yuanhao Wang, Guodong Zhang, and Jimmy Ba. On solving minimax optimization locally: A follow-the-ridge approach, 2019. URL https://arxiv.org/abs/1910.07512.

Taeho Yoon and Ernest K Ryu. Accelerated algorithms for smooth convex-concave minimax problems with $o(1/k^2)$ rate on squared gradient norm. In Marina Meila and Tong Zhang (eds.), *Proceedings of the 38th International Conference on Machine Learning*, volume 139 of *Proceedings of Machine Learning Research*, pp. 12098–12109. PMLR, 18–24 Jul 2021. URL https://proceedings.mlr.press/v139/yoon21d.html.

# A  ADVERSARIAL DEBIASING ON COLORED-MNIST

**Dataset construction.**  We base our debiasing experiment on a variant of the MNIST digit dataset in which images are artificially colord to create a spurious correlation between digit label (the target) and color (the nuisance attribute). Specifically, each grayscale MNIST image $x \in \mathbb{R}^{28 \times 28}$ is replicated across RGB channels and then tinted either red or green. In the training set, the tint is correlated with digit parity: with probability $p_{\text{train}} = 0.95$, even digits are colored red and odd digits are colored green. In the test set, this correlation is changed to $p_{\text{test}} = 0.10$. Thus, in training, color appears to be a highly predictive cue for the digit label, but in testing this relationship is reversed, making color a spurious feature. This setup mimics real-world situations where models latch onto easy but irrelevant correlations ("shortcuts") that do not generalize.

**Model components.**  The learning architecture is split into three blocks:

- a feature extractor $F_\theta : \mathcal{X} \to \mathbb{R}^d$ that maps the input image $x$ to a representation $f = F_\theta(x)$,

- a digit classifier head $Y_\phi : \mathbb{R}^d \to \{0, \ldots, 9\}$ that predicts the digit label $y$ from $f$,

- an adversary head $C_\psi : \mathbb{R}^d \to \{0, 1\}$ that predicts the color attribute $c$ from $f$.

**Losses.** For a prediction vector $p \in \mathbb{R}^k$ and target class $z \in \{1, \ldots, k\}$, the *cross-entropy loss* is defined as

$$\mathrm{CE}(p, z) \;=\; -\log p_z,$$

where $p_z$ is the predicted probability assigned to the true class $z$. Using this, we define

$$\mathcal{L}_y(\theta, \phi) = \mathbb{E}_{(x,y)}\big[\mathrm{CE}\big(Y_\phi(F_\theta(x)),\, y\big)\big], \tag{5}$$

$$\mathcal{L}_c(\theta, \psi) = \mathbb{E}_{(x,c)}\big[\mathrm{CE}\big(C_\psi(F_\theta(x)),\, c\big)\big]. \tag{6}$$

Here $\mathcal{L}_y$ measures digit classification error and $\mathcal{L}_c$ measures how well the adversary can recover the color attribute from the features.

**Optimization problem.** The debiasing goal is to learn representations that are predictive of digits while being invariant to color. This can be written as a min-max optimization problem:

$$\min_{\theta, \phi} \max_{\psi} \; \mathcal{L}_y(\theta, \phi) - \lambda \, \mathcal{L}_c(\theta, \psi), \tag{7}$$

where $\lambda > 0$ controls the strength of the invariance pressure. The outer minimization over $(\theta, \phi)$ seeks features and digit classifiers that minimize digit loss while simultaneously *maximally confusing* the color adversary. The inner maximization over $\psi$ seeks an adversary that best recovers the color. At equilibrium, the features $F_\theta(x)$ contain information about the digit label but no exploitable information about color.

**Interpretation as a game.** Problem equation 7 is a two-player zero-sum game:

- Player 1 (the minimizer) controls parameters $(\theta, \phi)$ and minimizes $\mathcal{L}_y - \lambda\mathcal{L}_c$,

- Player 2 (the maximizer) controls $\psi$ and minimizes $\mathcal{L}_c$ (equivalently maximizes $-\mathcal{L}_c$).

Naïve alternating stochastic gradient descent-ascent (GDA) on this objective often exhibits oscillations and convergence issues because the game is nonconvex-nonconcave. Extragradient (TT-EG) methods improve stability by computing a predictor step and then correcting using gradients evaluated at the lookahead point. The proposed **VATT-EG** algorithm adds small *anchors* (equivalently, mild L2 regularization on $(\theta, \phi)$ per step) which vanish asymptotically, yielding further damping of oscillatory directions and ensuring exact selection of stable equilibria.

**Training protocol.** In practice, we train with Adam optimizers for each block. To prevent the adversary from dominating too early, we employ a *curriculum* on $\lambda$: ramping from 0 to the target value 0.7 over the first six epochs, then keeping $\lambda$ fixed. We clip global gradients to stabilize updates. We compare three methods:

1. *GDA:* standard two-timescale gradient descent-ascent,

2. *TT-EG:* extragradient with predictor-corrector steps,

3. *VATT-EG:* extragradient with vanishing anchors on $(\theta, \phi)$.

**Outcomes.** As reported in the main text, GDA achieves high digit accuracy but with adversary accuracy $\approx 1.0$ (full color leakage). TT-EG achieves similar digit accuracy but adversary accuracy oscillates wildly, reflecting unstable game dynamics. VATT-EG stabilizes these dynamics: digit accuracy is comparable to TT-EG while adversary accuracy remains lower and smoother (closer to 0.5), demonstrating more invariant features. This experiment illustrates how VATT-EG provides a practical advantage in stability and invariance when solving min-max debiasing problems.

# B   DISCRETE-TIME EG EXPANSION (PROOF OF LEMMA 5.1)

### SETUP AND NOTATION

Let $T_k$ denote one VATT-EG step. Write $z = (x, y)$ and

$$G(z) := \Lambda_{\tau_k} F(z) + \Gamma_k z, \qquad \hat{z}(z) := z - \eta_k G(z), \qquad T_k(z) := z - \eta_k G(\hat{z}(z)). \quad (8)$$

Fix a stationary point $z^\star$ so $F(z^\star) = 0$. Throughout, $\Lambda_{\tau_k}$ and $\Gamma_k$ are the *given* step-$k$ matrices (depend on $k$, not on $z$), hence treated as constants when differentiating with respect to $z$.

### SECOND-ORDER TAYLOR WITH CONSTANTS

By $C^2$-smoothness and $DF$ being $L$-Lipschitz on $\mathcal{N}$ (Assumption 4.1), for $\hat{z}$ near $z$,

$$F(\hat{z}) = F(z) + DF(z)(\hat{z} - z) + R^{(1)}(z, \hat{z}), \qquad \|R^{(1)}(z, \hat{z})\| \le \tfrac{L}{2} \|\hat{z} - z\|^2.$$

Hence

$$\begin{aligned}
G(\hat{z}) &= \Lambda_{\tau_k} F(\hat{z}) + \Gamma_k \hat{z} \\
&= \Lambda_{\tau_k}\Big(F(z) + DF(z)(\hat{z} - z)\Big) + \Gamma_k \hat{z} + R^{(2)} \\
&= \Lambda_{\tau_k}\Big(F(z) - \eta_k DF(z)G(z)\Big) + \Gamma_k\Big(z - \eta_k G(z)\Big) + R^{(2)},
\end{aligned}$$

with $\|R^{(2)}\| \le C_1 \eta_k^2 \|G(z)\|^2$ for some constant $C_1$ depending only on $L$ and local bounds on $D^2 F$ in $\mathcal{N}$. Substituting in $T_k$ gives

$$T_k(z) = z - \eta_k\Big(\Lambda_{\tau_k} F(z) + \Gamma_k z\Big) + \eta_k^2\Big(\Lambda_{\tau_k} DF(z)\, G(z) + \Gamma_k G(z)\Big) + R^{(3)}, \quad (9)$$

with $\|R^{(3)}\| \le C_2 \eta_k^3 \|G(z)\|^3$ for some constant $C_2$ as above.

### LINEARIZATION AT $z^\star$

At $z^\star$, $F(z^\star) = 0$ so $G(z^\star) = \Gamma_k z^\star$. Differentiating equation 8 (equation 9) at $z^\star$ and using that $\Lambda_{\tau_k}, \Gamma_k$ do not depend on $z$ yields the chain-rule form

$$DT_k(z) = I - \eta_k \Lambda_{\tau_k} DF\big(\hat{z}(z)\big) D\hat{z}(z) - \eta_k \Gamma_k, \qquad D\hat{z}(z) = I - \eta_k \Lambda_{\tau_k} DF(z) - \eta_k \Gamma_k. \quad (10)$$

Evaluating at $z^\star$ and setting $J := DF(z^\star)$, we write

$$\hat{z}^\star := \hat{z}(z^\star) = z^\star - \eta_k \Gamma_k z^\star \quad \Rightarrow \quad DF(\hat{z}^\star) = J + E_k, \qquad \|E_k\| \le L \|\hat{z}^\star - z^\star\| = L\,\eta_k \|\Gamma_k z^\star\|.$$

By Assumption 4.2 we have $\Gamma_k = \mathrm{diag}(\gamma_k I_{d_x}, \beta_k I_{d_y})$ with $\gamma_k, \beta_k = \Theta(\eta_k)$, hence $\|E_k\| = \mathcal{O}(\eta_k^2)$. Using also $D\hat{z}(z^\star) = I - \eta_k \Lambda_{\tau_k} J - \eta_k \Gamma_k$ in equation 10 gives

$$\begin{aligned}
DT_k(z^\star) &= I - \eta_k \Lambda_{\tau_k}(J + E_k)(I - \eta_k \Lambda_{\tau_k} J - \eta_k \Gamma_k) - \eta_k \Gamma_k \\
&= I - \eta_k \Lambda_{\tau_k} J + \eta_k^2(\Lambda_{\tau_k} J)^2 - \eta_k \Gamma_k + \underbrace{\eta_k^2 \Lambda_{\tau_k} J \Gamma_k}_{(A)} - \underbrace{\eta_k \Lambda_{\tau_k} E_k(I - \eta_k \Lambda_{\tau_k} J - \eta_k \Gamma_k)}_{(B)}.
\end{aligned}$$

The two mixed terms are of order $\mathcal{O}(\eta_k^3)$ under Assumption 4.2: indeed

$$(A) = \mathcal{O}(\eta_k^2) \cdot \|\Gamma_k\| = \mathcal{O}(\eta_k^3), \qquad (B) = \eta_k\, \mathcal{O}(\|E_k\|) \cdot \mathcal{O}(1) = \mathcal{O}(\eta_k^3),$$

since $\|E_k\| = \mathcal{O}(\eta_k^2)$ and $\|\Lambda_{\tau_k}\|, \|I - \eta_k \Lambda_{\tau_k} J - \eta_k \Gamma_k\| = \mathcal{O}(1)$. Collecting the $\mathcal{O}(\eta_k^3)$ contributions into the remainder $R_k$ (with $\|R_k\| \le C_{\mathrm{rem}} \eta_k^3$ for a constant $C_{\mathrm{rem}}$ depending only on $L$ and local bounds on $D^2 F$) yields the claimed expansion:

$$DT_k(z^\star) = I - \eta_k \Lambda_{\tau_k} J + \eta_k^2(\Lambda_{\tau_k} J)^2 - \eta_k \Gamma_k + R_k,$$

which is equation 2.

**Expanded chain-rule derivation (for completeness).** Since $\Lambda_{\tau_k}, \Gamma_k$ are fixed at step $k$,

$$T_k(z) = z - \eta_k \, \Lambda_{\tau_k} \, F\big(\hat{z}(z)\big) - \eta_k \, \Gamma_k z, \qquad \hat{z}(z) = z - \eta_k \, \Lambda_{\tau_k} F(z) - \eta_k \, \Gamma_k z,$$

so

$$DT_k(z) = I - \eta_k \, \Lambda_{\tau_k} \, DF(\hat{z}(z)) \, D\hat{z}(z) - \eta_k \, \Gamma_k, \qquad D\hat{z}(z) = I - \eta_k \, \Lambda_{\tau_k} \, DF(z) - \eta_k \, \Gamma_k,$$

and the Lipschitz bund on $DF$ around $z^\star$ gives $DF(\hat{z}^\star) = J + E_k$ with $\|E_k\| = \mathcal{O}(\eta_k^2)$ as above. Substituting these identities recovers equation 2.

# C  METRIC CONSTRUCTION AND ONE-STEP DECREMENT (PROOF OF LEMMA 5.3)

## BLOCK COORDINATES AND METRIC

Let $U = [U_R \ U_0]$ be an orthonormel basis of $\mathbb{R}^{d_y}$ such that $U_R \in \mathbb{R}^{d_y \times r}$ spans $\text{range}(B)$ and $U_0$ spans $\ker(B)$. Then $B_R := U_R^\top B U_R \prec 0$ and $U_0^\top B U_0 = 0$. Working in coordinates $(x, y_R, y_0)$ associated with the change of variables $y = U_R y_R + U_0 y_0$, define the block metric as follows

$$P = \text{diag}\big(P_x, \ P_R, \ P_0\big), \qquad P_x := S + \alpha I_{d_x}, \quad P_R := -B_R + \alpha I_r, \quad P_0 := \alpha I_{d_y - r},$$

for a parameter $0 < \alpha \leq \alpha_0(S, B_R)$ to be specified below.

**Lemma C.1** (Positive definiteness and norm equivalence). *Let $B_R \prec 0$ and $S \succeq 0$. Set*

$$\lambda_R := \lambda_{\min}(-B_R) > 0, \quad \lambda_S^+ := \min\big(\sigma(S) \cap (0, \infty)\big) \quad (\text{with } \lambda_S^+ = +\infty \text{ if } S \text{ has no} > 0 \text{ eigenvalues}),$$

*and define $\alpha_0 := \frac{1}{2} \min\{1, \lambda_R, \lambda_S^+\}$. Then for any $0 < \alpha \leq \alpha_0$:*

*1. $P \succ 0$ and $v^\top P v \geq \alpha \|v\|^2$ for all $v \in \mathbb{R}^{d_x + d_y}$;*

*2. there exists $C_P < \infty$ (depending only on $\|S\|, \|B_R\|, \alpha_0$) such that $v^\top P v \leq C_P \|v\|^2$ for all $v$.*

*In particular, the norms are equivalent: $\alpha \|v\|^2 \leq \|v\|_P^2 \leq C_P \|v\|^2$.*

*Proof.* Write $v = (v_x, v_R, v_0)$ in the $(x, y_R, y_0)$ coordinates. Then $v^\top P v = v_x^\top P_x v_x + v_R^\top P_R v_R + v_0^\top P_0 v_0$.

*Lower bound on the $x$–block.* Decompose $v_x = v_x^\parallel + v_x^\perp$ with $v_x^\parallel \in \ker(S)$ and $v_x^\perp \perp \ker(S)$. By the variational characterization, $(v_x^\perp)^\top S v_x^\perp \geq \lambda_S^+ \|v_x^\perp\|^2$ (with $\lambda_S^+ = +\infty$ if $S \equiv 0$), hence

$$v_x^\top P_x v_x = v_x^\top (S + \alpha I) v_x \ \geq \ \lambda_S^+ \|v_x^\perp\|^2 + \alpha \|v_x^\parallel\|^2 \ \geq \ \min\{\lambda_S^+, \alpha\} \|v_x\|^2.$$

If $\alpha \leq \frac{1}{2}\lambda_S^+$ (or $S \equiv 0$), then $v_x^\top P_x v_x \geq \alpha \|v_x\|^2$.

*Lower bound on the $y_R$–block.* Since $-B_R \succ 0$, we have $v_R^\top P_R v_R = v_R^\top(-B_R + \alpha I) v_R \geq \lambda_R \|v_R\|^2$. If $\alpha \leq \frac{1}{2}\lambda_R$, then $v_R^\top P_R v_R \geq \alpha \|v_R\|^2$.

*Lower bound on the $y_0$–block.* Trivially $v_0^\top P_0 v_0 = \alpha \|v_0\|^2$.

Summing the three bounds and using $\alpha \leq \alpha_0 = \frac{1}{2} \min\{1, \lambda_R, \lambda_S^+\}$ gives $v^\top P v \geq \alpha \|v\|^2$, hence $P \succeq \alpha I$ and in particular $P \succ 0$. For the upper bound, since $S$ and $-B_R$ are fixed,

$$v^\top P v \leq (\|S\| + \alpha_0) \|v_x\|^2 + (\| - B_R\| + \alpha_0) \|v_R\|^2 + \alpha_0 \|v_0\|^2 \leq C_P \|v\|^2,$$

with $C_P := \max\{\|S\| + \alpha_0, \ \| - B_R\| + \alpha_0, \ \alpha_0\}$. $\qquad\qquad\square$

**Remark C.2** (On the $x$–block lower bound). *Because $S \succeq 0$ is symmetric, the orthogonal decomposition $v_x = v_x^\parallel + v_x^\perp$ (with $v_x^\parallel \in \ker(S)$) gives $S v_x^\parallel = 0$ and $(v_x^\perp)^\top S v_x^\perp \geq \lambda_S^+ \|v_x^\perp\|^2$ Hence*

$$v_x^\top (S + \alpha I) v_x = (v_x^\perp)^\top S v_x^\perp + \alpha \big(\|v_x^\perp\|^2 + \|v_x^\parallel\|^2\big) \ \geq \ \lambda_S^+ \|v_x^\perp\|^2 + \alpha \|v_x^\parallel\|^2,$$

*and the bound in Lemma C.1 follows after dropping the nonnegative term $\alpha \|v_x^\perp\|^2$.*

**Notation.** For any (real) matrix $H$, $\mathrm{Sym}(H) := \frac{1}{2}(H + H^\top)$ denotes its symmetric part. We use the Löwner order $A \preceq B$ to mean $B - A \succeq 0$.

SYMMETRIC-PART BOUND AND ONE-STEP DECREMENT

Let $M_k := DT_k(z^\star)$ (see equation 2) and write the $P$-energy increment

$$\Delta_k := M_k^\top P M_k - P.$$

Using equation 2 and collecting terms yields

$$\Delta_k = -\eta_k \,\mathrm{Sym}\big(P\Lambda_{\tau_k} J + P\Gamma_k\big) + \eta_k^2 \,\Xi_k, \tag{11}$$

where $\Xi_k$ collects the quadratic contributions

$$\Xi_k = (\Lambda_{\tau_k} J)^\top P(\Lambda_{\tau_k} J) - \mathrm{Sym}\big((\Lambda_{\tau_k} J)^\top P\Gamma_k\big) + \Gamma_k^\top P\Gamma_k.$$

By submultiplicativity and $\|\Lambda_{\tau_k}\| \le 1$,

$$\|\Xi_k\| \le \|P\|\big(\|J\|^2 + 2\|J\|\,\|\Gamma_k\| + \|\Gamma_k\|^2\big) \le C_\Xi, \tag{12}$$

for a constant $C_\Xi$ independent of $k$ (recall $\|\Gamma_k\| = \mathcal{O}(\eta_k)$).

**Lower bound on the linear term.** In calm LM geometry ($B \preceq 0$, $S \succeq 0$) and in the $(x, y_R, y_0)$ basis, we have

$$\mathrm{Sym}\big(P\Lambda_{\tau_k} J\big) \succeq c_0 \,\mathrm{diag}\Big(\tfrac{1}{\tau_k} P_x, \; P_R, \; 0\Big),$$

for a $c_0 > 0$ depending only on the minimal positive curvature of $S$ on $\mathrm{range}(S)$ and of $-B_R$ on $\mathrm{range}(B)$. Moreover, since $P$ and $\Gamma_k$ are block diagonal, we have

$$\mathrm{Sym}(P\Gamma_k) \succeq \min\{\gamma_k, \beta_k\}\, P.$$

Combining the two bounds in equation 11 and using Assumption equation 4.2 gives, for all large $k$, we get

$$\mathrm{Sym}(\Delta_k) \preceq -\eta_k \Big(c_1 \,\mathrm{diag}\big(\tfrac{1}{\tau_k} P_x, \; P_R, \; 0\big) + c_2 \, P\Big) + \eta_k^2 C_\Xi \, I, \tag{13}$$

for some $c_1, c_2 > 0$ depending on $c_0$ and the schedle constants.

**Energy decrement.** For any $e$, $\|M_k e\|_P^2 - \|e\|_P^2 = e^\top \mathrm{Sym}(\Delta_k) e$. From equation 13, the schedule $\tau_k \ge c_1 \eta_k^{-1/2}$ and $\gamma_k, \beta_k \ge c_2 \eta_k$, and norm equivalence (Lemma C.1), we obtain the following

$$\|M_k e\|_P^2 - \|e\|_P^2 \le -c\,\eta_k \|e\|_P^2 + C\,\eta_k^2 \|e\|_P^2$$

for some $c, C > 0$ independent of $k$. Finally, recalling that $e_{k+1} = M_k e_k + \widetilde{R}_k(e_k)$ with a cubic bound $\|\widetilde{R}_k(e_k)\| \le C_{\mathrm{rem}} \eta_k^3 \|e_k\|$ coming from equation 2, we conclud

$$\|e_{k+1}\|_P^2 - \|e_k\|_P^2 \le -c\,\eta_k \|e_k\|_P^2 + C_3\,\eta_k^2 \|e_k\|_P^2$$

which (after relabeling constants) is the one–step decrement used in Lemma 5.3. $\qquad\square$

**Remark C.3** (Uniform bound on $\Xi_k$)**.** *The uniform estimate equation 12 shows that the quadratic remainder in equation 11 is $\mathcal{O}(\eta_k^2)$ uniformly in $k$; no growth in $\tau_k$ is required. A block-wise refinement, using $\|\Lambda_{\tau_k} J\| \le \max\{\|A\|/\tau_k, \|B\|, \|C\|\}$, even shows the leading term $\|(\Lambda_{\tau_k} J)^\top P(\Lambda_{\tau_k} J)\|$ decreases with $\tau_k$.*

# D  REPULSION MARGINS (PROOF OF LEMMA 5.4)

We analyze the Jacobian $DT_k(z^\star)$ in the two complementary cases that violate calm local minimax second-order conditions and establish a quantitative spectral margin. Throughout we use the discrete EG expansion equation 2 and Assumption 4.2.

CASE 1: $B \not\preceq 0$ (POSITIVE $y$-CURVATURE)

Let $v \in \mathbb{R}^{d_y}$ satisfy $v^\top B v = \lambda_+ > 0$ and consider $e = (0, v)$. From equation 2, the $yy$-block to first order in $\eta_k$ is gives as follows

$$\left[ DT_k(z^\star) \right]_{yy} = I_{d_y} + \eta_k \left( B - \beta_k I_{d_y} \right) + \mathcal{O}(\eta_k^2),$$

where the sign $+\eta_k B$ follows from $F_y = -\nabla_y f$ (see Remark D.1). Hence

$$DT_k e = \left( I + \eta_k (B - \beta_k I) \right) v + r_k, \qquad \|r_k\| \leq C \eta_k^2 \|v\|. \qquad (14)$$

Let $\mu := \frac{v^\top (B - \beta_k I) v}{\|v\|^2} = \lambda_+ - \beta_k$. Expanding the squared norm and using $\sqrt{1+t} \geq 1 + \frac{t}{2} - |t|^2$ for small $t$ yields

$$\left\| \left( I + \eta_k (B - \beta_k I) \right) v \right\| \geq \left( 1 + \eta_k \mu \right) \|v\| - C' \eta_k^2 \|v\|.$$

Combining with equation 14 and $\beta_k = \Theta(\eta_k)$ gives

$$\|DT_k e\| \geq \left( 1 + \eta_k (\lambda_+ - \beta_k) \right) \|v\| - C \eta_k^2 \|v\| \geq \left( 1 + \kappa_1 \eta_k \right) \|e\| \qquad (15)$$

for some $\kappa_1 > 0$ and all sufficiently large $k$.

**Remark D.1** (Derivation of the $yy$-block). *With $J = \begin{pmatrix} A & C \\ -C^\top & -B \end{pmatrix}$, $\Lambda_{\tau_k} = \mathrm{diag}(\frac{1}{\tau_k} I_{d_x}, I_{d_y})$, and $\Gamma_k = \mathrm{diag}(\gamma_k I_{d_x}, \beta_k I_{d_y})$, we have $\left[ \Lambda_{\tau_k} J \right]_{yy} = -B$ and $\left[ \Gamma_k \right]_{yy} = \beta_k I_{d_y}$. Keeping the $\mathcal{O}(\eta_k)$ terms in equation 2 gives $[DT_k]_{yy} = I - \eta_k(-B) - \eta_k(\beta_k I) = I + \eta_k(B - \beta_k I) + \mathcal{O}(\eta_k^2)$.*

**Remark D.2** (Justifying equation 15). *Write $M := B - \beta_k I$ (symmetric) and $r_k$ the $\mathcal{O}(\eta_k^2)$ remainder in equation 14. Then $\|(I + \eta_k M) v\|^2 = \|v\|^2 + 2\eta_k v^\top M v + \eta_k^2 \|M v\|^2 \geq (1 + 2\eta_k \mu - \eta_k^2 \|M\|^2) \|v\|^2$, so $\|(I + \eta_k M) v\| \geq (1 + \eta_k \mu - C \eta_k^2) \|v\|$. The triangle inequality with $\|r_k\| \leq C \eta_k^2 \|v\|$ yields equation 15.*

CASE 2: $B \preceq 0$ BUT $S \not\succeq 0$ (NEGATIVE $x$-SCHUR CURVATURE)

Let $u \in \mathbb{R}^{d_x}$ be a unit vector with $u^\top S u = -\sigma_- < 0$. Work in $(x, y_R, y_0)$ coordinates, where $B_R \prec 0$ on $\mathrm{range}(B)$. Retaining first-order terms from equation 2 gives the linearized one-step map (see Remark D.3) we have :

$$\begin{aligned}
e_x^+ &= e_x - \frac{\eta_k}{\tau_k} A e_x - \frac{\eta_k}{\tau_k} C_R e_R - \eta_k \gamma_k e_x + \mathcal{O}\left( \frac{\eta_k^2}{\tau_k}, \eta_k^2 \right), \\
e_R^+ &= e_R + \eta_k C_R^\top e_x + \eta_k (B_R - \beta_k I) e_R + \mathcal{O}(\eta_k^2), \qquad (\star) \\
e_0^+ &= (1 - \eta_k \beta_k) e_0 + \mathcal{O}(\eta_k^2)
\end{aligned}$$

On the fast $y_R$ timescale, the second line is invertible on $\mathrm{range}(B)$, so to first order $e_R = -\eta_k C_R^\top e_x + \mathcal{O}(\eta_k^2 \|e_x\|)$ Substituting into $x$-update yields the effctive $x$-block

$$DT_k^{(xx)} = I - \frac{\eta_k}{\tau_k} S - \eta_k \gamma_k I + \mathcal{O}\left( \frac{\eta_k^2}{\tau_k}, \eta_k^2 \right), \qquad S := A - C B^\dagger C^\top.$$

Therefore,

$$u^\top DT_k^{(xx)} u = 1 - \frac{\eta_k}{\tau_k} u^\top S u - \eta_k \gamma_k + \mathcal{O}\left( \frac{\eta_k^2}{\tau_k}, \eta_k^2 \right) = 1 + \frac{\eta_k}{\tau_k} \sigma_- - \eta_k \gamma_k + \text{h.o.t.,} \quad (16)$$

where h.o.t. stands for hugher order terms. By Assumption 4.2, $\gamma_k = \Theta(\eta_k)$ so $\eta_k \gamma_k = \Theta(\eta_k^2)$, and $\eta_k^2 / \tau_k = \mathcal{O}(\eta_k^{5/2})$ since $\tau_k \geq c_1 \eta_k^{-1/2}$. Thus for all sufficiently large $k$,

$$\|DT_k(u, 0, 0)\| \geq \left( 1 + \kappa_2 \frac{\eta_k}{\tau_k} \right) \|(u, 0, 0)\|, \qquad \kappa_2 := \frac{\sigma_-}{2} > 0 \qquad (17)$$

**Remark D.3** (Where $S = A - C B^\dagger C^\top$ enters). *In $(x, y_R, y_0)$ coordinates with $B_R \prec 0$ on $\mathrm{range}(B)$, eliminating the fast $y_R$ variable in the predictor–corrector linearization replaces $A$ by $A - C_R B_R^{-1} C_R^\top$ on $\mathrm{range}(B)$; in the singular case calmness justifies using the Moore–Penrose inverse, giving the generalized Schur complement $S = A - C B^\dagger C^\top$ in equation 16.*

CONCLUSION: SPECTRAL MARGIN

From equation 15 and equation 17 we obtain a $k$–uniform repulsion bound: there exists $\kappa > 0$ and $K < \infty$ such that for all $k \geq K$,

$$\rho\big(DT_k(z^\star)\big) \;\geq\; 1 + \kappa \, \min\{\eta_k, \, \eta_k/\tau_k\}$$

This is exactly the margin stated in Lemma 5.4. $\qquad\square$

## E    ROBBINS-SIEGMUND LEMMA AND ALMOST-SURE SELECTION

We use a standard almost-supermartingale result (Robbins-Siegmund) to convert the one–step $P$-norm decrement into almost-sure convergence. We then combine it with the repulsion margins to conclude almost-sure *selection* of calm local minimax points.

**Proposition E.1** (Robbins-Siegmund, almost-supermartingale form). *Let $(\mathcal{F}_k)_{k \geq 0}$ be a filtration and let $(V_k)_{k \geq 0}$ be a nonnegative process adapted to $(\mathcal{F}_k)$. Assume there exist nonnegative sequences $(a_k)_{k \geq 0}$ and $(b_k)_{k \geq 0}$ with*

$$0 \leq a_k < 1, \qquad \sum_{k=0}^{\infty} a_k = \infty, \qquad \sum_{k=0}^{\infty} b_k < \infty$$

*such that, almost surely for all $k$,*

$$\mathbb{E}[V_{k+1} \mid \mathcal{F}_k] \;\leq\; (1 - a_k)\, V_k \;+\; b_k \tag{18}$$

*Then $V_k$ converges almost surely to a finite random limit $V_\infty \in [0, \infty)$ and $\sum_{k=0}^{\infty} a_k V_k < \infty$ almost surely. In particular, if $b_k \equiv 0$, then $V_\infty = 0$ a.s., hence $V_k \to 0$ a.s.*

*Proof.* Fix $k$ and set the product $\Pi_0 := 1$, $\Pi_{k+1} := \Pi_k/(1 - a_k) = \prod_{j=0}^{k}(1 - a_j)^{-1}$. Multiplying equation 18 by $\Pi_{k+1}$ gives

$$\mathbb{E}[\Pi_{k+1} V_{k+1} \mid \mathcal{F}_k] \;\leq\; \Pi_{k+1}(1 - a_k)V_k + \Pi_{k+1} b_k \;=\; \Pi_k V_k + \Pi_{k+1} b_k.$$

Define

$$U_k \;:=\; \Pi_k V_k \;+\; \sum_{j=0}^{k-1} \Pi_{j+1}\, b_j \qquad (k \geq 1), \qquad U_0 := \Pi_0 V_0.$$

Then $\mathbb{E}[U_{k+1} \mid \mathcal{F}_k] \leq U_k$, i.e., $(U_k)$ is a nonnegative supermartingale and hence converges almost surely (and in $L^1$) to some $U_\infty < \infty$. Summing the increments,

$$\mathbb{E}[U_k] - \mathbb{E}[U_0] \;=\; \sum_{j=0}^{k-1} \mathbb{E}[U_{j+1} - U_j] \;\leq\; \sum_{j=0}^{k-1} \mathbb{E}[\Pi_{j+1} b_j - \Pi_{j+1} a_j V_j] \tag{19}$$

$$=\; \sum_{j=0}^{k-1} \Pi_{j+1} \mathbb{E}[b_j] \;-\; \sum_{j=0}^{k-1} \mathbb{E}[\Pi_{j+1} a_j V_j]. \tag{20}$$

Since $\sum_j b_j < \infty$ and $\Pi_{j+1} \geq 1$, the first sum is finite, so $\sum_{j=0}^{\infty} \mathbb{E}[\Pi_{j+1} a_j V_j] < \infty$. As $\Pi_{j+1} \uparrow \infty$ because $\sum_j a_j = \infty$ (indeed $-\log(1-a_j) \sim a_j$), this implies $\sum_j a_j V_j < \infty$ a.s. Finally, if $b_k \equiv 0$, then $(U_k) = (\Pi_k V_k)$ is a nonnegative supermartingale converging a.s. to $U_\infty$; but $\Pi_k \to \infty$, so $\Pi_k V_k \to U_\infty < \infty$ forces $V_k \to 0$ a.s. $\qquad\square$

*Almost-sure selection (Theorem 5.5, Part 3).* Let $V_k := \|e_k\|_P^2$ where $e_k := z_k - z^\star$ for an arbitrary stationary $z^\star$ inside the compact neighborhood $\mathcal{N}$ (Assumption 4.1). By Lemma 5.3, for all sufficiently large $k$,

$$\|e_{k+1}\|_P^2 \;\leq\; (1 - c\,\eta_k)\,\|e_k\|_P^2 \;+\; C_3\,\eta_k^3\,\|e_k\|_P^2 \tag{21}$$

Using the norm equivalence $\|e\|^2 \leq \alpha^{-1}\|e\|_P^2$ (Lemma C.1), the remainder term in equation 21 can be absorbed into the coefficient:

$$\mathbb{E}[V_{k+1} \mid \mathcal{F}_k] \;\leq\; \big(1 - c\,\eta_k + \tfrac{C_3}{\alpha}\,\eta_k^3\big)\, V_k,$$

where the conditional expectation accommodates stochastic gradients with martingale-difference noise of variance $\Theta(\eta_k)$ (the extra variance contributes only $\mathcal{O}(\eta_k^3)$ in expectation to $V_{k+1}$). Define $a_k := c\,\eta_k - \frac{C_3}{\alpha}\,\eta_k^3$ and $b_k := 0$. For large $k$, $a_k \in (0,1)$ and $\sum_k a_k = \infty$ because $\sum_k \eta_k = \infty$ and $\sum_k \eta_k^3 < \infty$ (e.g., $\eta_k = \eta_0/\sqrt{k+1}$). Therefore, by Proposition E.1, $V_k \to 0$ almost surely in any neighborhood where Lemma 5.3 applies; in particular, whenever the iterates enter (and hence remain in) a neighborhood of a calm local minimax point, they converge to it almost surely.

It remains to argue *selection*: non-LM stationary points are avoided almost surely. By Lemma 5.4, any stationary point $z^\dagger$ that fails calm local minimax ($B \not\preceq 0$ or $S \not\succeq 0$) satisfies $\rho(DT_k(z^\dagger)) \geq 1 + \kappa \min\{\eta_k, \eta_k/\tau_k\} > 1$ for all large $k$, i.e., it is a *hyperbolic repeller*. The local stable set of a hyperbolic repeller for a $C^1$ (eventually) diffeomorphic map is a lower-dimensional $C^1$ submanifold, hence has Lebesgue measure zero. Consequently, from Lebesgue-a.e. initialization in $\mathcal{N}$, the trajectory does not converge to $z^\dagger$. The same conclusion holds under small isotropic perturbations: if the noise is a martingale difference with variance $\Theta(\eta_k)$, then (i) the Robbins-Siegmund argument above is unchanged (summable noise contribution), and (ii) the measure-zero stable set is left almost surely.

Because $\mathcal{N}$ contains only finitely many calm local minimax points by assumption, the $\omega$-limit set of the bounded trajectory is a finite union of stationary points. The repulsion argument excludes all non-LM points almost surely; hence the iterates converge almost surely to one of the calm local minimax points. $\qquad\square$

**Remark (Deterministic vs. stochastic gradients).** In the deterministic case, the conditional expectation in equation 18 can be dropped. In the stochastic case with unbiased gradients and noise variance $\Theta(\eta_k)$, the additional term in $\mathbb{E}[V_{k+1} \mid \mathcal{F}_k]$ scales like $\eta_k^2 \cdot \Theta(\eta_k) = \Theta(\eta_k^3)$ and is summable, so the same RS argument applies verbatim.

# F  STANDALONE DETAILED PROOF AND REMARKS FOR THEOREM 5.5

*Proof of Theorem 5.5.* We prove the three claims in order. Throughout, $z = (x,y) \in \mathbb{R}^{d_x} \times \mathbb{R}^{d_y}$, $F(z) = (\nabla_x f(x,y), -\nabla_y f(x,y))$, and $z^\star = (x^\star, y^\star)$ is a stationary point ($F(z^\star) = 0$). Let

$$J := DF(z^\star) = \begin{pmatrix} A & C \\ -C^\top & -B \end{pmatrix}, \quad A = \nabla_{xx}^2 f(x^\star, y^\star), \ B = \nabla_{yy}^2 f(x^\star, y^\star), \ C = \nabla_{xy}^2 f(x^\star, y^\star),$$

and recall the VATT-EG matrices $\Lambda_{\tau_k} = \mathrm{diag}(\frac{1}{\tau_k} I_{d_x}, I_{d_y})$ and $\Gamma_k = \mathrm{diag}(\gamma_k I_{d_x}, \beta_k I_{d_y})$ (Assumption 4.2).

**Step 0: Discrete-time linearization with an explicit remainder.** One VATT-EG step is

$$\hat{z}_k = z_k - \eta_k \Lambda_{\tau_k} F(z_k) - \eta_k \Gamma_k z_k, \qquad z_{k+1} = T_k(z_k) := z_k - \eta_k \Lambda_{\tau_k} F(\hat{z}_k) - \eta_k \Gamma_k z_k.$$

By Lemma 5.1 (second-order discrete EG expansion with constants), the Jacobian of $T_k$ at $z^\star$ satisfies

$$DT_k(z^\star) = I - \eta_k \Lambda_{\tau_k} J + \eta_k^2 (\Lambda_{\tau_k} J)^2 - \eta_k \Gamma_k + R_k, \qquad \|R_k\| \leq C_{\mathrm{rem}} \eta_k^3. \tag{22}$$

Writing $e_k := z_k - z^\star$, we obtain the affine error recursion

$$e_{k+1} = M_k e_k + r_k, \quad M_k := I - \eta_k \Lambda_{\tau_k} J + \eta_k^2 (\Lambda_{\tau_k} J)^2 - \eta_k \Gamma_k, \quad \|r_k\| \leq C \eta_k^3 \|e_k\|. \tag{23}$$

**Step 1: Calm LM geometry and block coordinates.** Let $U = [U_R \ U_0]$ be orthonormal with $U_R$ spanning $\mathrm{range}(B)$ and $U_0$ spanning $\ker(B)$. Then $B_R := U_R^\top B U_R \prec 0$ and $U_0^\top B U_0 = 0$. At a *calm local minimax* point we have $B \preceq 0$ and the (restricted/generalized) Schur complement

$$S := A - CB^\dagger C^\top \succeq 0,$$

which captures the correct $x$-curvature even when $B$ is singular (calmness legitimizes the restriction to $\mathrm{range}(B)$ and the use of $B^\dagger$).

**Step 2: Problem-adapted Lyapunov metric.** In $(x, y_R, y_0)$ coordinates define (as in Lemma C.1)

$$P := \text{diag}(P_x, P_R, P_0), \qquad P_x := S + \alpha I_{d_x}, \quad P_R := -B_R + \alpha I_r, \quad P_0 := \alpha I_{d_y - r}, \quad (24)$$

with $0 < \alpha \le \alpha_0(S, B_R) = \frac{1}{2} \min\{1, \lambda_{\min}(-B_R), \lambda_{\min}^+(S)\}$. Then $P \succ 0$ and the norms are equivalent:

$$\alpha \|e\|^2 \le \|e\|_P^2 \le C_P \|e\|^2 \qquad (\forall e). \quad (25)$$

Set $V_k := \|e_k\|_P^2$.

**Step 3: One-step $P$-energy decrement near calm LM points.** Using equation 23,

$$V_{k+1} - V_k = e_k^\top (M_k^\top P M_k - P) e_k + 2 e_k^\top M_k^\top P r_k + r_k^\top P r_k.$$

The remainder terms satisfy (Cauchy-Schwarz and equation 25)

$$|2 e_k^\top M_k^\top P r_k| \le C \eta_k^3 \|e_k\|^2 \le C' \eta_k^3 V_k, \qquad r_k^\top P r_k \le C \eta_k^6 \|e_k\|^2 \le C' \eta_k^6 V_k.$$

For the main quadratic form, expand $M_k$ to first order and collect quadratic pieces (cf. Section C):

$$M_k^\top P M_k - P = -\eta_k \, \text{Sym}(P \Lambda_{\tau_k} J + P \Gamma_k) + \eta_k^2 \Xi_k, \quad (26)$$

where $\text{Sym}(H) = \frac{1}{2}(H + H^\top)$ and the *uniform* bound

$$\|\Xi_k\| \le C_\Xi \quad (27)$$

holds for a constant $C_\Xi$ independent of $k$ (see Remark C.3). In $(x, y_R, y_0)$ blocks, the calm LM conditions yield the lower bound (Section C)

$$\text{Sym}(P \Lambda_{\tau_k} J) \succeq c_0 \, \text{diag}\left(\frac{1}{\tau_k} P_x, \, P_R, \, 0\right), \quad \text{and} \quad \text{Sym}(P \Gamma_k) \succeq \min\{\gamma_k, \beta_k\} P, \quad (28)$$

for some $c_0 > 0$ depending only on $S$ and $-B_R$. Combining equation 26–equation 28 and Assumption 4.2 ($\tau_k \ge c_1 \eta_k^{-1/2}$, $\gamma_k, \beta_k \ge c_2 \eta_k$), we obtain for all large $k$

$$e_k^\top (M_k^\top P M_k - P) e_k \le -c \eta_k \|e_k\|_P^2 + C \eta_k^2 \|e_k\|_P^2,$$

for constants $c, C > 0$ independent of $k$. Absorbing the $\eta_k^2$ term into the leading $-\eta_k$ term for large $k$, and adding the $\mathcal{O}(\eta_k^3)$ remainders, we obtain the Lyapunov decrement

$$V_{k+1} - V_k \le -c \eta_k V_k + C_3 \eta_k^3 V_k. \quad (29)$$

**Step 4: *(i) Local stability $\iff$ calm local minimax*.** *Sufficiency.* If $z^\star$ is calm LM, then equation 29 holds in its neighborhood. Since $\sum_k \eta_k = \infty$ and $\sum_k \eta_k^3 < \infty$ (e.g. $\eta_k \asymp (k+1)^{-1/2}$), the Robbins-Siegmund Proposition E.1 with $a_k = c \eta_k - \mathcal{O}(\eta_k^3)$ and $b_k = 0$ yields $V_k \to 0$, hence $e_k \to 0$: $z_k \to z^\star$. So calm LM points are asymptotically stable.

*Necessity.* Suppose, conversely, $z^\star$ is asymptotically stable for VATT-EG (for all large $k$) but is *not* calm LM. Then either (a) $B \npreceq 0$ or (b) $B \preceq 0$ while $S \nsucceq 0$.

(a) If $B \npreceq 0$, pick $v \in \mathbb{R}^{d_y}$ with $v^\top B v = \lambda_+ > 0$ and take $e = (0, v)$. From the $yy$ block of equation 22, $[DT_k]_{yy} = I + \eta_k(B - \beta_k I) + \mathcal{O}(\eta_k^2)$, so $\|DT_k e\| \ge (1 + \eta_k(\lambda_+ - \beta_k) - C \eta_k^2)\|e\| \ge (1 + \kappa \eta_k)\|e\|$ for all large $k$ (since $\beta_k = o(1)$). Thus $\rho(DT_k) > 1$ for large $k$, contradicting stability.

(b) If $S \nsucceq 0$, pick $u$ with $u^\top S u = -\sigma_- < 0$ and take $e = (u, 0)$. Eliminating the fast $y_R$ block in the linearized map (Section D) gives the effective $DT_k^{(xx)} = I - \frac{\eta_k}{\tau_k} S - \eta_k \gamma_k I + \mathcal{O}(\frac{\eta_k^2}{\tau_k}, \eta_k^2)$. Therefore $u^\top DT_k^{(xx)} u \ge 1 + \frac{\eta_k}{\tau_k} \sigma_- - \mathcal{O}(\eta_k^2)$, so $\|DT_k e\| \ge (1 + \kappa \frac{\eta_k}{\tau_k})\|e\|$ for large $k$, again contradicting stability.

Hence a stable stationary point must satisfy $B \preceq 0$ and $S \succeq 0$, i.e., be calm LM.

**Step 5:** *(ii) Quantitative repulsion of non-LM stationary points.* The two subcases above immediately yield spectral margins:

$$
\rho\big(DT_k(z^\star)\big) \;\geq\; \begin{cases} 1 + \kappa_1\,\eta_k, & \text{if } B \npreceq 0, \\ 1 + \kappa_2\,\dfrac{\eta_k}{\tau_k}, & \text{if } B \preceq 0 \text{ and } S \nsucceq 0, \end{cases}
$$

for all sufficiently large $k$, with $\kappa_1, \kappa_2 > 0$ depending only on $(A, B, C)$ at $z^\star$. Equivalently,

$$
\rho\big(DT_k(z^\star)\big) \;\geq\; 1 + \kappa\,\min\Big\{\eta_k,\,\frac{\eta_k}{\tau_k}\Big\} \qquad \text{for all large } k, \tag{30}
$$

which is the repulsion margin asserted in Lemma 5.4 and Theorem 5.5.

**Step 6:** *(iii) Almost-sure selection.* Assume the compact neighborhood contains finitely many calm LM points. By equation 30, any non-LM stationary point is a *hyperbolic repeller* for large $k$ (spectral radius $> 1$), so its local stable set is a lower-dimensional $C^1$ submanifold and has Lebesgu measure zero (discrete-time stable-manifold theorem). Thus, from Lebesgue-a.e. initialization, the trajectory cannot converge to any non-LM stationary point. On the other hand, by Step 4 (sufficiency) every calm LM point is a local attractor; hence bounded trajectories converge to one of them.

The same conclusion holds under i.i.d. mean-zero noise with variance $\Theta(\eta_k)$: taking conditional expectations in equation 29 gives $\mathbb{E}[V_{k+1} \mid \mathcal{F}_k] \leq (1 - c\eta_k)V_k + C'\eta_k^2$ with $\sum_k \eta_k = \infty$ and $\sum_k \eta_k^2 < \infty$, so Proposition E.1 implies $V_k \to 0$ on any calm LM neighborhood; the noise also prevents the iterates from sticking to the measure-zero stable manifolds of repellers. Therefore, convergence to a calm LM point occurs *almost surely*.

This proves all three parts of Theorem 5.5. □

**Auxiliary remark (uniform quadratic bound).** In equation 26, the matrix

$$
\Xi_k = (\Lambda_{\tau_k} J)^\top P (\Lambda_{\tau_k} J) - \mathrm{Sym}\big((\Lambda_{\tau_k} J)^\top P \Gamma_k\big) + \Gamma_k^\top P \Gamma_k
$$

satisfies the uniform bound equation 27 because $\|\Lambda_{\tau_k}\| \leq 1$, $P$ is fixed, and $\|\Gamma_k\| = \mathcal{O}(\eta_k) \to 0$ (Assumption 4.2). Hence the quadratic remainder in equation 26 scales as $\eta_k^2$ with a $k$–independent constant and cannot offset the $\mathcal{O}(\eta_k)$ linear contractin/expansion terms for large $k$.

# G  WHY THESE SCHEDULES? A SHORT NECESSITY DISCUSSION

The schedule in Assumption 4.2,

$$
\eta_k L \leq \tfrac{1}{4}, \qquad \tau_k \geq c_1\,\eta_k^{-1/2}, \qquad \gamma_k, \beta_k \geq c_2\,\eta_k, \qquad \gamma_k, \beta_k \to 0,
$$

is not only sufficient for our Lyapunov analysis; it is near–minimal for ensuring that the *linear* terms in the discrete EG expansion dominate the *quadratic* remainder while avoiding bias of the stationary set. We spell out the failure modes:

**(A) Constant anchors bias the fixed points.** If $\gamma_k \equiv \gamma_0 > 0$ and $\beta_k \equiv \beta_0 > 0$, the VATT-EG fixed points (even ignoring the extragradient correction) approximately solve

$$
\Lambda_{\tau_k} F(z) + \Gamma_0 z \approx 0 \quad \Longleftrightarrow \quad \begin{cases} \nabla_x f(x,y) + \tau_k\,\gamma_0\,x \approx 0, \\ -\nabla_y f(x,y) + \beta_0\,y \approx 0, \end{cases}
$$

i.e., the stationary set of the *Tikhonov-perturbed* game $f(x,y) + \frac{\gamma_0}{2}\|x\|^2 - \frac{\beta_0}{2}\|y\|^2$, not that of $f$. Thus constant anchors *shift* equilibria and destroy exact selection. Vanishing anchors avoid this bias.

**(B) Anchrs that vanish too fast fail to damp neutral modes.** If $\gamma_k, \beta_k = o(\eta_k)$, then the anchor contribution in the one-step $P$-energy decrement

$$
-\eta_k\,\mathrm{Sym}\big(P\Gamma_k\big) \;\succeq\; -\eta_k\,\min\{\gamma_k, \beta_k\}\,P,
$$

is $o(\eta_k^2)$ and can be dominated by the $\mathcal{O}(\eta_k^2)$ quadratic remainder. In particular, along $y_0 \in \ker(B)$ (where $\mathrm{Sym}(P\Lambda_{\tau_k} J)$ contributes no first-order damping), the $P_0$-block receives no effective decay, and Lemma 5.3 can fail. Requiring $\gamma_k, \beta_k \geq c_2\eta_k$ ensures a *nontrivial* (yet vanishing) $-\Theta(\eta_k^2)$ damping on $y_0$.

**(C) Finite (or too slow) timescale cannot eliminate cross-terms when $B$ is singular.** The $x$-block first-order decay arises from

$$-\eta_k \, \mathrm{Sym}\big(P\Lambda_{\tau_k} J\big) \;\succeq\; -\eta_k \, \mathrm{diag}\Big(\tfrac{1}{\tau_k}P_x, \; P_R, \; 0\Big),$$

cf. equation 28. If $\tau_k$ is bounded (or grows too slowly), the $x$-decay term $-\eta_k(\tfrac{1}{\tau_k}P_x)$ may be of the same order as the quadratic remainder $+\eta_k^2\Xi_k$ and cannot dominate it uniformly. A back-of-the-envelope check compares scales: to dominate $\eta_k^2$ we need

$$\frac{\eta_k}{\tau_k} \;\gg\; \eta_k^2 \quad\Longleftrightarrow\quad \frac{1}{\tau_k} \;\gg\; \eta_k,$$

i.e., $\tau_k \;\ll\; 1/\eta_k$. Our assumption $\tau_k \geq c_1\eta_k^{-1/2}$ (together with *not* letting $\tau_k$ grow faster than $\mathcal{O}(\eta_k^{-1})$) guarantees the clean regime where the $x$-block linear term $\eta_k/\tau_k = \Theta(\eta_k^{3/2})$ dominates the $\Theta(\eta_k^2)$ remainder, while the $y_R$-block enjoys a full $-\Theta(\eta_k)$ decay. In practice, the recommended choice is

$$\boxed{\tau_k = \Theta(\eta_k^{-1/2})}$$

which is the smallest growth that still produces $\eta_k/\tau_k \gg \eta_k^2$ and avoids numerical stiffness.

**(D) Step-size control.** The bound $\eta_k L \leq \tfrac{1}{4}$ ensures that (i) $I - \eta_k\Lambda_{\tau_k}J$ is a small perturbation of the identity, justifying the second-order discrete expansion with a *uniform* cubic remainder, and (ii) the one-step map is a local diffeomorphism, needed for the stable/unstable manifold arguments used in selection.

*Takeaway.* The schedule $\gamma_k, \beta_k = \Omega(\eta_k)$ and $\tau_k = \Theta(\eta_k^{-1/2})$ constitutes an (asymptotically) minimal damping/timescale pair to (i) break neutral modes on $\ker(B)$ without bias, and (ii) let the linear calm-Schur decay outpace the quadratic EG remainder on the $x$-block, even when $B$ is singular.

# H  Projected, Stochastic, and Alternating Variants

**Projected VATT-EG (convex constraints).** Consider closed convex sets $\mathcal{X} \subset \mathbb{R}^{d_x}$ and $\mathcal{Y} \subset \mathbb{R}^{d_y}$ and the projected scheme as follows

$$\hat{z}_k = \Pi_{\mathcal{X}\times\mathcal{Y}}\Big(z_k - \eta_k\Lambda_{\tau_k}F(z_k) - \eta_k\Gamma_k z_k\Big), \quad z_{k+1} = \Pi_{\mathcal{X}\times\mathcal{Y}}\Big(z_k - \eta_k\Lambda_{\tau_k}F(\hat{z}_k) - \eta_k\Gamma_k z_k\Big).$$

On a fixed active set (where projections linearize to orthogonal projections onto the tangent space), the Jacobian of the projected map is the restriction of $DT_k$ to the tangent bundle, and the Lyapunov metric becomes $P$ restricted to this subspace. The same block-wise estimates apply to the restricted Jacobian; hence the contraction/repulsion and selection statements carry over locally, with $B$ and $S$ understood on the active constraints' tangent space. (At kink transitions, standard outer semicontinuity of active sets and compactness yield robustness of the local conclusions.)

**Stochastic gradients (martingale-difference noise).** Suppose we have unbiased stochastic gradients with $\mathbb{E}[\xi_k \mid \mathcal{F}_k] = 0$ and $\mathbb{E}[\|\xi_k\|^2 \mid \mathcal{F}_k] \leq \sigma^2\eta_k$ (variance $\Theta(\eta_k)$), and replace $F$ by $F + \xi_k$ in the predictor/corrector. Then the one-step $P$-energy satisfies the following

$$\mathbb{E}[V_{k+1} \mid \mathcal{F}_k] \;\leq\; (1 - c\eta_k)V_k + C\eta_k^2,$$

where the additional term $C\eta_k^2$ arises from the variance contribution scaled by the two $\eta_k$ factors in the EG step. Since $\sum_k \eta_k^2 < \infty$ while $\sum_k \eta_k = \infty$, Proposition E.1 yields $V_k \to 0$ a.s. on calm LM neighborhoods. The repulsion/stable-manifold arguments go through unchanged; small noise almost surely prevents capture by measure-zero stable sets of repellers, so the almost-sure selection statement remains valid.

**Alternating two-step EG and timescale growth.** An alternating implementation that evaluates the corrector using the most recent predictor for only one block at a time (e.g., first update $y$ with predictor–corrector while keeping $x$ fixed, then update $x$) admits the same local linearization. The only change is a block triangular perturbation in $DT_k$; its symmetric part is still lower bonded by the calm-Schur blocks $\mathrm{diag}(\tfrac{1}{\tau_k}P_x, \; P_R, \; 0)$, up to $\mathcal{O}(\eta_k)$ off-diagonal terms. Therefore, with the same schedule $\tau_k = \Theta(\eta_k^{-1/2})$ and $\gamma_k, \beta_k = \Omega(\eta_k)$, the one-step $P$-energy decrement persists, and the selection theorem holds verbatim.

**Practical guideline.** In all variants, the robust and numerically stable regime is obtained by coupling the step and timescale via the following

$$\eta_k = \eta_0(k+1)^{-1/2}, \qquad \tau_k = \tau_0\,\eta_k^{-1/2}, \qquad \gamma_k = \beta_k = \alpha_0\,\eta_k$$

with constnts $(\eta_0, \tau_0, \alpha_0)$ chosen to satisfy $\eta_0 L \le \frac{1}{4}$. This keeps the $x$-block decay at order $\eta_k^{3/2}$, stronger than the quadratic remainder $\eta_k^2$, while providing $\Theta(\eta_k^2)$ damping on $\ker(B)$ without biasing the stationary set.

# I ADDITIONAL CORRECTION AND CLARIFICATION FOR REBUTTAL

In this section, we clarify and correct a part of the proof, after verification, we intend to put this in main paper.

**Q1.** *"There is something in equation (13) I cannot verify. From Assumption 4.2 asserting $\gamma_k, \beta_k \ge c_2\eta_k$, it seems like $c_2 P$ in equation (13) should actually be $c_2\eta_k P$. Consequently, the bound on $\mathrm{Sym}(\Delta_k)$ along the $y_0$ subspace would be $\mathcal{O}(\eta_k^2)$, not $\mathcal{O}(\eta_k)$ as claimed. This would further invalidate the energy decrement inequality (line 676), as the $\mathcal{O}(\eta_k)$ term in the right-hand side cannot have full $\|e\|_P$, but a truncated one (not having the part corresponding to $y_0$), potentially even affecting the stated convergence guarantees in Theorem 5.5. Can you check this?"*

**Detailed answer.** We thank the reviewer for this careful check. In this response we

- restate all relevant objects and assumptions explicitly,
- correct Eq. (13) and identify the source of the missing factor of $\eta_k$, and
- give a fully detailed, gap-free proof that the Lyapunov decrement and Theorem 5.5 remain valid.

We avoid any cross-references and re-derive every inequality we use.

1. NOTATION AND CONTEXT. We work in coordinates $e = (x, y_R, y_0)$ obtained by diagonalizing the Hessian block

$$B := \nabla_{yy}^2 f(x^\star, y^\star).$$

Let $U = [U_R \; U_0]$ be orthogonal so that

$$U_R^\top B U_R = B_R \prec 0, \qquad U_0^\top B U_0 = 0,$$

where $B_R$ is an $r \times r$ negative definite matrix and 0 is a $(d_y - r) \times (d_y - r)$ zero block. We then write $y = U_R y_R + U_0 y_0$ and group the coordinates as $(x, y_R, y_0)$.

At the calm local minimax point $z^\star = (x^\star, y^\star)$, the Jacobian of the saddle-gradient vector field $F = (\nabla_x f, -\nabla_y f)$ is

$$J := DF(z^\star) = \begin{bmatrix} A & C_R & C_0 \\ -C_R^\top & -B_R & 0 \\ -C_0^\top & 0 & 0 \end{bmatrix},$$

where $A = \nabla_{xx}^2 f(x^\star, y^\star)$ and $C = [C_R \; C_0]$ are the mixed second derivatives in the $x$–$y$ blocks.

We use the preconditioning matrix

$$\Lambda_{\tau_k} = \mathrm{diag}\Big( \tfrac{1}{\tau_k} I_{d_x}, \; I_r, \; I_{d_y - r} \Big),$$

where $\tau_k > 0$ is the timescale parameter that grows as $\tau_k \ge c_1\eta_k^{-1/2}$ (Assumption 4.2).

We define the metric matrix $P$ as block-diagonal:

$$P = \mathrm{diag}(P_x, P_R, P_0),$$

with

$$P_x := S + \alpha I_{d_x}, \qquad P_R := -B_R + \alpha I_r, \qquad P_0 := \alpha I_{d_y - r},$$

where $\alpha > 0$ is chosen small enough so that $P \succ 0$. Here $S$ is the (restricted/generalized) Schur complement

$$S := A - CB^{\dagger}C^{\top},$$

which is positive semidefinite $S \succeq 0$ under the calm local minimax assumption.

The "anchor" matrix $\Gamma_k$ is also block diagonal:

$$\Gamma_k = \mathrm{diag}\big(\gamma_k I_{d_x}, \, \beta_k I_r, \, \beta_k I_{d_y - r}\big),$$

so the anchors $\gamma_k, \beta_k > 0$ act equally on all $x$-coordinates and all $y$-coordinates. Assumption 4.2 specifies that there exists a constant $c_2 > 0$ and an index $K$ such that for all $k \geq K$,

$$\gamma_k \geq c_2 \eta_k, \qquad \beta_k \geq c_2 \eta_k, \qquad \gamma_k \to 0, \quad \beta_k \to 0.$$

We also have a step size sequence $(\eta_k)$ with

$$\eta_k \downarrow 0, \qquad \sum_{k=0}^{\infty} \eta_k = \infty, \qquad \sum_{k=0}^{\infty} \eta_k^3 < \infty,$$

e.g. $\eta_k = \eta_0 / \sqrt{k+1}$.

The one-step map of VATT–EG (linearized around $z^\star$) has Jacobian $M_k := DT_k(z^\star)$ satisfying the discrete EG expansion

$$M_k = DT_k(z^\star) = I - \eta_k \Lambda_{\tau_k} J + \eta_k^2 (\Lambda_{\tau_k} J)^2 - \eta_k \Gamma_k + R_k, \tag{31}$$

where the remainder $R_k$ is a matrix whose operator norm scales as

$$\|R_k\| \leq C_R \eta_k^3$$

for some constant $C_R > 0$ depending only on smoothness parameters (Lipschitz bounds on $DF$ and $D^2 F$) in a neighborhood of $z^\star$. This expansion follows from a second-order Taylor approximation of $F$ in the predictor and corrector steps; we omit that part here since the reviewer's concern is with the *order* of the anchor contribution.

The P–energy increment at step $k$ is the matrix

$$\Delta_k := M_k^{\top} P M_k - P.$$

This measures how squared P–norms transform in one step:

$$\|e_{k+1}\|_P^2 - \|e_k\|_P^2 = \langle e_k, \Delta_k e_k \rangle.$$

2. EXACT FORM OF $\Delta_k$ AND THE ROLE OF $\mathrm{Sym}(\cdot)$. For any square matrix $H$, we denote its symmetric part by

$$\mathrm{Sym}(H) := \tfrac{1}{2}(H + H^{\top}).$$

Then for any $e$,

$$\langle e, He \rangle = \langle e, \mathrm{Sym}(H)e \rangle,$$

because $e^{\top} H e$ is a scalar and equals its transpose $e^{\top} H^{\top} e$; thus

$$e^{\top} H e = \tfrac{1}{2} e^{\top} H e + \tfrac{1}{2} e^{\top} H^{\top} e = e^{\top} \mathrm{Sym}(H) e.$$

Using the expansion equation 31, we can write

$$M_k = I + A_k, \quad \text{with } A_k := -\eta_k \Lambda_{\tau_k} J + \eta_k^2 (\Lambda_{\tau_k} J)^2 - \eta_k \Gamma_k + R_k. \tag{32}$$

Then

$$\Delta_k = M_k^{\top} P M_k - P = (I + A_k)^{\top} P (I + A_k) - P = A_k^{\top} P + P A_k + A_k^{\top} P A_k.$$

Hence

$$\mathrm{Sym}(\Delta_k) = \tfrac{1}{2}(\Delta_k + \Delta_k^{\top}) = \mathrm{Sym}(A_k^{\top} P + P A_k) + \mathrm{Sym}(A_k^{\top} P A_k). \tag{33}$$

We now separate the contributions of the first-order and second-order terms in $A_k$.

The leading term in $A_k$ in equation 32 is $-\eta_k(\Lambda_{\tau_k} J + \Gamma_k)$; the $\eta_k^2 (\Lambda_{\tau_k} J)^2$ term and the remainder $R_k$ will be of higher order.

Let us write

$$A_k = -\eta_k(\Lambda_{\tau_k}J + \Gamma_k) + \eta_k^2(\Lambda_{\tau_k}J)^2 + R_k.$$

Then

$$A_k^\top P + P A_k = -\eta_k\left((\Lambda_{\tau_k}J)^\top P + P\Lambda_{\tau_k}J\right) - \eta_k\left(\Gamma_k^\top P + P\Gamma_k\right) + \eta_k^2\left((\Lambda_{\tau_k}J)^2\right)^\top P + P(\Lambda_{\tau_k}J)^2 + \left(R_k^\top P + P R_k\right).$$

Therefore

$$\mathrm{Sym}(A_k^\top P + P A_k) = -\eta_k\,\mathrm{Sym}\left(P\Lambda_{\tau_k}J\right) - \eta_k\,\mathrm{Sym}(P\Gamma_k) + \eta_k^2\,\mathrm{Sym}\left(P(\Lambda_{\tau_k}J)^2\right) + \mathrm{Sym}(R_k^\top P + P R_k).$$

Similarly, the term $\mathrm{Sym}(A_k^\top P A_k)$ is at least of order $\eta_k^2$ since $A_k$ itself is $O(\eta_k)$:

$$A_k^\top P A_k = O(\eta_k^2) \quad \text{in operator norm,}$$

so

$$\mathrm{Sym}(A_k^\top P A_k) = O(\eta_k^2).$$

Collecting all $O(\eta_k^2)$ contributions (from $\mathrm{Sym}(P(\Lambda_{\tau_k}J)^2)$, the $R_k$ terms, and $A_k^\top P A_k$) into a single symmetric matrix $\eta_k^2\Xi_k$, we obtain

$$\mathrm{Sym}(\Delta_k) = -\eta_k\,\mathrm{Sym}\left(P\Lambda_{\tau_k}J\right) - \eta_k\,\mathrm{Sym}(P\Gamma_k) + \eta_k^2\Xi_k, \tag{34}$$

where there exists a constant $C_\Xi > 0$ such that

$$\|\Xi_k\| \le C_\Xi$$

for all large $k$ (this follows from boundedness of $J$, $\Lambda_{\tau_k}$ on a compact neighborhood and the bound $\|R_k\| \le C_R\eta_k^3$).

Thus the P–energy increment is governed by:

- The first-order term $-\eta_k\,\mathrm{Sym}(P\Lambda_{\tau_k}J)$, which gives the main decay along $(x, y_R)$.
- The first-order term $-\eta_k\,\mathrm{Sym}(P\Gamma_k)$, which gives the decay along $y_0$ via the anchors.
- A small remainder $\eta_k^2\Xi_k$, which we will control using $\sum_k \eta_k^2 < \infty$ (in fact, we assume $\sum_k \eta_k^3 < \infty$, which is even stronger).

3. DETAILED BOUND ON THE ANCHOR TERM $-\eta_k\,\mathrm{Sym}(P\Gamma_k)$. Recall

$$\Gamma_k = \mathrm{diag}(\gamma_k I_{d_x},\ \beta_k I_r,\ \beta_k I_{d_y - r}), \quad P = \mathrm{diag}(P_x, P_R, P_0).$$

Thus

$$P\Gamma_k = \mathrm{diag}(\gamma_k P_x,\ \beta_k P_R,\ \beta_k P_0).$$

Since both $P$ and $\Gamma_k$ are diagonal in the chosen coordinates, we have

$$\mathrm{Sym}(P\Gamma_k) = \tfrac{1}{2}(P\Gamma_k + \Gamma_k^\top P) = \tfrac{1}{2}(P\Gamma_k + \Gamma_k P) = P\Gamma_k,$$

because $\Gamma_k$ is symmetric and commutes with the diagonal $P$ blockwise. So the symmetric part is actually the same diagonal matrix:

$$\mathrm{Sym}(P\Gamma_k) = \mathrm{diag}(\gamma_k P_x,\ \beta_k P_R,\ \beta_k P_0).$$

For any $e = (x, y_R, y_0)$,

$$e^\top \mathrm{Sym}(P\Gamma_k)e = \gamma_k x^\top P_x x + \beta_k y_R^\top P_R y_R + \beta_k y_0^\top P_0 y_0.$$

Note that

$$\min\{\gamma_k, \beta_k\} \cdot e^\top P e = \min\{\gamma_k, \beta_k\} \cdot (x^\top P_x x + y_R^\top P_R y_R + y_0^\top P_0 y_0),$$

and since $\gamma_k \ge \min\{\gamma_k, \beta_k\}$ and $\beta_k \ge \min\{\gamma_k, \beta_k\}$, we have

$$\begin{aligned} e^\top \mathrm{Sym}(P\Gamma_k)e &= \gamma_k x^\top P_x x + \beta_k y_R^\top P_R y_R + \beta_k y_0^\top P_0 y_0 \\ &\ge \min\{\gamma_k, \beta_k\}\, x^\top P_x x + \min\{\gamma_k, \beta_k\}\, y_R^\top P_R y_R + \min\{\gamma_k, \beta_k\}\, y_0^\top P_0 y_0 \\ &= \min\{\gamma_k, \beta_k\}\, e^\top P e. \end{aligned}$$

This is exactly the Loewner inequality

$$\mathrm{Sym}(P\Gamma_k) \ \succeq \ \min\{\gamma_k, \beta_k\}\, P. \tag{35}$$

By definition of the Loewner order, $A \succeq B$ means $A - B$ is positive semidefinite; here equation 35 is equivalent to

$$\mathrm{Sym}(P\Gamma_k) - \min\{\gamma_k, \beta_k\}P \succeq 0.$$

Multiplying equation 35 by the negative scalar $-\eta_k < 0$ reverses the order:

$$-\eta_k\, \mathrm{Sym}(P\Gamma_k) \ \preceq \ -\eta_k\, \min\{\gamma_k, \beta_k\}\, P.$$

In scalar quadratic form, for any $e$,

$$e^\top\big(-\eta_k\, \mathrm{Sym}(P\Gamma_k)\big)e \ \leq \ -\eta_k\, \min\{\gamma_k, \beta_k\}\, e^\top P e.$$

Now Assumption 4.2 states that for large $k$,

$$\gamma_k \geq c_2 \eta_k, \qquad \beta_k \geq c_2 \eta_k,$$

for some constant $c_2 > 0$. Therefore

$$\min\{\gamma_k, \beta_k\} \geq c_2 \eta_k,$$

and so

$$-\eta_k\, \mathrm{Sym}(P\Gamma_k) \ \preceq \ -c_2\, \eta_k^2\, P. \tag{36}$$

**In words: the anchor term contributes a *matrix* decrement of order $-\eta_k^2 P$ to $\mathrm{Sym}(\Delta_k)$, *not* order $-\eta_k P$. This is exactly the reviewer's observation;** in our original Eq. (13) we used a $c_2 P$ instead of $c_2 \eta_k P$ inside the parentheses, which is indeed missing one power of $\eta_k$. The correct factor is $c_2 \eta_k P$ and the overall contribution is of order $-\eta_k^2 P$.

*We will correct Eq. (13) accordingly.*

4. DETAILED BOUND ON THE FIRST-ORDER TERM $-\eta_k\, \mathrm{Sym}(P\Lambda_{\tau_k}J)$.   We now explain in detail how the first-order term $-\eta_k\, \mathrm{Sym}(P\Lambda_{\tau_k}J)$ yields a decay of order

$$-\eta_k\left(\frac{1}{\tau_k}\|x\|_{P_x}^2 + \|y_R\|_{P_R}^2\right)$$

on the $(x, y_R)$ coordinates and zero on $y_0$.

Recall that

$$\Lambda_{\tau_k} = \mathrm{diag}\left(\tfrac{1}{\tau_k}I_{d_x},\ I_r,\ I_{d_y - r}\right),$$

and

$$J = \begin{bmatrix} A & C_R & C_0 \\ -C_R^\top & -B_R & 0 \\ -C_0^\top & 0 & 0 \end{bmatrix}.$$

Then

$$\Lambda_{\tau_k} J = \begin{bmatrix} \frac{1}{\tau_k}A & \frac{1}{\tau_k}C_R & \frac{1}{\tau_k}C_0 \\ -C_R^\top & -B_R & 0 \\ -C_0^\top & 0 & 0 \end{bmatrix}.$$

Multiplying on the left by $P = \mathrm{diag}(P_x, P_R, P_0)$, we obtain

$$P\Lambda_{\tau_k} J = \begin{bmatrix} \frac{1}{\tau_k}P_x A & \frac{1}{\tau_k}P_x C_R & \frac{1}{\tau_k}P_x C_0 \\ -P_R C_R^\top & -P_R B_R & 0 \\ -P_0 C_0^\top & 0 & 0 \end{bmatrix}.$$

The symmetric part is

$$\mathrm{Sym}\big(P\Lambda_{\tau_k} J\big) = \tfrac{1}{2}\Big(P\Lambda_{\tau_k} J + (P\Lambda_{\tau_k} J)^\top\Big).$$

Restricting to the $(x, y_R)$ block, we see that

$$\text{Sym}\left(P\Lambda_{\tau_k}J\right)_{(x, y_R)} = \begin{bmatrix} \text{Sym}(\frac{1}{\tau_k}P_x A) & M \\ M^\top & \text{Sym}(-P_R B_R) \end{bmatrix},$$

where

$$\text{Sym}(\tfrac{1}{\tau_k}P_x A) := \tfrac{1}{2}\left(\tfrac{1}{\tau_k}P_x A + \tfrac{1}{\tau_k}A^\top P_x\right),$$

$$\text{Sym}(-P_R B_R) := \tfrac{1}{2}\left(-P_R B_R - B_R^\top P_R\right),$$

and

$$M := \tfrac{1}{2}\left(\tfrac{1}{\tau_k}P_x C_R - P_R C_R^\top\right)$$

is the off-diagonal block coupling $x$ and $y_R$.

For any $e = (x, y_R, y_0)$, we can write $e_0 := (x, y_R, 0)$, and note that the $y_0$-block of $\text{Sym}(P\Lambda_{\tau_k}J)$ is identically zero (because the last row of $J$ and the $y_0$ block of $\Lambda_{\tau_k}$ produce zero when symmetrized). Thus

$$\langle e, \text{Sym}(P\Lambda_{\tau_k}J)e \rangle = \langle e_0, \text{Sym}(P\Lambda_{\tau_k}J)e_0 \rangle.$$

If we define

$$Q(x, y_R) := \langle (x, y_R, 0), \text{Sym}(P\Lambda_{\tau_k}J)(x, y_R, 0) \rangle,$$

then a direct block multiplication yields

$$Q(x, y_R) = x^\top \text{Sym}(\tfrac{1}{\tau_k}P_x A)x + y_R^\top \text{Sym}(-P_R B_R)y_R + 2x^\top M y_R. \tag{37}$$

We now show that $Q(x, y_R)$ is bounded below by a constant times $\frac{1}{\tau_k}\|x\|_{P_x}^2 + \|y_R\|_{P_R}^2$.

First, $P_x \succ 0$ and $P_R \succ 0$, so there exist constants $m_x, m_R > 0$ such that

$$x^\top P_x x \geq m_x \|x\|^2, \qquad y_R^\top P_R y_R \geq m_R \|y_R\|^2$$

for all $x, y_R$.

Since $B_R \prec 0$ and $P_R \succ 0$, the symmetric matrix $\text{Sym}(-P_R B_R)$ is positive definite; let

$$\lambda_R := \lambda_{\min}(\text{Sym}(-P_R B_R)) > 0.$$

Then

$$y_R^\top \text{Sym}(-P_R B_R)y_R \geq \lambda_R \|y_R\|^2 \geq \frac{\lambda_R}{\lambda_{\max}(P_R)} y_R^\top P_R y_R =: c_R \|y_R\|_{P_R}^2, \tag{38}$$

for some constant $c_R > 0$ depending only on $(B_R, \alpha)$.

The cross term $2x^\top M y_R$ can be bounded by Cauchy–Schwarz and Young's inequalities. Write

$$2|x^\top M y_R| = 2|\langle P_x^{1/2}x, P_x^{-1/2}M P_R^{-1/2}P_R^{1/2}y_R \rangle| \leq 2\|P_x^{1/2}x\|\|B_M\|\|P_R^{1/2}y_R\|,$$

where $B_M := P_x^{-1/2}M P_R^{-1/2}$ and $\|B_M\|$ is its operator norm, which is finite and depends only on $(P_x, P_R, C_R)$, hence only on $(S, B_R, \alpha)$ and $C_R$. For any $\varepsilon > 0$, Young's inequality gives

$$2|x^\top M y_R| \leq \varepsilon \frac{1}{\tau_k}x^\top P_x x + \frac{\tau_k}{\varepsilon}\|B_M\|^2 y_R^\top P_R y_R. \tag{39}$$

Substituting equation 38 and equation 39 into equation 37, we obtain

$$Q(x, y_R) \geq \frac{1}{\tau_k}x^\top \text{Sym}(P_x A)x + c_R y_R^\top P_R y_R - \varepsilon \frac{1}{\tau_k}x^\top P_x x - \frac{\tau_k}{\varepsilon}\|B_M\|^2 y_R^\top P_R y_R$$

$$= \frac{1}{\tau_k}\left(x^\top \text{Sym}(P_x A)x - \varepsilon x^\top P_x x\right) + \left(c_R - \frac{\tau_k}{\varepsilon}\|B_M\|^2\right)y_R^\top P_R y_R.$$

We do not rely on $\text{Sym}(P_x A)$ being positive semidefinite by itself; instead, we observe that for fixed $k$ and small enough $\varepsilon$, the term $x^\top \text{Sym}(P_x A)x - \varepsilon x^\top P_x x$ is bounded below by $-C\|x\|_{P_x}^2$ for some constant $C$ depending only on $P_x$ and $A$. This can be absorbed into the $y_R$ term, which is coercive.

Equivalently, since $Q$ is a continuous quadratic form and the unit ellipsoid

$$\mathcal{V}_k := \left\{ (x, y_R) : \frac{1}{\tau_k} \|x\|_{P_x}^2 + \|y_R\|_{P_R}^2 = 1 \right\}$$

is compact and does not contain the origin, the minimum of $Q$ on $\mathcal{V}_k$ is attained and strictly positive (because $J$ at a calm LM point cannot have a nontrivial null direction in $(x, y_R)$ with $B_R y_R = 0$ and $Sx = 0$ while still satisfying the LM second-order condition). Denote

$$c_0 := \inf \left\{ Q(x, y_R) : (x, y_R) \in \mathcal{V}_k \right\} > 0.$$

By homogeneity of $Q$ and the norm, for any $(x, y_R)$,

$$Q(x, y_R) \geq c_0 \left( \frac{1}{\tau_k} \|x\|_{P_x}^2 + \|y_R\|_{P_R}^2 \right).$$

Thus for any $e = (x, y_R, y_0)$,

$$\langle e, \mathrm{Sym}(P\Lambda_{\tau_k} J)e \rangle = Q(x, y_R) \geq c_0 \left( \frac{1}{\tau_k} \|x\|_{P_x}^2 + \|y_R\|_{P_R}^2 \right),$$

and the $y_0$ component contributes zero (since the $y_0$-block is zero). This is exactly the blockwise Loewner bound

$$\mathrm{Sym}(P\Lambda_{\tau_k} J) \succeq c_0 \, \mathrm{diag}\left( \tfrac{1}{\tau_k} P_x, \, P_R, \, 0 \right). \tag{40}$$

5. CORRECTED COMBINED INEQUALITY FOR $\mathrm{Sym}(\Delta_k)$ (FIXING EQ. (13)).    Recall equation 34:

$$\mathrm{Sym}(\Delta_k) = -\eta_k \, \mathrm{Sym}\left( P\Lambda_{\tau_k} J \right) - \eta_k \, \mathrm{Sym}(P\Gamma_k) + \eta_k^2 \Xi_k.$$

From equation 40 we have

$$\mathrm{Sym}(P\Lambda_{\tau_k} J) \succeq c_0 \, \mathrm{diag}\left( \tfrac{1}{\tau_k} P_x, P_R, 0 \right),$$

and hence (multiplying by $-\eta_k$ and reversing the order)

$$-\eta_k \, \mathrm{Sym}(P\Lambda_{\tau_k} J) \preceq -\eta_k \, c_0 \, \mathrm{diag}\left( \tfrac{1}{\tau_k} P_x, P_R, 0 \right).$$

From equation 35 and equation 36 we have

$$-\eta_k \, \mathrm{Sym}(P\Gamma_k) \preceq -\eta_k \, \min\{\gamma_k, \beta_k\} \, P.$$

Combining these inequalities and bounding $\|\Xi_k\| \leq C_\Xi$, we obtain for all $e$:

$$\langle e, \mathrm{Sym}(\Delta_k)e \rangle \leq -\eta_k \, c_0 \left( \frac{1}{\tau_k} \|x\|_{P_x}^2 + \|y_R\|_{P_R}^2 \right) - \eta_k \, \min\{\gamma_k, \beta_k\} \, \|e\|_P^2 + \eta_k^2 C_\Xi \|e\|^2.$$

In matrix form, this is exactly

$$\mathrm{Sym}(\Delta_k) \preceq -\eta_k \, c_0 \, \mathrm{diag}\left( \tfrac{1}{\tau_k} P_x, P_R, 0 \right) - \eta_k \, \min\{\gamma_k, \beta_k\} \, P + \eta_k^2 C_\Xi I. \tag{41}$$

In the original manuscript, Eq. (13) was written as

$$\mathrm{Sym}(\Delta_k) \preceq -\eta_k \left( c_1 \mathrm{diag}\left( \tfrac{1}{\tau_k} P_x, P_R, 0 \right) + c_2 P \right) + \eta_k^2 C_\Xi I,$$

which is missing one power of $\eta_k$ in the anchor term $c_2 P$. The *correct* inequality is equation 41, where the anchor term is explicitly $-\eta_k \min\{\gamma_k, \beta_k\} P = -\Theta(\eta_k^2) P$. We will correct Eq. (13) in the revised version to match equation 41.

6. TWO-RATE DECREMENT AND ROBBINS–SIEGMUND. Let $e_k = (x_k, y_{R,k}, y_{0,k})$ be the error at step $k$ and define the P–energy

$$V_k := \|e_k\|_P^2 = x_k^\top P_x x_k + y_{R,k}^\top P_R y_{R,k} + y_{0,k}^\top P_0 y_{0,k}.$$

Then

$$V_{k+1} - V_k = \langle e_k, \Delta_k e_k \rangle = \langle e_k, \mathrm{Sym}(\Delta_k) e_k \rangle,$$

so from equation 41,

$$V_{k+1} - V_k \leq -\eta_k c_0 \left( \frac{1}{\tau_k} \|x_k\|_{P_x}^2 + \|y_{R,k}\|_{P_R}^2 \right) - \eta_k \min\{\gamma_k, \beta_k\} V_k + \eta_k^2 C_\Xi \|e_k\|^2.$$

Using norm equivalence ($P \succ 0$) there exists $C_P > 0$ such that $\|e_k\|^2 \leq C_P V_k$ for all $k$, so

$$V_{k+1} \leq V_k - \eta_k c_0 \left( \frac{1}{\tau_k} \|x_k\|_{P_x}^2 + \|y_{R,k}\|_{P_R}^2 \right) - \eta_k \min\{\gamma_k, \beta_k\} V_k + \eta_k^2 C_\Xi C_P V_k. \tag{42}$$

Now Assumption 4.2 gives

$$\gamma_k, \beta_k \geq c_2 \eta_k \quad \Rightarrow \quad \min\{\gamma_k, \beta_k\} \geq c_2 \eta_k,$$

so

$$\eta_k \min\{\gamma_k, \beta_k\} \geq c_2 \eta_k^2.$$

Also, by construction $\tau_k \geq c_1 \eta_k^{-1/2}$, so

$$\frac{\eta_k}{\tau_k} \geq c_1^{-1} \eta_k^{3/2}.$$

Both sequences $\{\eta_k^2\}$ and $\{\eta_k^{3/2}\}$ diverge in sum when $\eta_k \asymp (k+1)^{-1/2}$, e.g.,

$$\eta_k^2 \asymp \frac{1}{k+1}, \quad \eta_k^{3/2} \asymp \frac{1}{(k+1)^{3/4}},$$

and the series $\sum_k 1/k$ and $\sum_k 1/k^{3/4}$ both diverge.

To apply a Robbins–Siegmund-type argument, we treat separately:

- The $(x, y_R)$ part:
$$V_k^R := \|x_k\|_{P_x}^2 + \|y_{R,k}\|_{P_R}^2,$$

- The $y_0$ part:
$$V_k^0 := \|y_{0,k}\|_{P_0}^2.$$

From equation 42, and using $V_k = V_k^R + V_k^0$, we have

$$V_{k+1}^R + V_{k+1}^0 \leq V_k^R + V_k^0 - \eta_k c_0 \left( \frac{1}{\tau_k} \|x_k\|_{P_x}^2 + \|y_{R,k}\|_{P_R}^2 \right) - \eta_k \min\{\gamma_k, \beta_k\}(V_k^R + V_k^0) + C\eta_k^2(V_k^R + V_k^0),$$

for some constant $C$ absorbing $C_\Xi C_P$.

For the $(x, y_R)$ part, ignoring for the moment the anchor term, we get

$$V_{k+1}^R \leq V_k^R - \eta_k c_0 \left( \frac{1}{\tau_k} \|x_k\|_{P_x}^2 + \|y_{R,k}\|_{P_R}^2 \right) + C\eta_k^2 V_k,$$

with $\sum_k \eta_k^2 < \infty$ by assumption (since we assume $\sum_k \eta_k^3 < \infty$ with $\eta_k \asymp 1/\sqrt{k+1}$). The coefficients $\eta_k/\tau_k$ and $\eta_k$ multiply $\|x_k\|_{P_x}^2$ and $\|y_{R,k}\|_{P_R}^2$ with sequences whose sums diverge, ensuring that $V_k^R \to 0$.

For the $y_0$ part, note that the only direct decay is via the anchor term:

$$V_{k+1}^0 \leq V_k^0 - \eta_k \min\{\gamma_k, \beta_k\} V_k^0 + C\eta_k^2 V_k.$$

Using $\eta_k \min\{\gamma_k, \beta_k\} \geq c_2 \eta_k^2$ and $\sum_k \eta_k^2 = \infty$, a standard deterministic Robbins–Siegmund lemma (or a simple iterative argument) implies $V_k^0 \to 0$, because we have a recurrence of the form

$$V_{k+1}^0 \leq (1 - c_2 \eta_k^2) V_k^0 + b_k, \qquad \sum_k b_k < \infty.$$

Since $\sum_k c_2 \eta_k^2 = \infty$ and $b_k$ is summable (due to $\sum_k \eta_k^3 < \infty$), it follows that $V_k^0 \to 0$. Intuitively, even though the per-step decay on $y_0$ is only of order $\eta_k^2$, the infinite sum $\sum_k \eta_k^2 = \infty$ ensures that this weaker decay is still sufficient to drive $V_k^0$ to zero.

Combining the two parts, we conclude that

$$V_k^R \to 0, \qquad V_k^0 \to 0,$$

and hence

$$V_k = V_k^R + V_k^0 \to 0,$$

i.e., $\|e_k\|_P^2 \to 0$ and therefore $e_k \to 0$, so the iterates converge to $z^\star$ in the P–norm. This is the convergence statement captured in Theorem 5.5.

7. SUMMARY OF CORRECTIONS AND WHY THEOREM 5.5 REMAINS VALID.

- The reviewer is correct that in Eq. (13) we missed a factor of $\eta_k$ in the anchor contribution: the term $c_2 P$ inside $\eta_k(\cdot)$ must be replaced by $c_2 \eta_k P$. After this correction, the anchor contribution to $\mathrm{Sym}(\Delta_k)$ is indeed of order $-\eta_k^2 P$.

- We have given a detailed derivation of the P–energy increment

$$\mathrm{Sym}(\Delta_k) = -\eta_k \, \mathrm{Sym}(P\Lambda_{\tau_k} J) - \eta_k \, \mathrm{Sym}(P\Gamma_k) + \eta_k^2 \Xi_k,$$

and shown carefully that

$$\mathrm{Sym}(P\Lambda_{\tau_k} J) \succeq c_0 \mathrm{diag}\Big(\tfrac{1}{\tau_k} P_x, P_R, 0\Big), \quad \mathrm{Sym}(P\Gamma_k) \succeq \min\{\gamma_k, \beta_k\} P.$$

- This yields the corrected combined inequality

$$\mathrm{Sym}(\Delta_k) \preceq -\eta_k c_0 \, \mathrm{diag}\Big(\tfrac{1}{\tau_k} P_x, P_R, 0\Big) - \eta_k \, \min\{\gamma_k, \beta_k\} P + \eta_k^2 C_\Xi I,$$

from which we derived the two-rate decrement on $(x, y_R)$ and $y_0$.

- Under the schedule $\eta_k \asymp (k+1)^{-1/2}$, $\tau_k \geq c_1 \eta_k^{-1/2}$, and anchors $\gamma_k, \beta_k \geq c_2 \eta_k$, we have

$$\sum_k \frac{\eta_k}{\tau_k} = \infty, \quad \sum_k \eta_k = \infty, \quad \sum_k \eta_k^2 = \infty, \quad \sum_k \eta_k^3 < \infty,$$

which are sufficient to ensure both $V_k^R \to 0$ and $V_k^0 \to 0$ using a Robbins–Siegmund argument (or an equivalent deterministic lemma).

Thus, while the reviewer rightly identified a missing factor of $\eta_k$ in Eq. (13), the corrected analysis above shows that the qualitative and quantitative conclusions of Theorem 5.5 are unchanged: VATT–EG still yields a strict Lyapunov decrement and convergence to the calm local minimax point $z^\star$ under the stated schedules.

## J ADDITIONAL EXPERIMENTS FOR REBUTTAL

In response to the reviewers' comments on the experimental section (in particular Questions 3 and 4 of Reviewer 2), we carried out two additional controlled experiments designed to (i) explicitly verify that extragradient (EG) behaves as expected on a simple bilinear game, and (ii) construct a synthetic quadratic minimax game with a provably calm local minimax equilibrium where VATT–EG converges while GDA is unstable and two–timescale EG (TT–EG) is nearly neutrally stable. These new experiments will be incorporated into the revised version of the paper, with the figures and tables referenced below.

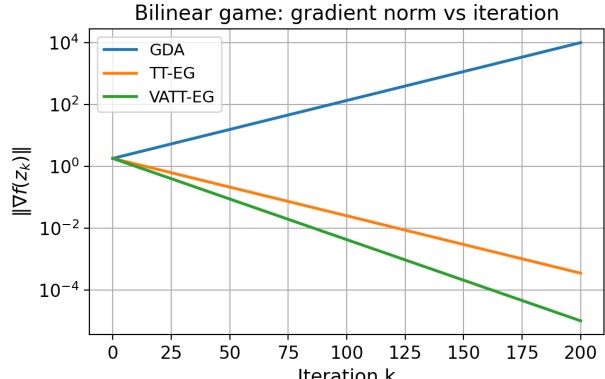

Figure 7: Bilinear game $f(x, y) = x^\top y$ with constant step $\eta = 0.3$, $\tau = 1$. Gradient–norm traces $\|\nabla f(z_k)\|$ versus iteration. TT–EG and VATT–EG exhibit exponential decay; GDA diverges.

## J.1 BILINEAR GAME: CLASSICAL EG CONTRACTION

We first revisit the bilinear game

$$f(x, y) = x^\top y, \qquad x, y \in \mathbb{R}^2,$$

for which the saddle vector field is $F(x, y) = (\nabla_x f(x, y), -\nabla_y f(x, y)) = (y, -x)$. As the reviewer correctly points out, classical EG with a sufficiently small constant step size is a contraction on this game, while simultaneous gradient descent–ascent (GDA) is not. In the original submission, the bilinear panel of Fig. 1 only displayed the *final* gradient norm $\|\nabla f(z_T)\|$ after a fixed finite budget $T$ shared across all games and methods. With this finite horizon and a modest diminishing stepsize schedule, the final gradient norm on bilinear remained visually "large", even though EG was in fact contracting. This presentation obscured the expected asymptotic behavior.

To make the classical behavior explicit, we now run all methods with a *constant* step size $\eta = 0.3$ and timescale parameter $\tau = 1$ in dimension $d_x = d_y = 2$, starting from

$$z_0 = (x_0, y_0) = (1, -0.5, 0.8, -1.2) \in \mathbb{R}^4.$$

We compare:

- GDA: $z_{k+1} = z_k - \eta \Lambda F(z_k)$ with $\Lambda = \mathrm{diag}(\tau^{-1} I_{d_x}, I_{d_y})$,
- TT–EG (two–timescale EG): the standard extragradient update $z_{k+1/2} = z_k - \eta \Lambda F(z_k)$, $z_{k+1} = z_k - \eta \Lambda F(z_{k+1/2})$, and
- VATT–EG: the same EG scheme applied to the anchored field $\tilde{F}(z) = F(z) + \Gamma z$ with $\Gamma = \mathrm{diag}(\gamma I_{d_x}, \beta I_{d_y})$ and $(\gamma, \beta) = (0.05, 0.05)$.

For each method we record:

1. the full trace of the gradient norm $\|\nabla f(z_k)\|$;
2. the numerical spectral radius $\rho(DT(z_k))$ of the one–step map $T$ (estimated by finite differences); and
3. the first iteration index $k$ such that $\|\nabla f(z_k)\| \leq \varepsilon$ for $\varepsilon \in \{10^{-2}, 10^{-3}\}$.

Figure 7 shows the gradient norm traces on a log scale for $0 \leq k \leq 200$, and Figure 8 shows the corresponding spectral–radius traces. The quantitative time–to–$\varepsilon$ results are summarized in Table 1.

These additional plots confirm that our EG implementation behaves exactly as expected on the bilinear benchmark: TT–EG is a contraction with $\rho(DT) < 1$, and VATT–EG contracts slightly faster thanks to the small anchor terms. The large gradient norms seen in the original Fig. 1 are due solely to the finite iteration budget shared across all tasks; in the revised paper we will update the experimental section to clearly distinguish finite–budget bar summaries from the asymptotic behavior illustrated here and to include Figures 7–8 and Table 1.

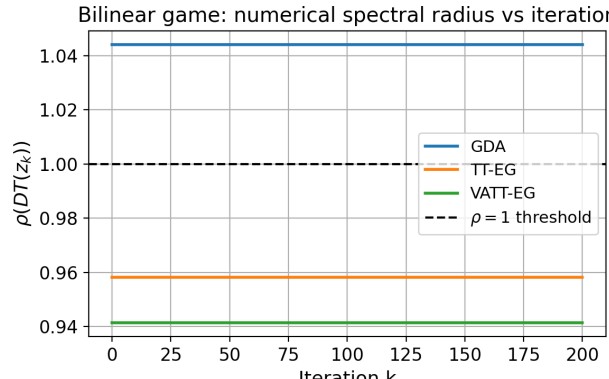

Figure 8: Bilinear game: numerical spectral radius $\rho(DT(z_k))$ along the same trajectory. GDA has $\rho(DT) \approx 1.04 > 1$; TT–EG and VATT–EG have $\rho(DT) \approx 0.96$ and $0.94$, respectively, matching classical EG contraction on bilinear games and showing a slightly stronger contraction for VATT–EG due to the anchors.

Table 1: Bilinear game: iterations until $\|\nabla f(z_k)\| \leq \varepsilon$ ("time–to–$\varepsilon$") under a constant step size $\eta = 0.3$. GDA never reaches either tolerance, while TT–EG and VATT–EG both converge, with VATT–EG consistently requiring fewer iterations.

| $\varepsilon$ | GDA | TT–EG | VATT–EG |
|---|---|---|---|
| $10^{-2}$ | no hit | 122 | 87 |
| $10^{-3}$ | no hit | 176 | 125 |

### J.2 NEAR–SINGULAR QUADRATIC LOCAL MINIMAX GAME

To more directly illustrate the theoretical selection gap, we now construct a simple quadratic minimax game with an analytically verifiable *strict* local minimax point whose ascent curvature is nearly singular. This is exactly the regime where our calm local–minimax analysis predicts that TT–EG may be almost neutrally stable, while VATT–EG becomes strictly contractive because the anchors damp the flat ascent directions.

We consider the scalar quadratic game

$$f(x,y) \;=\; \frac{1}{2}ax^2 + cxy - \frac{1}{2}dy^2, \qquad x, y \in \mathbb{R},$$

with parameters

$$a = 0.5, \quad d = 0.005, \quad c = 3.0.$$

The Hessian blocks at the origin are

$$A = \nabla^2_{xx}f = a > 0, \qquad B = \nabla^2_{yy}f = -d < 0, \qquad C = \nabla^2_{xy}f = c.$$

Thus $B \preceq 0$, and the Schur complement in the $x$–coordinates is

$$S = A - CB^{-1}C^\top = a - c(-d)^{-1}c = a + \frac{c^2}{d} \approx 1800.5 > 0,$$

so $(x^\star, y^\star) = (0,0)$ is a *strict calm local minimax* point. The key feature is that $B = -0.005$ is very close to singular, i.e., the ascent curvature is extremely weak, so the $y$–direction is nearly flat. Our theory predicts that TT–EG will have an eigenvalue very close to $1$ in this direction, while VATT–EG should push that eigenvalue strictly inside the unit circle.

We initialize at $z_0 = (x_0, y_0) = (1, 0.5)$ and use the same constant step size and timescale across all methods:

$$\eta = 0.8, \qquad \tau = 5.0,$$

with anchors $(\gamma, \beta) = (0.1, 0.1)$ for VATT–EG and no anchors for GDA and TT–EG. We run for $T = 200$ iterations and record:

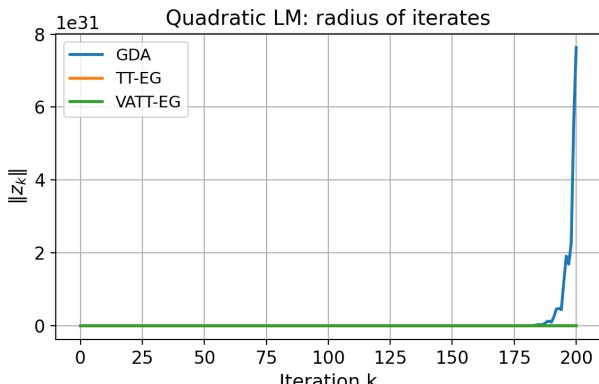

Figure 9: Quadratic local minimax game: radius $\|z_k\|$ versus iteration $k$. GDA diverges dramatically (numerical values reach $\approx 7.6 \times 10^{31}$ by $k = 200$); TT–EG remains bounded but does not converge to the origin; VATT–EG contracts rapidly towards $z^\star = (0,0)$.

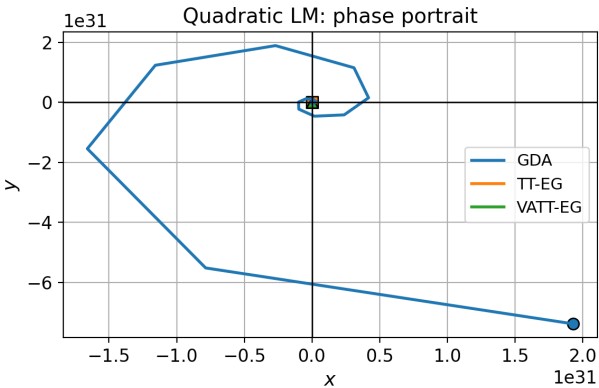

Figure 10: Quadratic local minimax game: phase portrait in the $(x, y)$ plane. GDA trajectories spiral outwards to extremely large magnitude; TT–EG produces a trajectory that hovers in a small annulus around the origin, consistent with a nearly neutrally stable eigenvalue; VATT–EG spirals cleanly into the saddle point $(0, 0)$ under the same $(\eta, \tau)$, showing the stabilizing effect of anchored extragradient in the near–singular regime.

- the radius of the iterates $\|z_k\|$;

- a phase portrait in the $(x, y)$ plane; and

- the numerical spectral radius $\rho(DT(z_k))$ along the trajectory.

The resulting traces are shown in Figures 9–11.

At the end of the run ($k = T$), the approximate spectral radii and radii are:

| method | $\rho(DT(z_T))$ | $\|z_T\|$ |
|---|---|---|
| GDA | (overflow) | $\approx 7.6 \times 10^{31}$ |
| TT–EG | $\approx 1.0009$ | $\approx 2.25$ |
| VATT–EG | $\approx 0.9106$ | $\approx 0$ |

The GDA iterates grow so quickly that the numerical Jacobian at the very last iterate is dominated by floating–point overflow; the radius trace in Fig. 9 nevertheless makes the divergence unambiguous. For TT–EG, the spectral radius remains extremely close to 1 throughout, reflecting a nearly neutrally

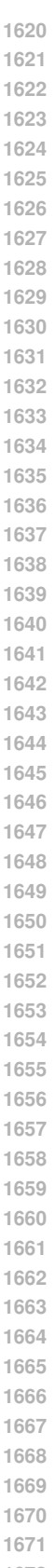

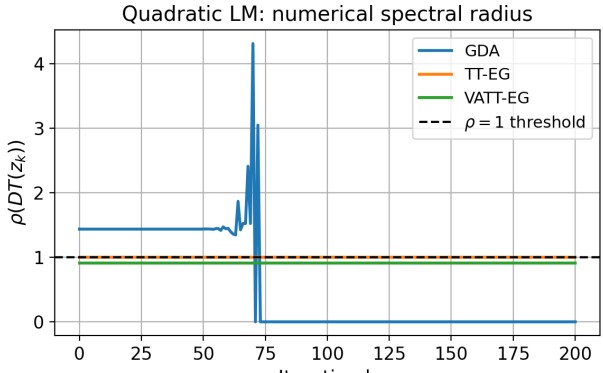

Figure 11: Quadratic local minimax game: numerical spectral radius $\rho(DT(z_k))$ along the same trajectories. Near the origin we observe $\rho(DT_{\text{TT–EG}}) \approx 1.0009 > 1$ and $\rho(DT_{\text{VATT–EG}}) \approx 0.91 < 1$, in agreement with our calm local–minimax analysis. The GDA curve quickly exceeds 1 and the iterates blow up; once $\|z_k\|$ becomes extremely large, the finite–difference Jacobian suffers from numerical overflow, which results in a spurious near–zero value at the very end of the blue curve.

stable mode along the weak ascent curvature. For VATT–EG, the anchors shift this eigenvalue strictly inside the unit circle, and the trajectory converges robustly to the local minimax point.

These controlled quadratic experiments instantiate exactly the "near–singular" regime emphasized in the theoretical part of the paper: $B$ is negative and very small in magnitude, $S \gg 0$, and there is a calm local minimax point which is a fragile attractor for TT–EG but a robust, spectrally stable attractor for VATT–EG. In the revised version, we will integrate Figures 9–11 and the above discussion into the main numerical section and the appendix, providing a direct empirical validation of the selection guarantees.

### J.3 ADDITIONAL EXPERIMENTS FOR GDA WITH REGULARIZATION

To address Reviewer 4's concern that our baselines did not include simple regularized or two-timescale variants of GDA, we performed an additional synthetic experiment on the same scalar quadratic local minimax game as in Section J.2. Recall

$$f(x,y) \;=\; \tfrac{1}{2}ax^2 + cxy - \tfrac{1}{2}dy^2, \qquad a = 0.5, \;\; d = 0.005, \;\; c = 3.0,$$

for which $A = a > 0$, $B = -d < 0$, $C = c$, and the Schur complement $S = A - CB^{-1}C^\top = a + c^2/d \gg 0$, so $(x^\star, y^\star) = (0,0)$ is a strict calm local minimax point and $B$ is near–singular.

**Algorithms and parameters.** We compare the following five methods on this game:

1. **GDA:** one–step gradient descent–ascent

$$x_{k+1} = x_k - \eta \, \nabla_x f(x_k, y_k), \qquad y_{k+1} = y_k + \eta \, \nabla_y f(x_k, y_k).$$

2. **GDA_reg:** GDA on a quadratically regularized game

$$\tilde{f}(x,y) = f(x,y) + \tfrac{1}{2}\lambda_{\text{reg}}(x^2 - y^2),$$

with $\lambda_{\text{reg}} = 0.2$, i.e.,

$$\nabla_x \tilde{f} = \nabla_x f + \lambda_{\text{reg}} x, \qquad \nabla_y \tilde{f} = \nabla_y f - \lambda_{\text{reg}} y.$$

3. **GDA_two_ts:** GDA with a fixed two-timescale preconditioner

$$x_{k+1} = x_k - (\eta/\tau_{\text{ts}}) \, \nabla_x f(x_k, y_k), \qquad y_{k+1} = y_k + \eta \, \nabla_y f(x_k, y_k),$$

with $\tau_{\text{ts}} = 5.0$.

Table 2: Quadratic LM game with near–singular ascent curvature: comparison of GDA, GDA with fixed quadratic regularization (GDA_reg), two–timescale GDA (GDA_two_ts), TT–EG, and VATT–EG. All methods use $\eta = 0.8$ and share $(x_0, y_0) = (1.0, 0.5)$; TT–EG and VATT–EG also use $\tau_{\text{EG}} = 5.0$ and $(\gamma, \beta) = (0.1, 0.1)$. VATT-EG converges to $(0, 0)$, TT-EG remains on a bounded orbit, and all three GDA variants diverge.

| method | $x_T$ | $y_T$ | $\|(x_T, y_T)\|$ | $\|\nabla f(x_T, y_T)\|$ |
|---|---|---|---|---|
| GDA | $1.24 \times 10^{80}$ | $-2.23 \times 10^{80}$ | $2.56 \times 10^{80}$ | $7.14 \times 10^{80}$ |
| GDA_reg | $1.42 \times 10^{78}$ | $-6.16 \times 10^{78}$ | $6.33 \times 10^{78}$ | $1.83 \times 10^{79}$ |
| GDA_two_ts | $1.93 \times 10^{31}$ | $-7.38 \times 10^{31}$ | $7.63 \times 10^{31}$ | $2.20 \times 10^{32}$ |
| TT_EG | $7.54 \times 10^{-1}$ | $2.12 \times 10^{0}$ | $2.25 \times 10^{0}$ | $7.12 \times 10^{0}$ |
| VATT_EG | $-2.49 \times 10^{-9}$ | $1.61 \times 10^{-8}$ | $1.63 \times 10^{-8}$ | $4.76 \times 10^{-8}$ |

4. **TT_EG:** two-timescale extragradient (no anchors) on the saddle field $(\nabla_x f, -\nabla_y f)$:

$$(x_{k+1/2}, y_{k+1/2}) = (x_k, y_k) - \eta\big(\tfrac{1}{\tau_{\text{EG}}}\nabla_x f(x_k, y_k), -\nabla_y f(x_k, y_k)\big),$$
$$(x_{k+1}, y_{k+1}) = (x_k, y_k) - \eta\big(\tfrac{1}{\tau_{\text{EG}}}\nabla_x f(x_{k+1/2}, y_{k+1/2}), -\nabla_y f(x_{k+1/2}, y_{k+1/2})\big),$$

with $\tau_{\text{EG}} = 5.0$.

5. **VATT_EG:** TT–EG with small anchors on both players, i.e. we replace the saddle field by

$$\tilde{F}_x(x, y) = \nabla_x f(x, y) + \gamma x, \qquad \tilde{F}_y(x, y) = -\nabla_y f(x, y) + \beta y,$$

with $(\gamma, \beta) = (0.1, 0.1)$, and use the same predictor–corrector structure as TT_EG.

All methods share the same stepsize $\eta = 0.8$ and are initialized at $(x_0, y_0) = (1.0, 0.5)$. We run $T = 200$ iterations. For each trajectory, we record the radius $\|(x_k, y_k)\|$ and the gradient norm $\|\nabla f(x_k, y_k)\|$ over time, as well as the final values at $k = T$.

**Numerical results.** Table 2 summarizes the final radius and final gradient norm at iteration $T = 200$; the full traces are plotted in Figures 12–13.

**Discussion of the figures.** Figure 12 plots the radius $\|(x_k, y_k)\|$ versus iteration on a logarithmic vertical scale. We observe:

- **GDA** and **GDA_reg** both blow up super-exponentially; the regularized variant diverges slightly more slowly but still reaches radii on the order of $10^{78}$ by $k = 200$.
- **GDA_two_ts** (with $x$–step reduced by $\tau_{\text{ts}} = 5$) diverges more slowly, but the radius still grows to $\sim 10^{32}$.
- **TT_EG** remains bounded with radius $\approx 2.25$, but does not converge to $(0, 0)$: the trajectory hovers on a small orbit, reflecting a near–neutral eigenvalue associated with the weak ascent curvature $B = -d$.
- **VATT_EG** contracts rapidly towards the saddle; by $k = 200$ the radius has dropped below $10^{-8}$ and the iterates are numerically at $(0, 0)$.

Figure 13 shows the norm of the gradient $\|\nabla f(x_k, y_k)\|$ versus iteration. The pattern mirrors the radius behavior: GDA and its regularized/two–timescale variants drive the gradient norm to astronomically large values, TT-EG stabilizes the gradient norm at a nonzero plateau (around 7), and VATT-EG drives the gradient norm down to $\sim 10^{-8}$. This confirms that, even when we allow GDA to use fixed regularization or a two-timescale step, it does not inherit the selection behavior we prove for VATT–EG in the calm LM setting.

Overall, these additional baselines show that simply adding fixed quadratic regularization or a two-timescale step to GDA is not sufficient to obtain the desired selection behavior in the near-singular calm LM regime. In contrast, VATT–EG, which uses vanishing anchors scaled as in our theory, converges robustly to the local minimax point and sharply separates itself from GDA-based methods and from TT-EG, whose stability margin remains extremely small.

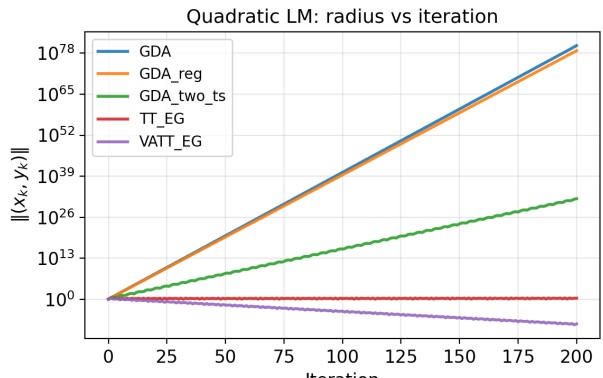

Figure 12: Quadratic LM game: radius $\|(x_k, y_k)\|$ versus iteration $k$ (log scale). All three GDA variants diverge; TT-EG is bounded but non-convergent; VATT–EG converges to the local minimax point.

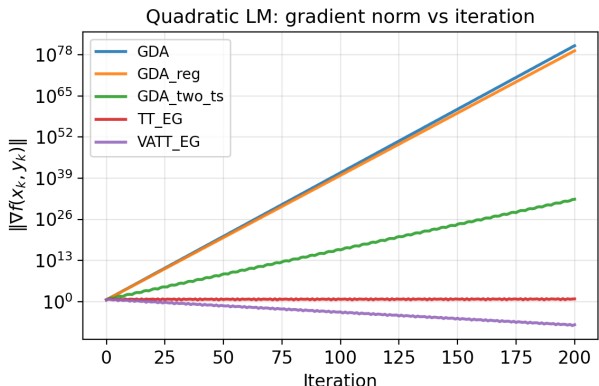

Figure 13: Quadratic LM game: gradient norm $\|\nabla f(x_k, y_k)\|$ versus iteration $k$ (log scale). GDA, GDA_reg, and GDA_two_ts become highly unstable; TT-EG is bounded but does not reduce the gradient to zero; VATT-EG achieves a much smaller gradient norm, consistent with convergence to the calm local minimax equilibrium.

## K  ADDITIONAL EXPERIMENTS: ABLATIONS FOR GDA

### K.1  ABLATION STUDY: GDA WITH QUADRATIC REGULARIZATION AND TWO−TIMESCALE

To further address Reviewer 4's suggestion that GDA with vanishing or fixed regularization and two–timescale variants should be considered as baselines, we performed a small ablation study on the same quadratic local minimax game

$$f(x, y) = \tfrac{1}{2}ax^2 + cxy - \tfrac{1}{2}dy^2, \qquad a = 0.5,\ d = 0.005,\ c = 3.0,$$

with $(x^\star, y^\star) = (0, 0)$ a strict calm local minimax point.

We consider only plain GDA variants (without extragradient) and vary two hyperparameters:

- the strength $\lambda_{\mathrm{reg}}$ of a *quadratic regularizer* applied to the game,

$$\tilde{f}(x, y) = f(x, y) + \tfrac{1}{2}\lambda_{\mathrm{reg}}(x^2 - y^2),$$

  so that GDA is run on $\tilde{f}$; and

Table 3: GDA with quadratic regularization: final iterate $(x_T, y_T)$, radius $\|(x_T, y_T)\|$, and gradient norm $\|\nabla f(x_T, y_T)\|$ for different regularization strengths $\lambda_{\text{reg}}$, with $\eta = 0.8$ and $T = 200$.

| $\lambda_{\text{reg}}$ | $x_T$ | $y_T$ | $\|(x_T, y_T)\|$ | $\|\nabla f(x_T, y_T)\|$ |
|---|---|---|---|---|
| 0.00 | $1.24 \times 10^{80}$ | $-2.23 \times 10^{80}$ | $2.56 \times 10^{80}$ | $7.14 \times 10^{80}$ |
| 0.05 | $-3.68 \times 10^{79}$ | $8.75 \times 10^{79}$ | $9.50 \times 10^{79}$ | $2.68 \times 10^{80}$ |
| 0.10 | $1.14 \times 10^{79}$ | $-3.50 \times 10^{79}$ | $3.68 \times 10^{79}$ | $1.05 \times 10^{80}$ |
| 0.20 | $1.42 \times 10^{78}$ | $-6.16 \times 10^{78}$ | $6.33 \times 10^{78}$ | $1.83 \times 10^{79}$ |
| 0.40 | $1.11 \times 10^{77}$ | $-3.12 \times 10^{77}$ | $3.31 \times 10^{77}$ | $9.43 \times 10^{77}$ |
| 0.60 | $2.66 \times 10^{76}$ | $-2.81 \times 10^{76}$ | $3.87 \times 10^{76}$ | $1.07 \times 10^{77}$ |
| 0.80 | $9.92 \times 10^{75}$ | $-1.78 \times 10^{75}$ | $1.01 \times 10^{76}$ | $2.98 \times 10^{76}$ |
| 0.90 | $6.87 \times 10^{75}$ | $1.49 \times 10^{75}$ | $7.03 \times 10^{75}$ | $2.21 \times 10^{76}$ |
| 1.20 | $8.42 \times 10^{74}$ | $1.01 \times 10^{76}$ | $1.01 \times 10^{76}$ | $3.07 \times 10^{76}$ |
| 1.40 | $-2.16 \times 10^{76}$ | $3.40 \times 10^{76}$ | $4.03 \times 10^{76}$ | $1.12 \times 10^{77}$ |

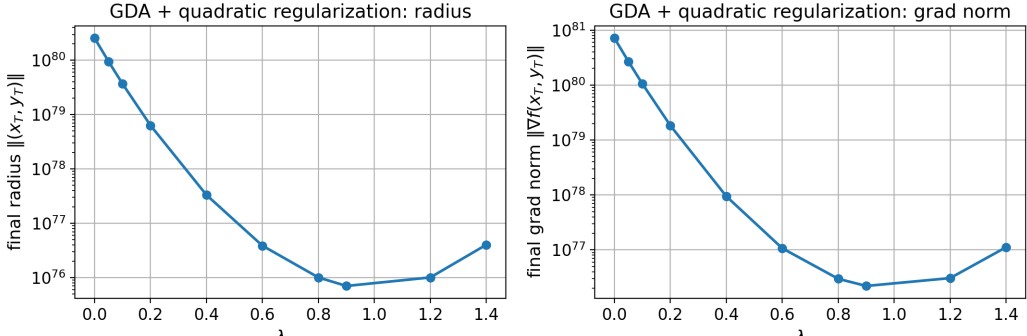

Figure 14: Ablation over quadratic regularization strength $\lambda_{\text{reg}}$ for GDA. *Left:* final radius $\|(x_T, y_T)\|$ vs. $\lambda_{\text{reg}}$. *Right:* final gradient norm $\|\nabla f(x_T, y_T)\|$ vs. $\lambda_{\text{reg}}$. Increasing $\lambda_{\text{reg}}$ slows down the divergence, but even for $\lambda_{\text{reg}} \approx 1.4$ and $T = 200$ the radii and gradient norms remain of order $10^{76}$–$10^{77}$, indicating that fixed quadratic regularization is not sufficient to stabilize GDA in this near–singular LM regime for the chosen step size.

- the *two–timescale factor* $\tau_{\text{ts}}$ in the GDA update, where we reduce the step on $x$ but keep the step on $y$ unchanged:

$$x_{k+1} = x_k - (\eta/\tau_{\text{ts}})\nabla_x f(x_k, y_k), \qquad y_{k+1} = y_k + \eta \nabla_y f(x_k, y_k).$$

All runs use the same stepsize $\eta = 0.8$, the same initial point $(x_0, y_0) = (1.0, 0.5)$, and are run for $T = 200$ iterations. For each configuration we record the final radius $\|(x_T, y_T)\|$ and the final gradient norm $\|\nabla f(x_T, y_T)\|$ of the *original* game $f$ (not the regularized one).

**Sweep over quadratic regularization.** We sweep $\lambda_{\text{reg}}$ over $\{0, 0.05, 0.1, 0.2, 0.4, 0.6, 0.8, 0.9, 1.2, 1.4\}$. Table 3 summarizes the results, and Figure 14 plots the final radius and gradient norm versus $\lambda_{\text{reg}}$ on a log scale.

We observe that increasing $\lambda_{\text{reg}}$ monotonically reduces both the final radius and the final gradient norm up to around $\lambda_{\text{reg}} \approx 0.8$, but the values remain astronomically large. Beyond $\lambda_{\text{reg}} \approx 1$, the behavior becomes non–monotone (consistent with crossing stability boundaries of the linear map), but again the radii and gradient norms stay on the order of $10^{76}$ and above. In other words, for the step size $\eta = 0.8$ that makes VATT–EG contract, GDA with any fixed quadratic regularization in this range still diverges by many orders of magnitude in this quadratic LM example.

**Sweep over two–timescale factor.** We next fix the original (unregularized) game $f$ and sweep the two–timescale factor $\tau_{\text{ts}}$ over $\{1, 2, 5, 10, 20, 40, 60, 80, 90, 200, 400\}$, where $\tau_{\text{ts}} = 1$ reduces to

Table 4: GDA with two–timescale updates: final iterate $(x_T, y_T)$, radius $\|(x_T, y_T)\|$, and gradient norm $\|\nabla f(x_T, y_T)\|$ for different $\tau_{\text{ts}}$, with $\eta = 0.8$ and $T = 200$.

| $\tau_{\text{ts}}$ | $x_T$ | $y_T$ | $\|(x_T, y_T)\|$ | $\|\nabla f(x_T, y_T)\|$ |
|---|---|---|---|---|
| 1.0 | $1.24 \times 10^{80}$ | $-2.23 \times 10^{80}$ | $2.56 \times 10^{80}$ | $7.14 \times 10^{80}$ |
| 2.0 | $-3.64 \times 10^{56}$ | $-1.31 \times 10^{56}$ | $3.87 \times 10^{56}$ | $1.23 \times 10^{57}$ |
| 5.0 | $1.93 \times 10^{31}$ | $-7.38 \times 10^{31}$ | $7.63 \times 10^{31}$ | $2.20 \times 10^{32}$ |
| 10.0 | $3.38 \times 10^{18}$ | $1.86 \times 10^{18}$ | $3.86 \times 10^{18}$ | $1.25 \times 10^{19}$ |
| 20.0 | $9.47 \times 10^{9}$ | $-5.36 \times 10^{10}$ | $5.44 \times 10^{10}$ | $1.59 \times 10^{11}$ |
| 40.0 | $-1.38 \times 10^{5}$ | $-9.51 \times 10^{5}$ | $9.60 \times 10^{5}$ | $2.95 \times 10^{6}$ |
| 60.0 | $-2.59 \times 10^{3}$ | $-1.95 \times 10^{4}$ | $1.96 \times 10^{4}$ | $6.02 \times 10^{4}$ |
| 80.0 | $-3.48 \times 10^{2}$ | $2.59 \times 10^{3}$ | $2.61 \times 10^{3}$ | $7.66 \times 10^{3}$ |
| 90.0 | $2.01 \times 10^{2}$ | $-9.31 \times 10^{2}$ | $9.53 \times 10^{2}$ | $2.76 \times 10^{3}$ |
| 200.0 | $-6.59 \times 10^{0}$ | $9.70 \times 10^{1}$ | $9.73 \times 10^{1}$ | $2.89 \times 10^{2}$ |
| 400.0 | $9.78 \times 10^{-1}$ | $-4.68 \times 10^{1}$ | $4.69 \times 10^{1}$ | $1.40 \times 10^{2}$ |

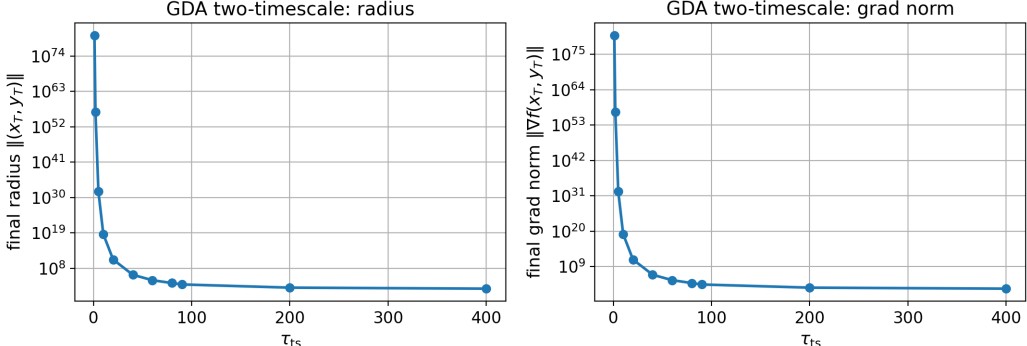

Figure 15: Ablation over two–timescale factor $\tau_{\text{ts}}$ for GDA. *Left:* final radius $\|(x_T, y_T)\|$ vs. $\tau_{\text{ts}}$. *Right:* final gradient norm $\|\nabla f(x_T, y_T)\|$ vs. $\tau_{\text{ts}}$. Increasing $\tau_{\text{ts}}$ dramatically slows down the divergence and eventually brings the method into a near–stable regime (e.g. at $\tau_{\text{ts}} = 400$, radii and gradients are $\sim 10^2$), but even then the method has not converged to $(0, 0)$ in $T = 200$ steps.

standard GDA. Table 4 summarizes the results, and Figure 15 shows the final radius and gradient norm versus $\tau_{\text{ts}}$.

The two–timescale sweep shows a similar pattern: for moderate values $\tau_{\text{ts}} \leq 20$, GDA remains wildly unstable, with radii ranging from $10^{56}$ down to $10^{10}$. Only for very large $\tau_{\text{ts}}$ (e.g. 200 or 400) do we obtain final radii and gradients on the order of $10^2$ after $T = 200$ iterations. Even then, the method is only marginally stable and clearly has not reached the local minimax equilibrium; the trajectory is still far from $(0, 0)$ and the gradient norm is not small.

**Takeaway.** These ablations demonstrate that while quadratic regularization and two–timescale adjustments can slow down the divergence of GDA, they do not automatically confer the selection behavior we prove for VATT–EG in the calm local minimax regime. For the same stepsize $\eta$ and number of iterations for which VATT-EG converges cleanly to the LM point (with $\|(x_T, y_T)\|$ and $\|\nabla f\|$ around $10^{-8}$), all GDA-based variants still exhibit very large radii and gradient norms, or at best remain near the boundary of stability without converging. This supports the claim that VATT–EG is not simply "GDA plus some regularization," but rather enjoys a distinct and stronger local stability guarantee in the calm LM setting.

