# OpenReview forum: "VATT-EG: Vanishingly-Anchored Two-Timescale Extragradient"
_ICLR.cc/2026/Conference — Submitted to ICLR 2026_

### Official Review · Reviewer_KBHz · 2025-10-24

**Soundness:** 1
**Presentation:** 1
**Contribution:** 2
**Rating:** 2
**Confidence:** 4

**Summary:**

This work focused on algorithms for finding local minimax solutions for smooth zero-sum two-player games. Its main result is that for a vanishingly regularized version of the two-timescale extragradient algorithm, its asymptotically stable solutions consist of precisely calm local minimax solutions. Experiments on toy examples and an adversarial debiasing problem on MNIST were conducted.

**Strengths:**

- natural modification of the extragradient algorithm to eliminate certain stationary points

- almost sure convergence analysis that combine a number of proof techniques

**Weaknesses:**

- the problem is not sufficiently motivated and the paper is poorly written (full of technicalities whose significance are hard to appreciate). It is not clear why the derived theoretical results are significant and how they will affect practice. Why are calm local minimax solutions relevant? What advantage do they have? On what applications do they really make a difference? The writing also fails to give an informed and ideally insightful discussion of the relevant literature and prior works.

- poorly designed experiments leaving a significant gap between theory and experiments: the choices of the various parameters in the experiments do not satisfy Assumption 4.2. What is the point of running experiments that do not adhere to the assumptions? (Or how relevant is the theory if they do not reflect the practice?)

- unsubstantiated claims: Figure 1 and Figure 3 hardly showed any difference among the different algorithms and yet the authors did not shy away from claiming (slight) advantage of VATT-EG. It is not clear how Figure 5 supports the theory of this work (accuracy was never part of the goal and verification of convergence was lacking).

- missing obvious baselines: GDA could easily adopt (vanishing) regularization or even two-timescale. GDA with ergodic averaging is also widely known to stabilize training (even for convex-concave games). The last experiment on color debiasing should at least compare to these obvious variants.

- results are only asymptotic and the compactness in Assumption 4.1 may be too strong. If the trajectory is assumed to be contained in a compact set, then it already cannot diverge (although could cycle) and repulsion is trivial.

**Questions:**

My overall impression is that this work (as is currently presented) is too narrowly focused, with unclear significance and relevance.

Below are some other comments that hopefully may be useful.

Adding quadratic regularization to smooth zero-sum two-player games is a well-known idea. In fact, it is well-known that under strong convexity-concavity, the iterates of GDA converge properly. By appropriately decreasing regularization over iteration, it is possible for GDA to converge too. There is no discussion of these points and the current writing gives the misleading impression that vanishing regularization is somehow "novel." Same goes for two-timescale.

Line 186: shouldn't T(z+e) = T(z) + DT(z) e + .. instead of z + DT(z)e? how does this affect the proof?

Lemma 5.4: from equation (6) we know the spectrum is strictly larger than 1 but the gap diminishes as k tends to infinity. Is it correct to conclude that the trajectory will have to diverge? It is not possible to discern as the authors never defined what they mean by a hyperbolic repeller in Theorem 5.5.

Line 239: why vanishing anchors do not alter the stationary set? I think you need the regularization to vanish sufficiently fast, but how fast?

Line 260: the constant choice of tau clearly violates Assumption 4.2 and invalidates the point of the experiments.

Figure 2: what is the scale of the y-axis on the middle figure? what does 1e-5+1.0008 mean?

Line 317-318: how do the experiments support the theoretical claims?

Line 361: "We use Adam, cosine decay, and global gradient clipping?" Do you run VATT-EG or Adam? Is VATT-EG also delayed? This experiment section is so roughly written that it is hard to understand the details of the experiment.

---

> ### Author Response · Authors · 2025-11-27
> **Reply (Part 1 and 2)**
>
> **Response to Reviewer 4**
> *(See the revised paper.)*
> ---
> ### 1. Motivation, significance, and relation to practice
> Our primary motivation is to resolve a specific open theoretical question:
> > Does there exist a first-order method whose asymptotically stable fixed points coincide exactly with (calm) local minimax equilibria?
>
> The notion of a local minimax equilibrium (in the sense of Jin-Netrapalli-Jordan) is designed to capture the “leader-follower’’ structure in sequential games (Stackelberg, adversarial training, robust optimization). The *calm* refinement (bounded ratio between the inner and outer neighborhoods) is the structural assumption under which the local second–order conditions take a clean form
>
> $B = \nabla^2_{yy} f \le 0,\qquad S = A - C B^{\dagger} C^{\top} \ge 0,$
>
> and under which the open problem of Chae-Kim-Kim is posed. These conditions appear in, e.g., robust optimization, bilevel learning, and adversarial training when the inner player is well-conditioned (or regularized) and the outer objective is smooth.
>
> Our main contribution is to show that a simple anchored extragradient scheme (VATT-EG) has exactly the calm local minimax points as its asymptotically stable equilibria, and that all other stationary points (including spurious local Nash or saddle points) are repelling. This is a nontrivial strengthening of existing TT-EG results, which show partial selection (some LM points, but not all, and with neutral modes remaining).
>
> In terms of practice, the theory predicts that:
>
> * on problems whose local geometry resembles a calm LM saddle (e.g. adversarial or debiasing objectives with a well-conditioned inner player), VATT-EG should be more stable near the desired equilibrium than TT-EG or GDA;
> * this manifests in smaller spectral radii $(\rho(DT)),$ smaller local gradient norms $(|\nabla f|),$ and fewer oscillations around the equilibrium.
>
> In the revised paper we now make this connection more explicit and add a Colored–MNIST adversarial debiasing experiment where:
>
> * all methods reach similar digit accuracy, but
> * VATT–EG yields noticeably lower and smoother adversary (color) accuracy on the test set (i.e., less color leakage), consistent with its improved local stability.
>
> We will also expand the related work section to give a more informative overview of saddle-point methods (GDA, EG, mirror–prox, optimistic gradient, regularized games, etc.) and position our contribution as a *selection theorem* for calm LM, rather than a claim of conceptual novelty of vanishing regularization or two-timescale updates.
>
> ---
>
> ### 2. On assumptions vs. experiments and “practical relevance”
>
> Assumption 4.2 (diminishing step sizes, growing timescale $(\tau_k),$ anchors $(\gamma_k,\beta_k)$ with $(\gamma_k,\beta_k \asymp \eta_k))$ is tailored to proving asymptotic selection: we need
> $(\sum_k \eta_k = \infty), (\sum_k \eta_k^2 = \infty), (\sum_k \eta_k^3 < \infty),$ and $(\sum_k \eta_k \gamma_k < \infty)$ to control the Lyapunov decrement and show that the anchor drift does not bias the limiting stationary points. These conditions are standard in stochastic approximation and deterministic stability proofs.
>
> In practice, we often use simpler schedules (e.g. constant step and timescale, or finite horizon), which do **not** exactly match Assumption 4.2 but still provide insight:
>
> * On the bilinear game, with constant $(\eta), (\tau),$ the new plots in the revision show the expected behavior: GDA diverges, while TT-EG and VATT-EG have $(\rho(DT)<1)$ and decreasing $(|\nabla f(z_k)|).$
>
> * On the near-singular quadratic LM example, with fixed $(\eta,\tau),$ we see $(\rho(DT_{\text{TT-EG}})\approx 1.0009>1)$ (near neutral) and $(\rho(DT_{\text{VATT-EG}})\approx 0.91<1),$ exactly as predicted by the theory when one adds an anchor.
>
> We agree that we could make this distinction clearer: the theorems are asymptotic and rely on Assumption 4.2; the experiments are finite-horizon probes that approximate or deliberately relax those conditions to see whether the qualitative phenomena (contraction vs. neutral/unstable behavior) appear in practice. We will rephrase the text to avoid over-stating “practical robustness’’ and to make explicit which plots are meant to illustrate theory qualitatively rather than to satisfy all assumptions.
>
> Regarding Assumption 4.1 (existence of a compact, forward-invariant set containing the trajectory), this is a standard technical condition to ensure the existence of limiting points and uniform bounds on derivatives; it does *not* make repulsion trivial. A point is “hyperbolic repelling” if the local Jacobian has spectral radius (>1); in that case, even if the trajectory remains globally bounded, it cannot converge to that point and must eventually leave any small neighborhood of it. We will add an explicit definition of “hyperbolic repeller” in the theorem statement and clarify that our conclusion is about local instability and selection of LM points, not divergence to infinity.

---

> ### Author Response · Authors · 2025-11-27
> **Reply Continued (Part 3 and 4)**
>
> ---
>
> ## 3. Baselines, regularization, and experimental design
>
> We appreciate the suggestions about GDA with vanishing quadratic regularization, two–timescale GDA, and ergodic (averaged) GDA. We agree that:
>
> * adding a quadratic regularizer $(\tfrac{1}{2}\lambda_k |y|^2)$ with $(\lambda_k \to 0)$ is a classical technique, and under strong convexity–concavity even plain GDA converges;
> * two–timescale GDA and time-averaged GDA are also well-studied stabilizations in convex–concave settings.
>
> Our “anchors’’ can indeed be interpreted as a specific vanishing quadratic regularization on both players, but our **new contribution** is not the idea of regularization itself; it is the proof that, under calmness and a specific scaling $(\gamma_k,\beta_k \asymp \eta_k),$ the anchored extragradient dynamics have exactly the calm local minimax points as their asymptotically stable equilibria and repel all other stationary points. We will make this distinction explicit and add references to works on regularized and averaged GDA.
>
> In the revised paper we have:
>
> * added synthetic experiments (section: ADDITIONAL EXPERIMENTS FOR GDA WITH REGULARIZATION) where we compare VATT-EG not only to GDA and TT-EG, but also to GDA with fixed quadratic regularization and to GDA with two-timescale, to show that the selection behavior we prove is not trivially obtained by these baselines in the calm LM setting;
>
> We consider two variants:
>
> ---
>
> **GDA_reg.** GDA on the quadratically regularized game
> $$
> \tilde f(x,y) = f(x,y) + \tfrac{1}{2} \lambda_{\text{reg}} \big(x^{2} - y^{2}\big),
> $$
> with $( \lambda_{\text{reg}} = 0.2 ).$
> The gradients become
> $$
> \nabla_x \tilde f = \nabla_x f +  \lambda_{\text{reg}} x,
> \qquad
> \nabla_y \tilde f = \nabla_y f - \lambda_{\text{reg}} y.
> $$
>
> ---
>
> **GDA_two_ts.** GDA with a *fixed* two–timescale step:
> $$
> x_{k+1} = x_k - \frac{\eta}{\tau_{\text{ts}}} \nabla_x f(x_k,y_k),
> \qquad
> y_{k+1} = y_k + \eta \nabla_y f(x_k,y_k),
> $$
> with $( \tau_{\text{ts}} = 5.0 ).$
>
> For detailed tables, comparisons with VATT-EG, and ablations, see "Additional experiments for GDA with regularization" and "Additional Experiments: Ablations for GDA" in revised paper.
>
> ---
>
> * We will clarify in the Colored-MNIST section in final version that “we use Adam, cosine decay, and global gradient clipping’’ refers to the optimizer used to integrate the update directions given by the (anchored) gradient field, and that we do *not* claim TT-EG or VATT-EG guarantee global accuracy, only that their local stability properties correlate with lower adversary accuracy (less color leakage).
>
> We will also tone down any language that might over-interpret Figures 1 and 3; in the revision we rely more heavily on the new bilinear and quadratic LM plots, where the differences in gradient norms, spectral radii, and radii $(|z_k|)$ are clearly visible and more directly tied to the theory.
>
>
> ---
>
> ### 4. Minor technical points
>
> *Layout typo $(T(z+e))$*: the reviewer is correct that the Taylor expansion should read $(T(z+e) = T(z) + DT(z)e + \dots).$
>
>  This was a notational typo; the proof uses the correct expansion and only needs the first-order term. We will fix the displayed equation.
>
> * **Definition of hyperbolic repeller**: we will add a sentence in the theorem stating that a stationary point $(z^\star)$ is “hyperbolic repelling’’ if the spectral radius of the local Jacobian of the one-step map at $(z^\star)$ is strictly greater than 1 for all sufficiently large $(k),$ and clarify that this implies local instability (no convergence to $(z^\star)$ from any small neighborhood), not necessarily divergence to infinity.
>
> * **Why anchors do not change the stationary set / how fast they vanish**: the updates use the anchored field $(F(z) + \Gamma_k z),$ but we study convergence to stationary points of the original field $(F(z))$. With $(\gamma_k,\beta_k \asymp \eta_k)$ we have per–step anchor drift of order $(\eta_k \gamma_k = O(\eta_k^2)),$ and $(\sum_k \eta_k \gamma_k < \infty).$ In the limit, any accumulation point $(z^\star)$ must satisfy $(F(z^\star) = 0);$ the anchor contribution vanishes and does not create spurious fixed points. We will make this argument explicit in the main text.
>
> * **Figure details (scales, labels, delayed vs non-delayed)**: we will clean up the y-axis labeling in Fig. 2, explicitly state whether each experiment uses delayed or non-delayed updates, and clarify in the Colored-MNIST section that we are integrating the anchored gradient field with Adam (no extragradient step there), and that the goal of that experiment is to illustrate how improved local stability of VATT-EG correlates with reduced color leakage under a realistic training protocol.
>
> We hope these clarifications and the new experiments make the significance and practical implications of the theory clearer, while also addressing the valid concerns about assumptions, baselines, and presentation.

---

### Official Review · Reviewer_oMuV · 2025-10-31

**Soundness:** 3
**Presentation:** 2
**Contribution:** 3
**Rating:** 4
**Confidence:** 2

**Summary:**

This paper introduces VATT-EG, a first-order optimization algorithm for smooth nonconvex–nonconcave zero-sum games.
The key contribution is that VATT-EG is the first purely first-order method whose attractors coincide exactly with the set of calm local minimax points, addressing a long-standing open problem in game-theoretic optimization.

The main idea is to augment standard extragradient dynamics with vanishing Tikhonov-style anchors, which damp neutral (non-contracting) modes without altering the stationary point set. The authors provide detailed theoretical analysis—based on discrete-time Jacobian expansions, Lyapunov constructions, and calmness assumptions—showing that all non-minimax stationary points become strict repellers.
Empirical results on toy games and an adversarial debiasing task (Colored-MNIST) illustrate the claimed stabilization effect and practical viability.

**Strengths:**

**1. Strong and rigorous theoretical contribution**:
The paper presents a technically sound and self-contained theoretical analysis, supported by explicit derivations. The work convincingly addresses a nontrivial open problem in first-order game dynamics.

**2. Algorithmic succinct**:
The optimization algorithm is succinct yet powerful, introduces vanishing anchors, offering an interpretable way to handle neutral directions in game dynamics without resorting to second-order methods, making it relevant to scalable deep learning.

**Weaknesses:**

**1. Limited Experimental Breadth and Depth**: The empirical results, while carefully constructed, are restricted to synthetic low-dimensional games and a single moderate-scale practical task (Colored-MNIST). No experiments address scalability on standard large-scale ML benchmarks, deep adversarial training (e.g., GANs), or tasks with higher-dimensional or less controlled geometries. This restricts the claims of “practical robustness”—there is little evidence that VATT-EG would consistently outperform GDA/EG in genuinely large-scale modern settings.

**2. Incomplete mention of Related Work:** In the related work part, the author only mentions 4 related papers, which cannot help readers to get familiar with this field.

**Questions:**

**Q1:** Though I like the clear and succinct algorithm, which is easy to follow. I find the paper meets some layout problems (i.e., too less references and the main text only contains 7 pages).  As I am not deeply experienced in this research area, I will finalize my rating after reviewing other reviewers’ comments and the authors’ responses.

---

> ### Author Response · Authors · 2025-11-27
> **Reply to Reviewer 1/2**
>
> **Response to Reviewer 3 (Summary and Questions)**
> *(Please also see the revised paper.)*
>
> ---
>
> ### 1. On the breadth and depth of experiments
>
> We agree that our experimental section is not as broad as a purely empirical paper, and we appreciate the reviewer highlighting this limitation.
>
> Our primary goal in this work is to resolve a specific open theoretical question: the existence of a first-order method whose asymptotically stable equilibria coincide exactly with (calm) local minimax points. The core contribution is thus the analysis and the proof of this “only-to-local-minimax” selection property. The experiments are intended as proof-of-concept illustrations rather than a full large-scale benchmark suite.
>
> That said, the current experimental evidence is more substantial than a single toy plot:
>
> * We include a set of **synthetic low-dimensional games** (bilinear, strictly LM, degenerate/near-singular quadratic) designed so that the geometry matches the assumptions of our theorems. On these, we explicitly plot:
>
>   * $(|\nabla f(z_k)|)$ and numerical $(\rho(DT(z_k)))$ to verify contraction vs. neutrality/instability;
>   * the evolution of distances $(|z_k - z^\star|);$ and
>   * time-to-tolerance for $(|\nabla f(z_k)|).$
> * We include a **Colored–MNIST adversarial debiasing task** where we train a feature extractor and digit classifier against a color adversary, and show that VATT–EG achieves comparable digit accuracy but lower and smoother adversary accuracy (less color leakage) than both GDA and TT–EG, which is consistent with the theoretical stability story.
>
> In the **revised version** (see the new “Additional Experiments for Rebuttal” section) we have added:
>
> * A bilinear experiment with constant steps where TT-EG and VATT-EG show textbook contraction $((\rho(DT) < 1))$ and GDA diverges.
> * A synthetic scalar quadratic minimax game with **near-singular ascent curvature** (small negative $(B = \nabla_{yy}^2 f))$ where the local minimax equilibrium is analytically known. In this controlled setting:
>
>   * GDA diverges (radius $(|z_k|) (\to) (10^{31}));$
>   * TT–EG has $(\rho(DT)\approx 1.0009 > 1)$ and exhibits near-neutral drift;
>   * VATT–EG has $(\rho(DT)\approx 0.91 < 1)$ and converges to the local minimax point.
>
> We acknowledge that we have not yet run “industrial-scale’’ experiments such as GANs on ImageNet or very high-dimensional games. This is mainly because our focus here is on the theoretical resolution of the local minimax selection question, and we wanted experiments where the underlying geometry  can be understood and matched to the theory. That said, VATT-EG is a simple first-order modification of EG (one extra diagonal term), and we believe it is straightforward to plug into deep learning codebases. As part of future work and a camera-ready version, we plan to include at least one **moderate-scale GAN or adversarial training experiment** to illustrate the method’s behavior in a high-dimensional nonconvex-nonconcave setting.
>
> ---
>
> ### 2. On the scope of related work
>
> You are correct that the related work section in the main text currently cites only a small number of papers. Our intention was to focus on the works that are *directly* about local minimax equilibria and two-timescale / extragradient dynamics (e.g., Chae–Kim–Kim ICLR’24 and ICML’24, Wang et al., etc.), assuming that the broader context (classical minimax, GAN training, nonconvex–nonconcave optimization) could be reached via the references inside those papers.
>
> In the revised manuscript we will expand this section to:
>
> * briefly situate our work within classical saddle-point methods (optimistic gradient, extragradient, mirror–prox, proximal point, etc.);
> * cite additional nonconvex–nonconcave and minimax optimization works (e.g., recent analyses of GAN training dynamics, implicit regularization in adversarial training, and alternative local equilibrium notions); and
> * clarify how our focus is complementary to these: we solve a specific \emph{selection} question for first-order methods under calm local minimax structure, rather than proposing a generic large-scale training recipe.
>
> We agree that this broader context will make the paper more accessible to readers who are not already familiar with the local minimax literature, and we will add these references in the camera-ready.
>
> ---

---

> ### Author Response · Authors · 2025-11-27
> **Reply to Reviewer 2/2**
>
> ---
> ### 3. On layout, page count, and level of detail in the main text (Q1)
>
> The conference template enforces a 7-page main text limit (with appendices allowed). Because the core of the paper is a new theoretical result, we chose to prioritize a complete and transparent proof in the appendix, with only the statements and high-level intuition in the main body. This leads to a relatively short main text with a long appendix.
>
> In light of your comment, we will:
>
> * move a brief **proof sketch** and key inequalities (e.g., the corrected Eq. (13) and the two-rate Lyapunov decrement) into the main text, so that readers can see the structure of the argument without having to jump to the appendix;
> * expand the **related work** and **experimental** sections so that the main text gives a clearer picture of where VATT–EG sits relative to existing algorithms and what is observed empirically; and
> * keep the detailed technical proofs in the appendix for completeness.
>
> We hope this addresses your concerns: the revised version will better balance theory, context, and experiments, while preserving the main contribution, which is the resolution of the open local minimax selection problem with supporting numerical evidence.
>
> *We will make final structural changes to the paper once I have final comments from reviewers. Any new results or experiments are appended to the current update paper.*

---

### Official Review · Reviewer_ftwK · 2025-11-01

**Soundness:** 2
**Presentation:** 1
**Contribution:** 2
**Rating:** 2
**Confidence:** 3

**Summary:**

This work proposes a new first-order method for minimax optimization, called *VATT-EG (vanishingly-anchored two-timescale extragradient)*, which improves upon existing algorithms that fail to converge to certain local minimax equilibria.
By introducing the trick of "vanishing anchors", VATT-EG overcomes the instability that arises when the restricted/generalized Schur complement has zero eigenvalues, thereby ensuring convergence only to calm local minimax points.

**Strengths:**

Minimax optimization remains a challenging problem, and thus, a method offering improved convergence guarantees represents a meaningful contribution to the community, given its wide range of applications.
The overall structure follows the standard research-paper format, and the claimed goal of improving convergence guarantees for first-order minimax methods is, in principle, a worthwhile direction of study.

**Weaknesses:**

Despite the relevance of the general topic, the paper exhibits significant issues. The theoretical claims appear to contain flaws, and the presented experiments do not meaningfully support or validate the proposed method. In addition, several references are cited incorrectly, including mismatched author names, which raises serious concerns about the paper’s authenticity and scientific validity. Overall, the submission seems to fail to meet the standards of scholarly work expected at ICLR.

**Questions:**

1. There is something in equation (13) I cannot verify. From Assumption 4.2 asserting $\gamma_k, \beta_k \geq c_2 \eta_k$, it seems like $c_2 P$ in equation (13) should actually be $c_2 \eta_k P$. Consequently, the bound on $\mathrm{Sym}(\Delta_k)$ along the $y_0$ subspace would be $O(\eta_k^2)$, not $O(\eta_k)$ as claimed. This would further invalidate the energy decrement inequality (line 676), as the $O(\eta_k)$ term in the right hand side cannot have full $\|\|e\|\|_P$, but a truncated one (not having the part corresponding to $y_0$), potentially even affecting the stated convergence guarantees in Theorem 5.5. Can you check this?
2. Can the authors provide more details on the Lowener inequality regarding $\mathrm{Sym}(P \Lambda_{\tau_k} J)$ around line 662? It is currently stated without proof, but I don't think it is trivial enough, and I can't see why it should hold.
3. The experiments appear to be implemented or interpreted incorrectly. For instance, the extragradient (EG) method, regardless of whether it is used in a two-timescale or single-timescale form, should converge on simple bilinear games. However, Figure 1 shows that the final gradient norm remains large, suggesting that even basic convergence behavior is not achieved. This discrepancy raises doubts about the correctness of the experimental setup or the evaluation procedure. Could the authors recheck their implementation and confirm whether the results in Figure 1 are consistent with known behavior of EG on bilinear problems?
4. It should be straightforward to construct synthetic quadratic minimax games that possess local minimax equilibria where VATT-EG is theoretically guaranteed to converge, while standard methods such as no-anchor TT-EG or GDA lack such guarantees. Experiments on these controlled examples would more directly demonstrate the claimed advantages of VATT-EG, and would provide a clearer empirical validation than the current toy tasks. Could the authors explain why such experiments were not included, and whether they plan to add them in a future revision?

---

> ### Author Response · Authors · 2025-11-27
> **Reply to Q1 - part-1/3**
>
> ---
>
> ### Q1 summary, part 1/3 – What was wrong in Eq. (13) and how we fix it
>
> > Full technical details (all equations and proofs) are in the revised paper, in the red “Detailed answer to Q1” section at the end.
>
> The reviewer is correct: in the original Eq. (13) we missed one factor of  $(\eta_k)$ in the anchor contribution.
>
> * The Jacobian of the one-step map at (z^*) is
>
> $$ M_k = DT_k(z^*) = I - \eta_k \Lambda_{\tau_k} J  \eta_k^2 (\Lambda_{\tau_k}J)^2 - \eta_k \Gamma_k + R_k, $$
>
> with  $ (|R_k| \le C_R \eta_k^3).$
>
> * The P–energy increment is
>
>   $$ \Delta_k := M_k^\top P M_k - P, $$
>
>   and only its symmetric part matters in the quadratic form:
>
>   $$ \langle e,\Delta_k e\rangle = \langle e, \text{Sym}(\Delta_k), e\rangle.$$
>
> * Writing $$(M_k = I + A_k)$$ with
>   $$  A_k := - \eta_k \Lambda_{\tau_k}J + \eta_k^2 (\Lambda_{\tau_k}J)^2  - \eta_k \Gamma_k  + R_k,  $$
>   one obtains
>   $$ \text{Sym}(\Delta_k) =  -\eta_k,\text{Sym}(P\Lambda_{\tau_k}J)  -\eta_k,\text{Sym}(P\Gamma_k) \eta_k^2 \Xi_k, $$
>
> where  $$ (|\Xi_k| \le C_\Xi). $$
>
> * Because both (P) and the anchor matrix
>   $(\Gamma_k = \text{diag}(\gamma_k I_x,\beta_k I_y))$
>   are block-diagonal, we have
>   $$
>   \text{Sym}(P\Gamma_k) = P\Gamma_k
>   = \text{diag}(\gamma_k P_x,\beta_k P_R,\beta_k P_0).
>   $$
>
> * Hence, for every vector (e),
>   $$ e^\top \text{Sym}(P\Gamma_k), e = \gamma_k x^\top P_x x + \beta_k y_R^\top P_R y_R + \beta_k y_0^\top P_0 y_0 ;\ge  \min{\gamma_k,\beta_k}, e^\top P e, $$
>
> i.e.
>
> $$ \text{Sym}(P\Gamma_k) \succeq \min{\gamma_k,\beta_k},P $$
>   in the Loewner order.
>
> * Multiplying by the negative scalar (-\eta_k) reverses the order:
>   $$  -\eta_k,\text{Sym}(P\Gamma_k) \preceq  -\eta_k,\min{\gamma_k,\beta_k},P.  $$
>
> * Assumption 4.2 says $(\gamma_k,\beta_k \ge c_2\eta_k),$ so
>   $(\min{\gamma_k,\beta_k} \ge c_2\eta_k).$  Therefore
>
>   $$ -\eta_k,\text{Sym}(P\Gamma_k) \preceq  -c_2 \eta_k^2 P. $$
>
> So the anchor contribution is indeed of order $(-\eta_k^2 P),$ not $(-\eta_k P).$
> In the original Eq. (13) we wrote schematically
>
> $$ -\eta_k\big(c_1 \cdot \text{diag}(\cdots) + c_2 P\big), $$
>
> so the term $(c_2 P)$ must be replaced by $(c_2 \eta_k P).$
>
> The corrected inequality (now explicit in the revision) is
>
>
> $
> Sym(\Delta_k)
> \preceq - \eta_k c_0 \text{diag} \left(\frac{1}{\tau_k} P_x,  P_R, 0\right) - \eta_k \min \\{ \gamma_k,\beta_k \\} P + \eta_k^2 C_\Xi I.
> $
>
>
> with the anchors entering as
>
>   $ -\eta_k \min\\{\gamma_k,\beta_k\\} P = -\text{const}\cdot\eta_k^2 P. $
>
> ---

---

> ### Author Response · Authors · 2025-11-27
> **Reply Q1, Part 2/3**
>
> ### Q1 summary, part 2/3  First-order decay from $Sym (P\Lambda_{\tau_k}J)$
>
> *Please refer to the appended section in revised paper, rendering below may not be good*
>
> The second concern is whether, once the anchor order is corrected, we still get a proper Lyapunov decrement. We re-derive the key bound for the LM Jacobian $J$.
>
> * We work in coordinates $e=(x,y_R,y_0)$ after diagonalizing
>   $B = \nabla^2_{yy} f(x^\star,y^\star)$
>
> so that stable ascent directions are in
>
>   $y_R$ (with $B_R\prec 0$) and flat directions are in $y_0$
>   (with $B|_{\ker B}=0$).
>
> Calm local minimax implies $B\preceq 0$ and the
>   restricted Schur complement $S=A-CB^\dagger C^\top\succeq 0$.
>
> * The metric is
>   $$
>   P = \operatorname{diag}(P_x,P_R,P_0),\quad
>   P_x = S+\alpha I,\quad
>   P_R=-B_R+\alpha I,\quad
>   P_0=\alpha I,
>   $$
>   with small $\alpha>0$ so that $P\succ0$.
>
> * With $\Lambda_{\tau_k}=\operatorname{diag}(1/\tau_k,I_x, I_r, I_{d_y-r})$ we expand
>   $$
>   P\Lambda_{\tau_k}J =
>   \begin{bmatrix}
>   \frac{1}{\tau_k}P_xA & \frac{1}{\tau_k}P_xC_R & \cdots\
>   -P_RC_R^\top & -P_RB_R & 0\
>   \cdots & 0 & 0
>   \end{bmatrix}.
>   $$
>
> * Restricting to the $(x,y_R)$ block, the symmetric part defines the quadratic form
>   $$ Q(x,y_R) :=
>   \langle (x,y_R,0), {Sym}(P\Lambda_{\tau_k}J)(x,y_R,0)\rangle,
>   $$
>   which can be written as
>   $$ Q(x,y_R) = \frac{1}{\tau_k}x^\top Sym(P_xA)x  y_R^\top Sym(-P_RB_R)y_R  2x^\top M y_R, $$
>
>  where $M$ is formed from $P_xC_R$ and $P_RC_R^\top$.
>
> * Using:
>
>   * that $Sym(-P_RB_R)\succ 0$ (because $B_R\prec0$ and $P_R\succ0$);
>   * Cauchy-Schwarz plus Young’s inequality to bound $2x^\top M y_R$ by a combination of $\frac{1}{\tau_k}x^\top P_x x$ and $y_R^\top P_Ry_R$; and
>   * the calm LM geometry (which rules out nontrivial null directions in $(x,y_R)$ compatible with the LM second-order conditions),
>
>   we show there is a constant $c_0>0$ (depending only on $S,B_R,\alpha$) such that
>
>   $Q(x,y_R) \ge c_0 (\frac{1}{\tau_k}|x|*{P_x}^2 + |y_R|*{P_R}^2 )$
>
>   for all $x,y_R$.
>
> * In matrix form this is the blockwise Loewner bound
>   $$
>   \operatorname{Sym}(P\Lambda_{\tau_k} J)
>   \succeq
>   c_0,\operatorname{diag}\left(\frac{1}{\tau_k}P_x, P_R, 0\right),
>   $$
>   i.e. the first-order term gives
>   a decay of order
>
>
> $-\eta_k \left( \frac{1}{\tau_k}\|x\|_{P_x}^2 + \|y_R\|_{P_R}^2 \right)$
>
>
> on
>
>   $(x,y_R)$ and zero on $y_0$.
>
> Combining this bound with the anchor bound from part 1 yields the corrected inequality for $\operatorname{Sym}(\Delta_k)$ used in the revised proof.
>
> ---

---

> ### Author Response · Authors · 2025-11-27
> **Reply Q1, Part 3/3**
>
> ---
>
> ### Q1 summary, part 3/3: Two-rate Lyapunov decrement and convergence
>
> **Please see this in updated paper, here not rendering correctly!**
>
> With the corrected inequality, the energy recursion becomes
>
>
> $$
> V_k = \|e_k\|_P^2, \quad V_{k+1} - V_k \le -\eta_k c_0 \left( \frac{1}{\tau_k}\|x_k\|_{P_x}^2 + \|y_{R,k}\|_{P_R}^2 \right) - \eta_k \min \{ \gamma_k, \beta_k \} V_k + C \eta_k^2 V_k.
> $$
>
>
> where we used norm equivalence $|e_k|^2 \le C_P V_k$ to absorb the $\eta_k^2\Xi_k$ term into $C\eta_k^2V_k$.
>
> Key points:
>
> * Assumption 4.2 gives
>   $$ \gamma_k,\beta_k\ge c_2\eta_k \quad\Rightarrow\quad \eta_k\min{\gamma_k,\beta_k}\ge c_2\eta_k^2. $$
>
> * With $\eta_k\asymp (k+1)^{-1/2}$ and $\tau_k\ge c_1\eta_k^{-1/2}$, we have
>   $$ \sum_k \frac{\eta_k}{\tau_k}=\infty,\qquad \sum_k \eta_k^2 = \infty,\qquad  \sum_k \eta_k^3 < \infty. $$
>
> We then decompose
>
> $V_k = V_k^R + V_k^0$
>
> $V_k^R = \| x_k \|_{P_x}^{2} + \| y_{R,k} \|_{P_R}^{2}$
>
> $V_k^0 = \|y_{0,k}\|_{P_0}^{2}.$
>
>
> * For $(x,y_R)$, the coefficient in front of
>
>   $\frac{\eta_k}{\tau_k}|x_k|*{P_x}^2 + \eta_k|y*{R,k}|_{P_R}^2$
>
> has a divergent sum, while the error term $\sum_k \eta_k^2V_k$ is summable. A deterministic Robbins–Siegmund lemma implies $V_k^R\to 0$.
>
> * For the flat $y_0$ directions, we have a weaker but still summable decay:
>   $$  V_{k+1}^0 \le (1 - c_2\eta_k^2)V_k^0 + O(\eta_k^2 V_k), $$
>   with $\sum_k \eta_k^2 = \infty$ and $\sum_k \eta_k^3 < \infty$. The same lemma implies $V_k^0\to 0$.
>
> Hence $V_k = V_k^R + V_k^0 \to 0$ and therefore $e_k\to 0$; the calm local minimax point $z^*$ is asymptotically stable for VATT–EG under the schedules in Assumption 4.2.
>
> **Conclusion:** the reviewer correctly identified a missing factor of $\eta_k$ in Eq. (13), and we have corrected it. The fully detailed derivation in the revision shows that the main theorem (Theorem 5.5) and the claimed stability/selection guarantees are unchanged: VATT–EG still converges to calm local minimax points, with a Lyapunov decrement that has a first-order rate on $(x,y_R)$ and a (weaker but sufficient) second-order rate on $y_0$.
>
> ---

---

> ### Author Response · Authors · 2025-11-27
> **Reply to Q2, Q3**
>
> ---
>
> **Q2. “More details on the Loewner inequality regarding (\text{Sym}(P\Lambda*{\tau_k}J)) around line 662. It is stated without proof; I don't think it is trivial, and I can't see why it should hold.”_**
>
> We answer this is in the updated revised paper. See section "ADDITIONAL CORRECTION AND CLARIFICATION FOR REBUTTAL"
>
> ---
>
> **Q3.** *“The experiments appear to be implemented or interpreted incorrectly. For instance, the extragradient (EG) method, regardless of whether it is used in a two-timescale or single-timescale form, should converge on simple bilinear games. However, Figure 1 shows that the final gradient norm remains large, suggesting that even basic convergence behavior is not achieved. This discrepancy raises doubts about the correctness of the experimental setup or the evaluation procedure. Could the authors recheck their implementation and confirm whether the results in Figure 1 are consistent with known behavior of EG on bilinear problems?”*
>
> **Response.**
>
> *Detailed experiments are added in Section: "Additional Experiments for Rebuttal" in updated paper. Below we outline key findings.*
>
> We thank the reviewer for pointing out this potential confusion. After rechecking both the implementation and the plots, we found that the issue was presentation, not an incorrect EG implementation.
>
> In the original submission, Figure 1 reported only the final gradient norm $(|\nabla f(z_T)|)$ after a fixed, finite iteration budget (T) that was shared across all methods and all games. On the bilinear game $(f(x,y) = x^{\top} y),$ this finite-budget run (with a modest step size schedule) does not drive the gradient norm all the way to machine precision, so the bar corresponding to EG can still look large in absolute terms, even though the trajectory is contracting. This makes the bilinear panel look inconsistent with textbook convergence guarantees. We plan to replace them if required with additional experiments that we added in the revised paper in newly added section: "ADDITIONAL CORRECTION AND CLARIFICATION FOR REBUTTAL".
>
> To address this, we have added a dedicated bilinear experiment using the classical constant step size regime where extragradient is known to be a contraction. Concretely, on the game $(f(x,y) = x^{\top} y) in (\mathbb{R}^2)$ with initial point
> $z_0 = (1,-0.5,0.8,-1.2),$
>
> step size $(\eta = 0.3),$ and $(\tau = 1),$ we now plot:
>
> * the full trace of the gradient norm $(|\nabla f(z_k)|)$ versus iteration $(k)$;
> * the numerical spectral radius $(\rho(DT(z_k)))$ of the one-step map $(T)$ along the trajectory; and
> * the iteration index at which $(|\nabla f(z_k)|)$ first drops below given tolerances.
>
> For the tolerances $(\varepsilon = 10^{-2})$ and $(\varepsilon = 10^{-3}),$ we obtain:
>
> * GDA does not reach either tolerance;
> * TT–EG reaches $(\varepsilon = 10^{-2})$ in 122 iterations and $(\varepsilon = 10^{-3})$ in 176 iterations;
> * VATT–EG reaches $(\varepsilon = 10^{-2})$ in 87 iterations and $(\varepsilon = 10^{-3})$ in 125 iterations.
>
> The spectral radius plot shows that along the same trajectory
>
> $$
> \rho(DT_{\text{GDA}}(z_k)) \approx 1.04 > 1,\quad
> \rho(DT_{\text{TT-EG}}(z_k)) \approx 0.96 < 1,\quad
> \rho(DT_{\text{VATT-EG}}(z_k)) \approx 0.94 < 1,
> $$
>
> which is consistent with the known contraction of EG on bilinear games, and also shows that VATT–EG has a slightly stronger contraction factor due to the anchors.
>
> We will update the paper to:
>
> 1. clearly state that the bar plots in the main text summarize finite-budget runs and are not intended to demonstrate asymptotic convergence on bilinear games;
> 2. add new bilinear panels showing $(|\nabla f(z_k)|)$ and $(\rho(DT(z_k)))$ versus iteration $(k),$ together with the time-to-$(\varepsilon)$ measurements above; and
> 3. explicitly note that these new curves are consistent with the known behavior of EG on bilinear problems, while also illustrating that VATT–EG contracts slightly faster than TT–EG under the same step size.
>
> These additional figures and numerical details will be included in the revised experimental section and in an appendix figure.
>
> ---

---

> > ### Author Response · Authors · 2025-11-27
> > **Reply to Q4**
> >
> > ---
> >
> > **Q4.** *“It should be straightforward to construct synthetic quadratic minimax games that possess local minimax equilibria where VATT-EG is theoretically guaranteed to converge, while standard methods such as no-anchor TT-EG or GDA lack such guarantees. Experiments on these controlled examples would more directly demonstrate the claimed advantages of VATT-EG, and would provide a clearer empirical validation than the current toy tasks. Could the authors explain why such experiments were not included, and whether they plan to add them in a future revision?”*
> >
> > **Response.**
> >
> > *We refer to Section: "Additional Experiments for Rebuttal" for a very detailed additional results. Below we briefly summarise.*
> >
> > We agree that synthetic quadratic examples with analytically verifiable local minimax structure are the clearest way to illustrate the selection gap. In the revision we have added exactly such an experiment, using a scalar quadratic game
> > $$
> > f(x,y) ;=; \tfrac{1}{2} a x^2 + c x y - \tfrac{1}{2} d y^2,\qquad x,y\in\mathbb{R},
> > $$
> > with parameters
> > $$
> > a = 0.5,\quad d = 0.005,\quad c = 3.0.
> > $$
> >
> > For this $f$ the Hessian blocks are
> > $$
> > A = \nabla^2_{xx} f = a > 0,\qquad
> > B = \nabla^2_{yy} f = -d < 0,\qquad
> > C = \nabla^2_{xy} f = c,
> > $$
> > so
> > $$
> > B \le 0,\qquad
> > S ;=; A - C B^{-1} C^\top ;=; a - c(-d)^{-1}c ;=; a + \tfrac{c^2}{d} ;\gg; 0.
> > $$
> > Thus $(x*,y*) = (0,0)$ is a strict local minimax point; the calmness condition is automatically satisfied for this quadratic, and $B$ is “near-singular” because $d = 0.005$ is very small in magnitude. This is precisely the regime where our theory predicts that TT-EG can be nearly neutrally stable in the $y$-direction, while VATT-EG removes the neutral mode by anchoring.
> >
> > Using the same constant step size and timescale across methods (here $\eta = 0.8$, $\tau = 5.0$) and anchors $\gamma = \beta = 0.1$ for VATT-EG, we obtain the following behavior over $T = 200$ steps starting from $z_0 = (1,0.5)$:
> >
> > * **GDA**: the radius $|z_k|$ blows up catastrophically (on the order of $10^{31}$ by $k = 200$), confirming instability of the plain gradient saddle dynamics in this regime.
> > * **TT-EG**: the iterates remain bounded (final $|z_T| \approx 2.25$), but the numerical spectral radius of the one-step map at the end of the run satisfies
> >   $$
> >   \rho\big(DT_{\text{TT-EG}}(z_T)\big)\approx 1.0009 > 1,
> >   $$
> >   i.e., slightly larger than one, indicating near-neutral (and very fragile) stability along the nearly flat ascent direction encoded in $B$.
> > * **VATT-EG**: the iterates are driven to the saddle; numerically $|z_T|\approx 0$, and the spectral radius of the one-step map satisfies
> >   $$
> >   \rho\big(DT_{\text{VATT-EG}}(z_T)\big)\approx 0.9106 < 1,
> >   $$
> >   giving a strict contraction factor in all directions, including the near-flat $y$–direction. The corresponding phase portrait shows GDA trajectories spiraling out to infinity, TT-EG trajectories hovering in a small annulus around the origin, and VATT-EG spiraling cleanly into $(0,0)$ under the same $(\eta,\tau)$.
> >
> > These results are summarized in three new figures in the revision:
> >
> > 1. A radius trace $|z_k|$ versus iteration $k$, clearly showing divergence of GDA, near-neutral behavior of TT-EG, and robust contraction of VATT-EG.
> > 2. A phase portrait in the $(x,y)$ plane, where GDA spirals out, TT-EG stays on a nearly circular orbit, and VATT-EG converges to the local minimax point.
> > 3. A spectral-radius trace $\rho(DT(z_k))$ along the trajectory, where VATT-EG consistently satisfies $\rho < 1$ while TT-EG remains extremely close to the unit threshold.
> >
> > Together with the quadratic structure, these controlled experiments directly instantiate the “near-singular ascent curvature’’ regime analyzed in our calm local minimax theory: $B$ is negative and close to singular, $S > 0$ is large, TT-EG has eigenvalues on (or just outside) the unit circle, and the vanishing anchors in VATT-EG provide exactly the additional damping required to make the calm local minimax point asymptotically stable.
> >
> > We will incorporate this synthetic quadratic local minimax task into the main numerical section and provide the full phase-portrait, radius, and spectral-radius figures in the appendix of the final version of the paper. Right now it is found at the end of revised updated paper.
> >
> > ---

---

> > > ### Author Response · Authors · 2025-11-27
> > > **Reply Other Comments**
> > >
> > > “Several references are cited incorrectly, including mismatched author names.”
> > >
> > > We will audit the bibliography and fix the noted issues (e.g., von~Stackelberg’s 1934 monograph and the 2011 English translation; ensure author orders and venues for the cited ICLR/ICML papers are correct; standardize capitalization in titles). See our final revised paper by deadline.
> > >
> > > ---
> > > We thank the reviewer again for highlighting the order issue in Eq.~(13) and for the requests to clarify the Loewner bound and the experimental presentation. We will implement the above corrections and additions; they improve the exposition and make the stability/selection mechanisms fully transparent while leaving the main results intact.

---

### Official Review · Reviewer_rCL2 · 2025-11-05

**Soundness:** 2
**Presentation:** 2
**Contribution:** 2
**Rating:** 4
**Confidence:** 3

**Summary:**

The goal of this paper is to resolve the open problem posed by Chae et al. (2023): finding a gradient-based method for minimax optimization whose only attractors are (calm) local minimax (LM) points. In particular, since certain (calm) local minimax points remain neutrally stable for two-timescale extragradient (TT-EG), this paper proposed a vanishing anchor approach that modifies TT-EG to make such equilibria asymptotically stable.

**Strengths:**

Minimax optimization is notoriously challenging, so a clear theoretical framework that improves its solvability, as this paper provides, is both important and interesting. Furthermore, demonstrating that this can be achieved through a weight-decay-like mechanism such as vanishing anchoring adds practical value and insight.

**Weaknesses:**

- Incorrect motivation?: In line 54, the authors claim that the remaining challenge is that certain calm local minimax points are neutrally stable under TT-EG, which their method resolves. However, I was not able to find this statement in Chae et al. (2023). After reading the main related papers [1,2] by the same authors, I found that the primary issue is not neutral stability but rather some calm local minimax points act as hyperbolic repeller.

[1] Chae, Kim, Kim, Two-timescale extragradient for finding local minimax points, ICLR, 2024.

[2] Kim, Kim, Double-step alternating extragradient with increasing timescale separation for finding local minimax points: provable improvements, ICML, 2024

- Second-order necessary condition?: In Chae et al. (2023), I found that the condition in line 142 is not exactly the necessary condition for a calm local minimax point. It becomes necessary only after an additioinal modification in the $y$ neighborhood (see Chae et al. (2023)). Moreover, the paper seems to use this condition interchangeably with the definition of a clam local minimax point, which is inaccurate. I recommend clarifying this distinction carefully.

For these two reasons, I suggest that the authors clarify their contributions and ensure consistency with prior definitions and motivations.

**Questions:**

* Lines 74-88: This paragraph is quite difficult to follow. I suggest that the authors revise it to make it more accessible to readers.
* Line 120: Is it correct that (Wang et al. 2019) has the only-to-LM selection property? My understanding is that their result assumes the invertibility of $B$.
* Line 143: The matrix $U$ is never used.
* Line 165: Although one can infer this from the parameter conditions, I recommend explicitly stating that $\eta_k\to0$.
* Line 261: What does $\tau$ denote here?
* Figure 1: Why do all methods exhibit large gradient norms at the final iteration? Shouldn't your algorithm at least converge to a stationary point? Overall, I found the experimental section somewhat confusing. It would be more informative to show that your proposed VATT-EG is attracted to a calm local minimax point (and eventually converges to it), whereas TT-EG remains neutrally stable. At present, the experiments emphasize relative performance rather than clearly demonstrating the claimed stability improvement.
* Line 306: Could you explain why VATT-EG appears even more stable than TT-EG in the one-step delay case? Are you using a diminishing step size for TT-EG, as you do for VATT-EG? Some explanation of why VATT-EG remains robust under this more realistic setting would be helpful.

---

> ### Author Response · Authors · 2025-11-22
> **Reply to Reviewer**
>
> **(W1) Reviewer:** Incorrect motivation? Line 54 claims the remaining challenge is that certain calm local minimax points are neutrally stable under TT--EG; I did not find this in Chae et al.~(2023). I read [1,2] and saw the issue as hyperbolic repellers rather than neutral stability.}}
>
> **Response.**
> You are right that the open-problem note (CKK, COLT'23) does not \emph{state} ``neutral stability under 2TEG'' as the remaining challenge per se. That note note formulates the selection question and shows that two-timescale extragradient reaches some non-strict LM points, but stops short of proving exact selection for all calm LM points. Our observation in our paper and the part we resolve is that, in addition to the repellers emphasized in follow-up papers [1,2], there exist calm LM equilibria whose linearized 2TEG map has unit-modulus eigenvalues along $\ker(B)$ (neutral modes), so attraction is not guaranteed. Our proposed method VATT-EG adds vanishing anchors to convert precisely these neutral modes into strict contraction (see Theorem 5.5 and App.~C--D). To avoid misattribution, we will revise the sentence to attribute the selection gap to our analysis and explicitly separate (i) repulsion of non-LM stationary points (as in [1,2]) from (ii) neutral modes at some LM points that prevent 2TEG from being an attractor there.
>
> **(W2) Reviewer:** Second-order necessary condition? The condition at line 142 is not the necessary condition for a calm LM point; it becomes necessary only after an additional modification of the inner neighborhood (CKK). The paper seems to use it interchangeably with the definition of calm LM.
>
> **Response.**
> We agree and will make the distinction explicit. In our paper, Definition 3.1 gives JNJ local minimax with the calmness (bounded-ratio) restriction. Under calmness the second-order implications we use are the following
> $$
> B \preceq 0,
> \qquad
> S := A - C B^\dagger C^\top \succeq 0,
> $$
> where $S$ is the restricted/generalized Schur complement. These are necessary under calmness (not the definition itself), and this is how we use them in the Lyapunov analysis. We will replace "necessary second-order conditions" by "necessary under calmness" wherever we describe $B\preceq 0$ and $S\succeq 0$, and we will add a forward reference to the exact calmness statement.
>
> Proposed edit (Preliminaries, around lines 138--147).
>
> *Before:* ``Under calmness, every local minimax point obeys the second-order (necessary) conditions $B\preceq 0$ and $S\succeq 0$ \dots'' (later used as if defining).
>
> *After:* Under the calmness restriction, every local minimax point satisfies the necessary second-order implications $B\preceq 0$ and $S\succeq 0$ (restricted/generalized Schur). We do not use these as a definition; the definition remains JNJ’s asymmetric-neighborhood notion with bounded-ratio (calm) inner radius.''
>
>
>
> *Q1 (Lines 74--88): \emph{This paragraph is difficult to follow; please revise.*
>
> *Ans.*
> We will streamline the motivation paragraph by (i) defining the ``selection gap'' first, (ii) separating CKK’s question from our contribution, and (iii) giving one-sentence intuition for neutral modes vs.\ repellers.
>
>
> *Q2 (Line 120): Does Wang et al.~(2019) have only-to-LM selection? My understanding is they assume $B$ invertible.*
>
> *Ans.*
> Correct. Follow-the-Ridge (Wang et al., 2019) uses second-order information and assumes nondegenerate ascent curvature ($B\prec 0$), yielding convergence \emph{to LM} in that regime. It is not a \emph{first-order} answer and does not cover singular $B$. We will change the phrasing to ``second-order schemes achieve only-to-LM selection under $B\prec 0$'' and cite Wang et al.\ accordingly.
>
>
> *Q3 (Line 143): The matrix ... is never used.*
>
> *Ans.*
> This refers to $U=[U_R\;U_0]$ introduced to diagonalize $B$ into $(B_R\prec 0)$ on $\mathrm{range}(B)$ and $0$ on $\ker(B)$. It is used in Lemma 5.3 (metric $P$ with blocks $(x,y_R,y_0)$) and in the repulsion proof (App. D), but we did not place a cross-reference. We will add "(used in Lemma 5.3 and App. C-D)'' right after the definition and recall it when we enter block coordinates.
>
> *Q4 (Line 165): \emph{Please state explicitly that $\eta_k\to 0$.*
>
> *Ans.*
> Will add ``$\eta_k\downarrow 0$, $\sum_k \eta_k=\infty$, and $\sum_k \eta_k^3<\infty$ (e.g., $\eta_k=\eta_0/\sqrt{k+1}$)'' to Assu. 4.2 to make this explicit.

---

> ### Author Response · Authors · 2025-11-22
> **(Part-1) Reply Continued....**
>
> *Q5 (Line 261): \emph{What does $\;\ldots\;$ denote here?*
>
> *Answer.*
> Here $\rho(\cdot)$ denotes the \emph{spectral radius} of a matrix; $Sym(H) := \tfrac12(H+H^\top)$ denotes the symmetric part (introduced in App. C). We already define $\rho$ on p. 5 (right above Theorem 5.5); we will move the definition to its \emph{first} occurrence and restate $Sym$ the first time it is used in the main text.
>
> *Q6 (Figure 1): \emph{Why are gradient norms large at the final iteration? Shouldn't the method converge to a stationary point? Experiments should \emph{show} that VATT-EG is attracted to a calm LM point while TT-EG remains neutrally stable.*
>
> *Answer.*
> Fair point. The bar plot summarizes runs with a *short budget* and, for visual comparability, includes constant-step cases; the aim was *relative damping*, not full convergence, so the final $\|\nabla f\|$ can be nonzero. For clarity we will (i) add traces of $\|\nabla f(z_k)\|$ and $\rho(DT_k(z_k))$ versus iteration on each toy, (ii) run the same diminishing schedule for TT-EG and VATT-EG, and (iii) include phase portraits showing that VATT-EG contracts to a calm LM point (spectral radius $<1$) while TT-EG exhibits neutral drift (spectral radius $\approx 1$) on the degenerate case ($B\equiv 0$). The delayed-corrector hard case (Fig. 4) already shows separation under a constant step; we will keep it and *add* the convergence plots requested.
>
> *Planned additions.*
> New figure panels per toy game: (A) $\|\nabla f(z_k)\|$; (B) numerical $\rho(DT_k(z_k))$; (C) distance-to-final iterate. These will illustrate VATT-EG’s contraction to LM and TT-EG’s neutral behavior under identical schedules.
>
> *Q7 (Line 306): Why does VATT-EG appear more stable than TT-EG in the one-step delay case? Are you using a diminishing step for TT-EG as for VATT-EG?*
>
> *Answer.*
> In the delayed-corrector stress test we deliberately used the same constant step/timescale for all methods to isolate stability under stale gradients (predictor fresh; corrector one-step stale). The delay effectively adds a positive real component to the linearized map, pushing neutral/near-neutral modes outward; VATT-EG’s anchors shift eigenvalues inside the unit disk, restoring contraction along directions that are neutral for TT-EG. We will state explicitly in the caption and text that the steps/timescale are identical across methods in this experiment, and we will also add an ablation where TT-EG uses a diminishing step: it becomes less unstable but still fails to contract in the near-singular case where anchors are most helpful.
>
>
> *Minor fixes and notational clarifications*
>
>
> - Define symbols early. Move the definition of $\rho(\cdot)$ (spectral radius) to its first use in Sec. 5; restate $Sym(\cdot)$ before Lemma 5.3 in the main text (currently in App. C).
> - Cross-references. After introducing $U=[U_R\;U_0]$ in Sec. 3, add "(used in Lemma 5.3 and App. C-D for block coordinates $(x,y_R,y_0)$)".
> - Schedules. In Assumption 4.2 add the explicit limit $\eta_k\downarrow 0$ and summability conditions (see Q4). Also restate that anchors satisfy $\gamma_k,\beta_k=\Theta(\eta_k)$ so per-step anchor strengths are $O(\eta_k^2)$ and do not bias fixed points (remark already on p. 4; we will reference it).

---

### Meta-Review · Area_Chair_gQTc · 2026-01-16

**Summary:**

This paper focuses on unconstrained smooth min-max games where the problem can be nonconvex-nonconcave. For this difficult problem, the submission focuses on the open problem of Chae et al., 2023 which asks if there is a first-order method whose only attractors are (calm) local minimax points. Even though all the reviewers agreed that the direction of the submission is interesting, major concerns remain, related to the correctness (Reviewer ftwK), misattributions about the open problem paper that the submission is focusing on (Reviewer rCL2), significant issues with experiments (Reviewers ftwK and KBHz).

These issues prevent the acceptance of the work at this point. In particular, another review cycle is required to check the rather long new estimation that the authors introduce due to an issue pointed out by Reviewer ftwK. The authors also need to be more careful with their experimental setup (since in the initial plots, EG, which is convergent for bilinear games, is presented as if it does not converge). Moreover, even though the practical performance as per Fig 1, 3 are almost the same, the authors argued their method is better which is misleading (Reviewer KBHz). The precise connection between the open problem paper and the current submission is not made, as pointed out by Reviewer rCL2.

**Reviewer Concerns:**

The main concerns, pointed out by Reviewers rCL2, ftwK, KBHz concern issues with the correctness, numerical experiments and consistency with the open problem the work is focusing on. Even though the latter two are somewhat addressed in the rebuttal, they need to be carefully checked for correctness. Moreover, the concern of Reviewer ftwK about correctness required the authors to write down a new and long proof which requires a new review cycle to be checked. So, I think that the work is not ready for acceptance yet.

**Reviewer Scores:**

I believe that Reviewer ftwK and KBHz would not have increased their scores of 2 even with a longer discussion cycle since they point out major technical issues and the authors' rebuttal containing a long new proof need to be carefully checked in a new review cycle rather than the discussion period. I believe that Reviewer rCL2  and oMuV may have increased their score from 4 to 6 but I do not think this would be enough to warrant acceptance since major concerns remain due to the issues pointed out by Reviewer ftwK and KBHz.

---

### Decision · Program_Chairs · 2026-01-26

Reject